# AeroCom phase III multi-model evaluation of the aerosol lifecycle and optical properties using ground and space based remote sensing as well as surface in situ observations

Jonas Gliß[1], Augustin Mortier[1], Michael Schulz[1], Elisabeth Andrews[2], Yves Balkanski[3], Susanne E. Bauer[20,19], Anna M. K. Benedictow[1], Huisheng Bian[4,5], Ramiro Checa-Garcia[3], Mian Chin[5], Paul Ginoux[6], Jan J. Griesfeller[1], Andreas Heckel[7], Zak Kipling[9], Alf Kirkevåg[1], Harri Kokkola[10], Paolo Laj[11], Philippe Le Sager[12], Marianne Tronstad Lund[15], Cathrine Lund Myhre[13], Hitoshi Matsui[14], Gunnar Myhre[15], David Neubauer[16], Twan van Noije[12], Peter North[7], Dirk J. L. Olivié[1], Samuel Rémy[21], Larisa Sogacheva[17], Toshihiko Takemura[18], Kostas Tsigaridis[19,20], and Svetlana G. Tsyro[1]

[1]Norwegian Meteorological Institute, Oslo, Norway
[2]Cooperative Institute for Research in Environmental Sciences,University of Colorado, Boulder, Colorado, USA
[3]Laboratoire des Sciences du Climat et de l'Environnement, LSCE/IPSL, CEA-CNRS-UVSQ, Gif sur Yvette Cedex, France
[4]Maryland Univ. Baltimore County (UMBC), Baltimore, MD, USA
[5]NASA Goddard Space Flight Center, Greenbelt, Maryland, USA
[6]NOAA, Geophysical Fluid Dynamics Laboratory, Princeton, NJ, USA
[7]Dept. of Geography, Swansea University, Swansea, UK
[9]European Centre for Medium-Range Weather Forecasts, Reading, UK
[10]Atmospheric Research Centre of Eastern Finland, Finnish Meteorological Institute, Kuopio, Finland
[11]Univ. Grenoble Alpes, CNRS, IRD, Grenoble INP, Institute for Geosciences and Environmental Research (IGE), Grenoble, France
[12]Royal Netherlands Meteorological Institute, De Bilt, the Netherlands
[13]NILU -Norwegian Institute for Air Research, Kjeller, Norway
[14]Graduate School of Environmental Studies, Nagoya University, Nagoya, Japan
[15]CICERO Center for International Climate and Environmental Research, Oslo, Norway
[16]Institute for Atmospheric and Climate Science, ETH Zurich, Zurich, Switzerland
[17]Finnish Meteorological institute, Climate Research Program, Helsinki, Finland
[18]Research Institute for Applied Mechanics, Kyushu University, 6-1 Kasuga-koen, Kasuga, Fukuoka, Japan
[19]Center for Climate Systems Research, Columbia University, New York, USA
[20]NASA Goddard Institute for Space Studies, New York, USA
[21]HYGEOS, Lille, France

**Correspondence:** Jonas Gliß (jonasg@met.no)

**Abstract.**

Within the framework of the AeroCom (Aerosol Comparisons between Observations and Models) initiative, the state of the art modelling of aerosol optical properties is assessed from 14 global models participating in the phase III control experiment (AP3). The models are similar to CMIP6 / AerChemMIP Earth System Models (ESMs) and provide a robust multi-model ensemble. Inter-model spread of aerosol species lifetimes and emissions appears to be similar to that of mass extinction coefficients (MECs), suggesting that aerosol optical depth (AOD) uncertainties are associated with a broad spectrum of parameterised aerosol processes.

Total AOD is approximately the same as in AeroCom phase I (AP1) simulations. However, we find a 50% decrease in the OD of black carbon (BC), attributable to a combination of decreased emissions, lifetimes and BC MEC. Relative contributions from sea salt (SS) and dust (DU) have shifted from approximately equal in AP1 to SS contributing about 2/3 of the natural aerosol optical depth (OD) in AP3. This shift is linked with a decrease in DU mass burden, a lower DU MEC, and a slight decrease in DU lifetime, suggesting coarser DU particle sizes in AP3 compared to AP1.

Relative to observations, the AP3 ensemble median and most of the participating models underestimate all aerosol optical properties investigated, that is, total AOD as well as fine and coarse AOD ($AOD_f$, $AOD_c$), Ångström exponent (AE), dry surface scattering ($SC_{dry}$) and absorption ($AC_{dry}$) coefficients. Compared to AERONET, the models underestimate total AOD by circa $21\% \pm 20\%$ (as inferred from the ensemble median and interquartile range). Against satellite data, the ensemble AOD biases range from -37% (MODIS-Terra) to -16% (MERGED-FMI, a multi-satellite AOD product), which we explain by differences between individual satellites and AERONET measurements themselves. Correlation coefficients (R) between model and observation AOD records are generally high (R > 0.75), suggesting that the models are capable of capturing spatio-temporal variations in AOD. We find a much larger underestimate in coarse $AOD_c$ (~-45% $\pm$ 25%) than in fine $AOD_f$ (~-15% $\pm$ 25%) with slightly increased inter-model spread compared to total AOD. These results indicate problems in the modelling of DU and SS. The $AOD_c$ bias is likely due to missing DU over continental land-masses (particularly over the US, SE-Asia and S-America), while marine AERONET sites and the AATSR SU satellite data suggest more moderate oceanic biases in $AOD_c$.

Column AEs are underestimated by about $10\% \pm 16\%$. For situations where measurements show AE > 2, models underestimate AERONET AE by circa 35%. In contrast, all models (but one) exhibit large overestimates in AE when coarse aerosol dominates (bias ca +140% if observed AE < 0.5). Simulated AE does not span the observed AE variability. These results indicate that models overestimate particle size (or underestimate the fine mode fraction) for fine dominated aerosol and underestimate size (or overestimate the fine mode fraction) for coarse dominated aerosol. This must have implications for lifetime, water uptake, scattering enhancement and the aerosol radiative effect, which we can not quantify at this moment.

Comparison against GAW in situ data results in mean bias and inter-model variations of $-35\% \pm 25\%$ and $-20\% \pm 18\%$ for $SC_{dry}$ and $AC_{dry}$, respectively. The larger underestimate of $SC_{dry}$ than $AC_{dry}$ suggests the models will simulate an aerosol single scattering albedo that is too low. The larger underestimate of $SC_{dry}$ than ambient air AOD is consistent with recent findings that models overestimate scattering enhancement due to hygroscopic growth. The broadly consistent negative bias in AOD and surface scattering suggests an underestimate of aerosol radiative effects in current global aerosol models.

Considerable inter-model diversity in the simulated optical properties is often found in regions that are, unfortunately, not or only sparsely covered by ground based observations. This includes, for instance, the Sahara desert, Amazonia, central Australia and the South Pacific. This highlights the need for a better site coverage in the observations, which would enable us to better assess the models, but also the performance of satellite products in these regions.

Using fine mode AOD as a proxy for present day aerosol forcing estimates, our results suggest that models underestimate aerosol forcing by circa -15%, however, with a considerably large interquartile range suggesting a spread between -35% and +10%.

# 1 Introduction

The global aerosol remains one of the largest uncertainties for the projection of future Earth's climate, in particular because of its impact on the radiation balance of the atmosphere (IPCC, 2014). Aerosol particles interact with radiation through scattering and absorption, thus directly altering the atmosphere's radiation budget (aerosol-radiation interactions, or ARI). Moreover, they serve as cloud condensation nuclei (CCN) and can thus influence further climate relevant components such as clouds and their optical properties (e.g., cloud droplet number concentrations, cloud optical depth) and lifetime as well as cloud coverage and precipitation patterns (aerosol-cloud interactions, or ACI) (IPCC, 2014). Since 2002, the "Aerosol Comparisons between Observation and Models" (AeroCom) project has attempted to federate global aerosol modelling groups to provide state-of-the art multi-model evaluation and, thus, to provide updated understanding of aerosol forcing uncertainties and best estimates. Multi-model ensemble results have often been shown to be more robust than individual model simulations, outperforming them when compared with observations. This paper attempts to provide a new reference, including multi-model ensemble median fields to inform further model development phases.

Aerosol optical properties such as the aerosol scattering and absorption coefficients, the aerosol optical depth (AOD) and the Ångström exponent (AE) are important components of aerosol direct forcing calculations, as they determine how aerosols interact with incoming and outgoing long and shortwave radiation. A special case is aerosol absorption, because it is capable of changing the sign of aerosol forcing. Improved insight about aerosol optical properties, including their spatial and temporal distributions, would be very helpful to better constrain the aerosol radiation interactions. The evaluation of these parameters is thus the focus of this paper.

A challenging part of modelling the global aerosol is its comparatively high variability in space and time (e.g., Boucher et al., 2013), as compared to well-mixed greenhouse gases such as carbon dioxide and methane. The radiative impact aerosols exert depends on the amount and the properties of the aerosol. Emissions, secondary formation of aerosol and lifetime combined lead to different amounts of aerosol in transport models. In addition, atmospheric aerosol particles undergo continuous alteration (e.g., growth, mixing) due to micro-physical processes that occur on lengths and timescales that cannot be resolved by global models, such as nucleation, coagulation, gas-to-particle conversion or cloud processing.

Natural aerosols constitute a large part of the atmospheric aerosol. They are dominated by sea salt (SS) and dust (DU) which make up more than 80% of the total aerosol mass. Natural aerosol precursors include volcanic and biogenic sulphur ($SO_4$), volatile organic compounds (BVOCs) as well as BC and OA from wildfires. Sea salt and dust emissions are strongly dependent on local meteorology and surface properties and, thus, require parameterisations in global models with comparatively coarse resolution. These parameterisations are sensitive to simulated near-surface winds, soil properties (in case of dust) and model resolution (e.g., Guelle et al., 2001; Laurent et al., 2008). Major sources of natural $SO_4$ aerosol are marine emissions of dimethyl-sulfide (DMS) and volcanic $SO_2$ emissions (e.g., Seinfeld and Pandis, 2016). Uncertainties in natural aerosol emissions constitute a major source of uncertainty for estimates of the radiative impact of aerosols on the climate system (e.g., Carslaw et al., 2013), mainly because of non-linearities in the aerosol-cloud interactions and in the resultant cloud albedo effect (Twomey, 1977).

Major absorbing species are black carbon, followed by dust and, to a certain degree, organic aerosols (e.g., Samset et al., 2018, and references therein). Also anthropogenic dust may exert forcing on the climate system (e.g., Sokolik and Toon, 1996). The absorptive properties of dust aerosol are dependent on the mineralogy and size of the dust particles, resulting in some dust types being more absorbing than others (e.g., Lafon et al., 2006). This has direct implications for forcing estimates (e.g., Claquin et al., 1998). Several measured parameters can be used to evaluate model simulations of aerosol optical properties.

AOD is the vertically integrated light extinction (absorption + scattering) due to an atmospheric column of aerosol. AAOD (the absorption aerosol optical depth) is the corresponding equivalent for the absorptive power of an aerosol column and tends to be small relative to AOD (ca $5 - 10\%$ of AOD). Both AOD (dominated by scattering) and AAOD (absorption) are of particular relevance for aerosol forcing assessments (e.g., Bond et al., 2013). Remote sensing of these parameters by sun photometers, for instance, within the Aerosol Robotic Network (AERONET Holben et al., 1998), or via satellite borne instruments have provided an enormous observational database to compare with model simulations.

The AE describes the wavelength dependence of the light extinction due to aerosol and can be measured via remote sensing using AOD estimates at different wavelengths. AE depends on the aerosol species (and state of mixing), due to differences in the refractive indices and size domains (e.g., Seinfeld and Pandis, 2016). It is a qualitative indicator of aerosol size since it is inversely related to the aerosol size (i.e., smaller AE suggests larger particles). However, for mid-visible wavelengths (e.g., around $0.5\,\mu$m, as used in this paper), the spectral variability of light extinction flattens for particle sizes exceeding the incident wavelength. This can create considerable noise in the AE versus size relationship, especially for multi-modal aerosol size distributions, as discussed in detail by Schuster et al. (2006). Global AE values, which combine data from regions dominated by different aerosol types, have potential to further complicate the interpretation of model simulated AE in comparison with observations. Nonetheless, the comparison of modelled AE with observations can still provide qualitative insights into the modelled size distributions.

Model and observational estimates of fine and coarse mode AOD can provide another view onto the light extinction in both size regimes. This is because these parameters also depend on the actual amount (mass) of aerosol available in each mode. The coarse mode is dominated by the natural aerosols (sea salt and dust). Hence, individual assessment of extinction due to fine and coarse particle regimes can provide insights into differences between natural and anthropogenic aerosols. It should be noted that the split between fine and coarse mode is not straightforward in models (for example, some size bins may span the size cut) or for remote sensing instruments which rely on complex retrieval algorithms.

The comparison to surface in situ measurements of scattering and absorption coefficients offers a valuable performance check of the models, independent of remote sensing. One factor that impacts both remote sensing and in situ measurements is water uptake by hygroscopic aerosols. In general, water uptake will enhance the light extinction efficiency (e.g., Kiehl and Briegleb, 1993). This is mostly relevant for scattering, since absorbing aerosols such as dust and black carbon typically become slightly hygroscopic as they age, due to mixing with soluble components (e.g., Cappa et al., 2012). Even at low relative humidity (RH < 40%, a range that is often considered "dry" for the purposes of Global Atmosphere Watch (GAW) in situ measurements GAW Report 227, 2016) aerosol light scattering can be enhanced by up to 20% due to hygroscopic growth (e.g., Zieger et al., 2013). Recent work showed that some models tend to overestimate the scattering enhancement factor at low RH (and high RH)

and hence, overestimate the light scattering coefficients at relatively dry conditions (Latimer and Martin, 2019; Burgos et al., 2020).

Kinne et al. (2006) provided a first analysis of modelled column aerosol optical properties of 14 aerosol models participating in the initial AeroCom phase 1 (AP1) experiments. They found that, on a global scale, AOD values from different models compared well to each other and generally well to global annual averages from AERONET (model biases of the order of -20% to +10%). However, they also found considerable diversity in the aerosol speciation among the models, mainly related to differences in transport and water uptake. They concluded that this diversity in component contribution added (via differences in aerosol size and absorption) to uncertainties in associated aerosol direct radiative effects. Textor et al. (2006) used the same model data as Kinne et al. (2006) and focused on the diversities in the modelling of the global aerosol, by establishing differences between modelled parameters related to the aerosol lifecycle, such as emissions, lifetime and column mass burden of individual aerosol species. One important result from Textor et al. (2006) is that the model variability of global aerosol emissions is highest for dust and sea salt, which is attributed to the fact that these emissions were computed online in most models, while the agreement in the emissions of the other species (OA, $SO_4$, BC) were due to the usage of similar emission inventories. Since then, in the framework of AeroCom, several studies have investigated different details and aspects of the global aerosol modelling, focusing on individual aerosol species and forcing uncertainty. However, it became clear that a common base or control experiment was again needed to compare the current aerosol models contributing to assessments such as the Coupled Model Intercomparison Project Phase 6 (CMIP6, Eyring et al., 2016) or the upcoming report of the Intergovernmental Panel on Climate Change (IPCC), against updated measurements of aerosol optical properties and to assess aerosol life cycle differences. This study aims to provide this basic assessment and will also facilitate interpretation of other recent AeroCom phase III experiments.

This study thus investigates modelled aerosol optical properties simulated by the most recent models participating in the AeroCom phase III 2019 control experiment (AeroCom wiki, 2020, in the following denoted AP3-CTRL) on a global scale. It makes use of the increasing amount of observational data which have become available during the past two decades. We extend the assessment by Kinne et al. (2006) and use ground and space-based observations of the columnar variables of total, fine and coarse AOD and AE and, for the first time, surface in situ measurements of scattering and absorption coefficients, primarily from surface observatories contributing to Global Atmospheric Watch (GAW), obtained from the World Data Centre for Aerosols (GAW-WDCA) archive.

This paper is structured as follows. Section 2 introduces the observation platforms, parameters and models used, followed by a discussion of the analysis details for the model evaluation (e.g., statistical metrics, re-gridding and co-location). The results are split into two sections. Section 3 provides an inter-model overview of the diversity in globally averaged emissions, lifetimes, burdens as well as mass-extinction and mass-absorption coefficients (MECs, MACs) and optical depths (ODs) for each model and aerosol species[1]. This is followed by a discussion of the diversity of simulated aerosol optical properties (AOD, AE, scattering and absorption coefficients) in the context of the species specific aerosol parameters (e.g., lifetime, burden, etc.) from each model. Section 4 presents and discusses the results from the comparison of modelled optical properties with the

---

[1]Note that throughout this paper AOD denotes total "aerosol optical depth", while OD denotes "optical depth" of individual species (e.g., $OD_{SO_4}$)

different observational data sets. The observational assessment section ends with a short discussion of the representativity of the results.

## 2 Data and Methods

In this section, we first describe the ground and space based observation networks / platforms and variables that are used in this study (Section 2.1). Section 2.2 introduces the 14 global models used in this paper. Finally, Section 2.3 contains relevant
information related to the data analysis (e.g., computation of model ensemble, co-location methods and metrics used for the model assessment).

### 2.1 Observations

Several ground and space-based observations have been utilised in order to perform a comprehensive evaluation at all scales (Tab. 1). These are introduced in the individual paragraphs below. Figure 1 shows maps of the annual mean values of the vari-
155 ables considered (from some of the observation platforms used). It is discussed below in Sect. 2.1.7. Note that the wavelengths in Table 1 reflect the wavelengths used for comparison with the models, however, the original measurement wavelengths may be different as noted below.

### 2.1.1 AERONET

The Aerosol Robotic Network (AERONET Holben et al., 1998) is a well established, ground-based remote sensing network
based on sun photometer measurements of columnar optical properties. The network comprises several hundred measurement sites around the globe (see Fig. 1a,c,d,e for the 2010 sites). In this paper, cloud screened and quality assured daily aggregates of AERONET AODs, $AOD_f$, $AOD_c$ and AE from the version 3 (Level 2) Sun and SDA products (e.g., O'Neill et al., 2003; Giles et al., 2019)) have been used. No further quality control measures have been applied due to the already high quality of the data. Only site locations below 1000 m altitude were considered in this analysis.
The sun photometers measure AOD at multiple wavelengths. For comparison with the model output (which is provided at 550 nm), the measurements at 500 nm and 440 nm were used to derive the total AOD at 550 nm, using the provided AE data to make the wavelength adjustment (the 500 nm channel was preferred over the 440 nm channel). Similarly, the $AOD_f$ and $AOD_c$ data provided at 500 nm via the AERONET spectral deconvolution algorithm (SDA) product, were shifted to 550 nm using the AE data. The SDA product (O'Neill et al., 2003) computes $AOD_f$ and $AOD_c$ in an optical sense, based on the spectral curvature
of the retrieved AODs in several wavelength channels and assuming bimodal aerosol size distributions. Thus, as pointed out by O'Neill et al. (2003) it does not correspond to a strict size cut at a certain radius, such as the R=0.6 $\mu$m established in the AERONET Inversion product (Dubovik and King, 2000). Compared to the Inversion product, the SDA product used here tends to overestimate the coarse contribution (O'Neill et al., 2003) which suggests that, on average, the effective cut applied in the SDA product is closer to the strict threshold of R=0.5 $\mu$m required from the models within the AP3-CTRL experiment (see
Sect. 2.2 for details). The implications of this difference are discussed in Section 4. It should also be noted the AE provided

by AERONET is calculated from a multi-wavelength fit to the four AERONET measurement wavelengths, rather than from selected wavelength pairs.

Data from the short term DRAGON campaigns (Holben et al., 2018) were excluded in order to avoid giving too much weight to the associated campaign regions (with high density of measurement sites) in the computation of network averaged statistical parameters used in this study. No further site selection has been performed, since potential spatial representativity issues associated with some AERONET sites were found to be of minor relevance for this study (Sect. 4.5).

The sun photometer measurements only occur during daylight and cloud free conditions. Thus, the level 2 daily averages used here represent daytime averages rather than 24h averages (as provided by the models). Because of the requirements for sunlight and no clouds, the diurnal coverage at each site shows a more or less pronounced seasonal cycle depending on the latitude (e.g., only mid-day measurements at high latitudes in winter) and the seasonal prevalence of clouds in some regions. This is a clear limitation when comparing with 24h monthly means output from the models (as done in this study). However, these representativity issues were found to have minor impact for the model assessment methods used in this study (details are discussed in Sect. 4.5).

### 2.1.2 Surface in situ data

Surface in situ measurements of the aerosol light scattering (SC) and absorption coefficients (AC) were accessed through the GAW-WDCA database EBAS (http://ebas.nilu.no/). As with AERONET, only sites with elevations below 1000 m were considered. Annual mean values of scattering and absorption are shown in Figure 1g,h. The in situ site density is highest in Europe, followed by North America, while other regions are poorly represented. The EBAS database also includes various observations of atmospheric chemical composition and physical parameters, although those were not used here. For both scattering and absorption variables, only level 2 data from the EBAS database were used (i.e., quality controlled, hourly averaged, reported at standard temperature and pressure (STP); $T_{std} = 273.15$ K, $P_{std} = 1013.25$ hPa). All data in EBAS have version control, and a detailed description of the quality assurance and quality control procedures for GAW aerosol in situ data are available in Laj et al. (2020). Additionally, for this study, data was only considered if it was associated with the EBAS categories *aerosol* or *pm10*. The *aerosol* category indicates the aerosol was sampled using a whole air inlet, while *pm10* indicates the aerosol was sampled after a 10 $\mu$m aerodynamic diameter size cut.

Invalid measurements were removed based on values in the flag columns provided in the data files. Furthermore, outliers were identified and removed using value ranges of $\{-10, 1000\}$ Mm$^{-1}$ and $\{-1, 100\}$ Mm$^{-1}$ for scattering and absorption coefficients, respectively. The outliers were removed in the original 1h time resolution before averaging to monthly resolution for comparison with the monthly model data.

For the in situ AC data used in this study, most of the measurements are performed at wavelengths other than 550 nm (see Sect. 1 in supplement 2). These were converted to 550 nm assuming an absorption Ångström exponent (AAE) of 1 (i.e., a 1/$\lambda$ dependence, e.g., Bond and Bergstrom, 2006). This is a fairly typical assumption when the spectral absorption is not measured. For about 50% of the sites, absorption was measured at ~530 nm meaning that even if the true AAE had a value of 2, the wavelength-adjusted AC value would only be underestimated by ca 4%. For another 25% of the sites, absorption was measured

at ~670 nm. For these sites the impact of an incorrect AAE value is larger (ca 26% overestimation for an actual AAE of 2 and ca 6% for AAE=1.25). The remaining 25% of sites typically utilized wavelengths between these two values. Schmeisser et al. (2017) suggest that, across a spatially and environmentally diverse set of sites measuring spectral in situ absorption (many included here), that the AAE is typically between 1 and 1.5.

The majority of in situ scattering sites used here included a measurement at 550 nm (see Tab. 2 in supplement 2), so for these data no wavelength adjustment was necessary. The remaining few sites measuring around 520 nm were shifted to 550 nm assuming a scattering AE (SAE) of 1 (we note that this is rather at the lower end of typically measured SAEs, see Andrews et al., 2019). However, we assess the uncertainties similar to those discussed above for AC, indeed, the change in model bias as compared to an assumed SAE=1.5 was found to be <0.5%. As mentioned previously, the in situ measurements are, ideally, made at low RH (RH $\leq$ 40%) but are not absolutely dry (i.e., RH = 0%). Control of sample relative humidity is not always perfect so, depending on site and conditions, the measurement RH could exceed 40%. Because the model data with which the in situ scattering data will be compared is reported at RH = 0%, only measurements at RH $\leq$ 40% were considered to minimize discrepancies due to potential scattering enhancement at higher RH values. While maintaining the measurement RH < 40% is typically assumed to minimize the confounding effect of water on aerosol properties (GAW Report 227, 2016), Zieger et al. (2013) suggest that there may be noticeable scattering enhancement even at RH = 40% for some types of aerosol (see their Figure 5b).

While observations from other platforms and networks relied solely on 2010 data for the model assessment (see Tab. 1), many in situ sites began measurements after 2010 so a slightly different approach was taken in order to maximise the number of sites with monthly aggregated data. For any given in situ site, all data available between 2005 – 2015 was used to compare with the 2010 model output. The climatology for each in situ site was computed requiring at least 30 valid daily values for each of the climatological months over the 10 year period. Prior to that, daily values were computed from the hourly data applying a minimum 25% coverage constraint (i.e., at least 6 valid hourly values per day). It should be noted that the in situ data is collected continuously day and night regardless of cloud conditions and, thus, daily data will represent the full diurnal cycle in most cases. As can be seen in column "Cov" in Tables 1 & 2 of the supplementary material 2, for most of the in situ sites, the 25% coverage constraint for the resampling from hourly to daily was typically met. Note that about half of all available hourly SC measurements in the 2005‐2015 period were not considered here, either because the measured RH exceeded 40% or because RH data were missing in the data files.

A few urban in situ sites were removed from consideration for the model analysis, as these sites are likely not representative on spatial scales of a typical model grid. For scattering coefficients the sites excluded are: Granada; Phoenix; National Capitol - Central, Washington D.C; and for absorption coefficients: Granada; Leipzig Mitte; Ústí n.L.-mesto. After applying the RH constraint, removing urban sites from consideration, and resampling to monthly climatology, data from 39 sites with scattering data and from 39 sites with absorption data (not necessarily the same sites as for scattering) were available for model assessment (see Table 1).

Tables 1 & 2 in the supplementary material 2 provide detailed information about each of the absorption and scattering sites
used. This includes the original measurement wavelengths as well as temporal coverage for the computation of the climatology.

### 2.1.3 Satellite data sets - Introduction

In addition to the ground-based observations, data from four different satellite data sets (MODIS Aqua & Terra, AATSR SU
v4.3 and a merged AOD satellite data set) were used to evaluate optical properties from the AP3 models. The four satellite data
sets are introduced below.

Even though the satellite observations usually come with larger uncertainties and may exhibit potential biases against ground-
based column observations (e.g., Gupta et al., 2018), we believe that it is a valuable addition to not only evaluate models at
ground sites but also incorporate satellite records for an assessment of model performance. The main advantage of satellite
data is the spatial coverage relative to ground-based measurements. Satellites provide more coverage over land masses than
AERONET and in addition, they are the primary observational tool for column optical properties over oceans.

Because of AERONET's reliability and data quality, it is generally accepted as the gold standard for column AOD measure-
ments. Therefore, all four satellites used in this paper, were evaluated against AERONET data, in order to establish relative
biases and correlation coefficients. Details related to this satellite assessment are discussed in supplement 2 and are briefly
mentioned in the introduction sections for each individual satellite below. The results from this satellite assessment are also
available online (see Mortier et al., 2020a), allowing an interactive exploration of the data and results (down to the station
level) and include many evaluation metrics (e.g., various biases, correlation coefficients, root mean square error (RMSE)).
These comparisons of the individual satellites against AERONET provide context for the differences in the model assessments
discussed below in Section 4. It should be noted, however, that the retrieved biases for each satellite data set provide insights
into the performance of each satellite product at AERONET sites, which are land dominated. Satellites often have different
retrieval algorithms over land and ocean (e.g., Levy et al., 2013), and the aerosol retrieval tends to be more reliable over dark
surfaces, such as the oceans, than over bright surfaces, such as deserts (e.g., Hsu et al., 2004).

### 2.1.4 MODIS data

Daily gridded level 3 AOD data from the Moderate Resolution Imaging Spectroradiometer (MODIS) have been used from
both satellite platforms (Terra and Aqua) for evaluation of the models. The merged land and ocean global product (named
*AOD_550_Dark_Target_Deep_Blue_Combined_Mean* in the product files) of the recent collection 6.1 was used. This is an
updated and improved version of collection 6 (e.g., Levy et al., 2013; Sayer et al., 2014). For changes between both data sets,
see Hubanks (2017).

Details about the MODIS data sets used are provided in Table 1. Compared to AERONET, both Aqua and Terra exhibit
positive AOD biases, suggesting an overestimation of ca +9% and +17%, respectively, at AERONET sites and for the year
2010 (for details see supplement 2). The larger overestimate for Terra is in agreement with the findings from Hsu et al. (2004).

### 2.1.5 AATSR SU v4.3 data

The AATSR SU v4.3 data set provides gridded AOD and associated parameters from the Advanced Along Track Scanning Radiometer (AATSR) instrument series, developed by Swansea University (SU) under the ESA Aerosol Climate Change Initiative (CCI). The AATSR instrument on ENVISAT covers the period 2002 – 2012 and in this study, data from 2010 are used. The instrument's conical scan provides two near simultaneous views of the surface, at solar reflective wavelengths from 555 nm to 1.6 µm.

Over land, the algorithm uses the dual-view capability of the instrument to allow estimation without a priori assumptions on surface spectral reflectance (North, 2002; Bevan et al., 2012). Over ocean, the algorithm uses a simple model of ocean surface reflectance including wind-speed and pigment dependency at both nadir and along-track view angles. The retrieval directly finds an optimal estimate of both the AOD at 550 nm, and size, parameterised as relative proportion of fine and coarse mode aerosol. The local composition of fine and coarse mode is adopted from the MACv1 aerosol climatology (Kinne et al., 2013). The local coarse composition is defined by fraction of non-spherical dust and large spherical particles typical of sea salt aerosol, while fine mode is defined by relative fractions of weak and strong absorbing aerosol. A full description of these component models is given in de Leeuw et al. (2015). Further aerosol properties including AE (calculated between 550 and 856 nm) and absorption aerosol optical depth (AAOD, not used in this study) are determined from the retrieved AOD and composition. Aerosol properties are retrieved over all snow-free and cloud-free surfaces. The most recent version AATSR SU V4.3 (North and Heckel, 2017) advances on previous versions by improved surface modelling and shows reduced positive bias over bright surfaces. Retrieval uncertainty and comparison with sun photometer observations show highest accuracy retrieval over ocean and darker surfaces, with higher uncertainty over bright surfaces (e.g., desert, snow) and for large zenith angles (Popp et al., 2016). This study uses the level 3 output, which is provided at daily and monthly 1° x 1° resolution, intended for climate model comparison. Specifically, AATSR SU values for AE and total, fine and coarse AODs are used. The AE calculation is only performed for $0.05 < AOD < 1.5$ due to increased retrieval uncertainty of AE at low and high AODs.

In comparison with AERONET, the AATSR data exhibits an AOD bias of ~-4%, suggesting a slight underestimation of AOD at AERONET sites, in contrast to the two MODIS products used (see Tab. 1). To our knowledge, this AATSR product (SU V4.3) has not been evaluated against AERONET in the literature. Thus, these results comprise an important finding of this study. Biases of $AOD_f$, $AOD_c$ and AE against AERONET were found to be + 1.6%, - 14.7% and + 14.3%, respectively (see web visualisation, Mortier et al., 2020a).

Initial comparisons within the Aerosol CCI project suggest that the fine mode fraction of total AOD may be overestimated over the ocean, with consequently some high bias in AE. The AE provided by AATSR is estimated for the range 550-870 nm, and some difference may be expected also with AERONET derived AE using a different wavelength range (e.g., Schuster et al., 2006).

### 2.1.6 Merged satellite AOD data

The MERGED-FMI data set, developed by the Finnish Meteorological Institute, includes gridded level 3 monthly AOD products merged from 12 available satellite products (Sogacheva et al., 2020). It should be noted that MODIS and AATSR products are considered inside this MERGED-FMI data set. It is available for the period 1995-2017, however, here only 2010 data are used.

Compared to AERONET measurements from 2010, this merged satellite product has shown excellent performance with the highest correlation (R=0.89) among the four satellites used and only a slight underestimation of AOD (bias of -5.4%) at AERONET sites (see supplement 2 and Mortier et al., 2020a). The merging method is based on the results of the evaluation of the individual satellite AOD products against AERONET. Those results were utilised to infer a regional ranking which was then used to calculate a weighted AOD mean. Because it is combined from the individual products of different spatial and temporal resolution, the AOD merged product is characterised by the best possible coverage, compared with other individual satellite products. The AOD merged product is at least as capable of representing monthly means as the individual products (Sogacheva et al., 2020). Standard pixel-level uncertainties for the merged AOD product were estimated as the root mean squared sum of the deviations between that product and other eight merged AOD products calculated with different merging approaches applied for different aerosol types (Sogacheva et al., 2020).

### 2.1.7 Global distribution of optical properties investigated

The previous sections introduced the individual ground and space-based observation records and optical properties variables that will be used in this paper for the model assessment. Figure 1 provides an overview of the global distribution of these optical properties. The displayed global maps show annual mean values of all variables considered, both for the ground-based networks and for a selection of the satellite observations. Fig. 1a,c,d shows yearly average mean values of the observed AERONET AODs (total, coarse and fine, respectively). Column Ångström exponents from AERONET are shown in Fig. 1e. Dust dominated regions such as Northern Africa and Southwest Asia are clearly visible both in the coarse AOD and the AE, but also in the total AOD, indicating the importance of dust for the global AOD signal. The satellite observations of AOD (MERGED-FMI) and AE (ATSR-SU) (Fig. 1b,f) are particularly useful in remote regions and over the oceans where ground-based measurements are less common. Thus, they add substantially to the global picture when assessing models. For example, satellites capture the nearly constant AOD background of around 0.1 over the ocean (mostly arising from sea salt) which cannot be obtained from the land dominated, ground-based observation networks. The AE from AATSR-SU shows a latitudinal southwards decreasing gradient in remote ocean regions, indicating dominance of coarse(r) particle size distributions, which is likely due to cleaner and, thus, more sea salt dominated regions. Transatlantic dust transport results in an increased particle size west of the Sahara (e.g., Kim et al., 2014) as is captured by AATSR-SU. Finally, it is difficult to observe global patterns in the in situ scattering and absorption data due to the limited spatial coverage of the measurements, as can be seen in the lowermost panels (Fig. 1g,h). The differences in the spatial coverage for each observation data set will be important to keep in mind when interpreting the results presented in Sect. 4.

## 2.2 Models

This study uses output from 14 models that are participating in the AeroCom AP3-CTRL experiment. Details on the AeroCom phase III experiments can be found on the AeroCom wiki page (AeroCom wiki, 2020). The wiki also includes information on how to access the model data from the different AeroCom phases and experiments, which is stored in the AeroCom database. Note, that the database location and information about it might change in the future, the intention is however to keep updated information available via the website https://aerocom.met.no (last access: 14.09.2020). Table 2 provides an overview of the models used in this paper. For the AP3-CTRL experiment, modellers were asked to submit simulations of at least the years 2010 and 1850, with 2010 meteorology and prescribed (observed) sea-surface temperature and sea ice concentrations, and using emission inventories from CMIP6 (Eyring et al., 2016), when possible. Details concerning the anthropogenic and biomass burning emissions are given in (CEDS, Hoesly et al., 2018) and (BB4CMIP, van Marle et al., 2017). In this paper, only the 2010 model output is used. The year 2010 was chosen as a reference year by the AeroCom consortium and is used throughout many phase II and III experiments for inter-comparability of different experiments and model generations. The AeroCom phase I simulations (e.g., Dentener et al., 2006; Kinne et al., 2006; Schulz et al., 2006; Textor et al., 2006) used the year 2000 as a reference year. One of the main reasons to update the reference year from 2000 to 2010 was that many more observations became available between 2000 and 2010 and also to account for changes in the present day climate, for instance, due to changing emissions and composition (e.g., Klimont et al., 2013; Aas et al., 2019; Mortier et al., 2020b).

Detailed information about the models on emissions, humidity growth and particularly their treatment of aerosol optics has been collected from the modelling groups through a questionnaire. The tabulated responses are provided in supplementary material 1. The first table (Spreadsheet "Table: General questions") contains general information that applies to the total aerosol, such as mixing assumptions, treatment of clear-sky optics and water uptake parameterisations. The second table (Spreadsheet "Table: Species specific") contains aerosol species specific information such as the complex refractive index at 550 nm, humidity growth factors, particle density, as well as details regarding the emission data sets used. Further information related to OA emissions and secondary formation is provided for most models in a third spreadsheet ("Table: OA details"). In addition, Sect. 4 of supplement 2 provides further information for each of the models, mostly complementary to Table 2.

### 2.2.1 Model diagnostics

Requested diagnostics fields for AP3-CTRL are available online (see, AeroCom diagnostics sheet, 2020). In addition, variables for dry (at RH = 0%) extinction ($EC_{dry}$) and absorption ($AC_{dry}$) coefficients were requested (at model surface level) from the modelling groups participating in this study. These are needed for the comparison with the GAW surface in situ observations (Sect. 2.1.2). Note that in a few cases, some diagnostic fields used in this study could not be provided by some of the modelling groups.

To obtain model values that were comparable with observations, additional processing was required for some variables. The $AOD_c$ fields were not directly submitted but were computed as the difference: $AOD - AOD_f$. The AE fields were computed

from the provided AOD at 440 nm and 870 nm[2] via $AE = -\ln(AOD_{440}/AOD_{870}) / \ln(440/870)$. Dry scattering coefficients ($SC_{dry}$), for the comparison with the surface in situ data were computed via $SC_{dry} = EC_{dry} - AC_{dry}$. Some of the models that provided these data submitted dry EC, but ambient AC (indicated in Table 2). For these models, dry scattering was derived in the same way $SC_{dry} = EC_{dry} - AC_{amb}$ consistent with the idea that absorbing aerosol tends to be hydrophobic. The latter may be violated to some degree for models that include internally mixed BC modes with hydrophilic species, such as $SO_4$. However, an investigation of differences between dry and ambient absorption coefficients revealed that the overall impact on the results is minor, both for models with internally mixed BC modes and for models with externally mixed modes.

Some of the models reported the columnar optical properties based on clear-sky (CS) assumptions, while others assumed all-sky (AS) conditions to compute hygroscopic growth and extinction efficiencies. These choices are indicated in Table 2 and details related to the computation of CS optics can be found in supplement 1.

The following modelled global average values have been retrieved of species specific model parameters to be compared in section 3 to assess lifecycle aspects of model diversity:

1. Emissions and formation of aerosol species (in units of Tg / yr). The secondary aerosol formation of $SO_4$, $NO_3$ and OA by chemical reactions in the atmosphere is difficult to diagnose. Thus, they are diagnosed here from total deposition output.

2. Lifetimes of major aerosol species (in units of days), computed from column burden and provided wet+dry deposition rates. The lifetimes can give insights into the efficiency of removal processes in the models.

3. Global mass burdens (in units of Tg) for each species. These values enable comparisons amongst the models in terms of aerosol amount present on average.

4. Modelled speciated optical depths (ODs) at 550 nm. This unitless quantity provides another way of looking at contributions from different species to total AOD based on their optical properties rather than their burden.

5. Modelled mass extinction coefficients (MECs) at 550 nm for each species (in units of $m^2/g$), calculated by dividing the species optical depth with the corresponding species mass burden (e.g., $OD_{DU}$ / $LOAD_{DU}$). The MEC determines the conversion of aerosol mass to light extinction, and can provide insights into the variability of modelled size distributions or hygroscopicity.

6. Additionally, modelled mass absorption coefficients (MACs) at 550 nm for light absorbing species (BC, DU, OC) are presented. These are calculated by dividing the species absorption optical depth (AAOD) with the corresponding species mass burden (e.g., $AAOD_{BC}$ / $LOAD_{BC}$).

We note again that detailed introductions for each model are provided in supplement 1 and in Section 4 of supplement 2, in addition to the summary Table 2.

---

[2]For GISS-OMA, 550 nm and 870 nm AODs were used for AE calculation as 440 nm AOD data was missing.

## 2.3 Data processing and statistics

Most of the analysis in this study was performed with the software pyaerocom (Github: https://github.com/metno/pyaerocom, Website: https://pyaerocom.met.no/, last access: 14.09.2020). pyaerocom is an open source Python software project that is being developed and maintained for the AeroCom initiative, at the Norwegian Meteorological Institute. It provides tools for harmonisation and co-location of model and observation data, and dedicated algorithms for the assessment of model performance at all scales. Evaluation results from different AeroCom experiments are uploaded to a dedicated website that allows exploration of the model and observation data and evaluation metrics. The website includes interactive visualisations of performance charts (e.g., biases, correlation coefficients), scatter plots, bias maps and individual station and regional timeseries data, for all models and observation variables, as well as bar charts summarising regional statistics. All results from the optical properties evaluation discussed in this paper are available via a web interface (see Mortier et al., 2020c).

The ground and space based observations are co-located with the model simulations by matching with the closest model grid-point in the originally provided model resolution.

In the case of ground-based observations (AERONET and GAW in situ), the model grid-point closest to each measurement site is used. For the satellite observations, both the model data and the (gridded) satellite product are re-gridded to a resolution of $5° \times 5°$ and the closest model grid-point to each satellite pixel is used. The choice of this rather coarse resolution is a compromise, mostly serving the purpose of increasing the temporal representativity (i.e., more data points per grid cell) in order to meet the time resampling constraints (defined below). For the comparison of satellite AODs with models, a minimum AOD of 0.01 was required, due to the increased uncertainties related to satellite AOD retrievals at low column burdens. The low AODs were filtered in the original resolution of the level 3 gridded satellite products, prior to the co-location with the models.

Since many model fields were only available in monthly resolution, the co-location of the data with the observations (and the computation of the statistical parameters used to compare the models) was performed in monthly resolution. Any model data provided in higher temporal resolution was averaged to obtain monthly mean values, prior to the analysis. For the higher resolution observations (see Table 1), the computation of monthly means was done using a hierarchical resampling scheme, requiring at least ~25% coverage. Practically, this means that the daily AERONET data were resampled to monthly, requiring at least 7 daily values in each month. For the hourly in situ data, first a daily mean was computed (requiring at least 6 valid hourly values) and from these daily means, monthly means were computed requiring at least 7 daily values. Data that did not match these coverage constraints were invalidated.

Throughout this paper, the discussion of the results will use two statistical parameters to assess the model performance: the normalised mean bias (NMB) is defined as $\mathrm{NMB} = \frac{\sum_i^N (m_i - o_i)}{\sum_i^N o_i}$ where $m_i$ and $o_i$ are the model and observational mean, respectively, and the Pearson correlation coefficient (R). More evaluation metrics, such as normalised RMSE or fractional gross error are available online in the web visualisation (Mortier et al., 2020c), but are not further considered within this paper.

Section 4.5 presents several sensitivity studies that were performed in order to investigate the spatio-temporal representativity of this analysis strategy, which is based on network-averaged, monthly aggregates. This was done because representativity (or

lack thereof) comprises a major source of uncertainty (e.g., Schutgens et al., 2016, 2017; Sayer and Knobelspiesse, 2019). The focus here was to assess how such potential representation errors affect the biases and correlation coefficients used in this paper to assess the model performance and comparison with other models.

**2.3.1 AeroCom ensemble mean and median**

For all variables investigated in this paper, the monthly AeroCom ensemble mean (ENS-MEAN) and median (ENS-MED) fields were computed and have been made available in the AeroCom database, for future reference. This was done in order to enable an assessment of the AP3 model ensemble, which we consider to represent the most likely modelling output of the state of the art aerosol model versions participating in the AP3-CTRL exercise.

The ensemble fields were computed in a latitude / longitude resolution of $2° \times 3°$, which corresponds to the lowest available model resolution (i.e., of models EC-Earth and TM5, see Table 2). Model fields were all re-gridded to this resolution before the ensemble mean and median were computed. In this paper, only the output from the median model is used. Note that results from the mean model are not further discussed below but are available online (see Mortier et al., 2020c). In addition to the median 450 (50th percentile), also the 25th (Q1) and 75th (Q3) percentiles were computed and evaluated against the observations like any other model. This was done to enable an assessment of model diversity in the retrieved biases and correlation coefficients.

In addition, local diversity fields were computed for each variable by dividing the the interquartile range (IQR = Q1 – Q3) by the ensemble median: $\delta_{\mathrm{IQR}}$ = IQR / median, which corresponds to the central 50% of the models as a measure of diversity (this is different than Kinne et al., 2006, who use the central 2/3). Note that the IQR is not necessarily symmetrical with respect 455 to the median. In order to enable a better comparison with the AP1 results from Textor et al. (2006) and Kinne et al. (2006), a second set of diversity fields were computed as follows: $\delta_{\mathrm{std}} = \sigma$ / (ensemble mean), where $\sigma$ is the standard deviation.

Note that the ensemble AE fields were computed from the individual models' AE fields. In case of the ensemble median, this will give slightly different results compared to a computation of a median based on median 440 and 870 AOD fields. This is because the median computation is done in AE space and not in AOD space.

Please also note that the ensemble total AOD includes results from INCA which are not included in $AOD_f$ and $AOD_c$ (see Tab. 2). This results in a slightly smaller total AOD in the ensemble when inferred from $AOD_f$+$AOD_c$ (which does not include INCA) compared to the computed AOD field (which includes INCA).

**2.3.2 Model STP correction for comparison with GAW in situ data**

One further model processing note: since the GAW in situ measurements are reported at STP conditions (Section 2.1.2), the 465 2010 monthly model data were converted to STP using the following formula:

$$X_{\mathrm{STP}} = X_{\mathrm{amb}} \times \left( \frac{P_{\mathrm{std}}}{P_{\mathrm{amb}}} \right) \cdot \left( \frac{T_{\mathrm{amb}}}{T_{\mathrm{std}}} \right) \tag{1}$$

$X_{\mathrm{STP}}$ and $X_{\mathrm{amb}}$ are the model value of absorption (or scattering) at STP and ambient conditions, respectively. $P_{\mathrm{amb}}$ and $T_{\mathrm{amb}}$ are the ambient air pressure and temperature at the corresponding site location. The correction factor was estimated on a monthly basis, where $P_{\mathrm{amb}}$ was estimated based on the station altitude (using the barometric formula and assuming a standard atmosphere implemented in the python geonum library, Gliß, 2017) and $T_{\mathrm{amb}}$ was estimated using monthly near surface (2m) temperature data from ERA5. This correction may introduce some statistic error mostly due to natural fluctuations in the pressure and possible uncertainties in the ERA5 temperature data. However, we assess this additional uncertainty to be small for the annual average statistics discussed below.

## 3 Results and discussion - Model diversity of aerosol lifecycle and optical properties

The focus of this section is to establish a global picture and to try to understand model diversity in relevant parameters related to the aerosol lifecycle (i.e., global emissions, lifetimes and burdens) as well as the simulated aerosol optical properties (i.e., speciated MECs, MACs and ODs). The goal is to develop an understanding of how, based on the models, processes and parameterisations link emissions to optical properties. A comparison of modelled optical properties with the various observation records is presented in the following Section 4.

Most of the discussion in this section focuses on the model ensemble median and associated diversities ($\delta_{\mathrm{IQR}}$). Section 3.1 focuses on diversity in the treatment of the different aerosol species in the models, starting with an overview of simulated global aerosol emissions, lifetimes and mass burdens (Sect. 3.1.1), followed by a discussion of simulated ODs, MECs and MACs for each species (Sect. 3.1.2). Section 3.2 provides and discusses the global distribution of the simulated aerosol optical properties and their spatial diversity.

### 3.1 Lifecycle and optical properties for each aerosol species

Table 3 provides an overview of global annual mean values of emissions, lifetimes, burdens, ODs, MECs and, where available, MACs, for each aerosol species (i.e., BC, DU, $NO_3$, OA, $SO_4$ and SS) and for each model. Gaps in the table indicate where models did not provide a requested variable. Also included are the median (MED) and diversity estimates ($\delta_{\mathrm{IQR}}$, $\delta_{\mathrm{std}}$) for each species and variable. Note that these are computed directly from the values provided in Table 3, not using the ensemble median fields. For comparison, median and $\delta_{\mathrm{std}}$ from the AeroCom phase 1 (AP1) simulations are provided as well. The colours in the table provide an indication of the sign and bias of the individual model values relative to the AP3 median.

Figure 2 provides a different view of the data provided in Table 3, by illustrating how the diversity of the individual parameters contributes to the resulting model ensemble diversity in species OD, similar to illustrations used earlier in Schulz et al. (2006) (their Figure 8) and Myhre et al. (2013) (their Figure 14). This visualisation makes it easier to link the diversity in speciated ODs with the uncertainty in modelling the processes controlling the OD of each species.

### 3.1.1 Aerosol lifecycle: from emissions to mass burdens

As explained above, global aerosol emission and formation (in Table 3) were estimated either using the provided emission fields as for primary aerosols BC, DU, SS and POA, or using the equivalent total emissions as for $SO_4$, OA and $NO_3$ based on total deposition. For simplicity we also call the equivalent total emissions, which include secondary formation from precursors, "emissions" in this section. Note, that only major aerosol species are included in our study; aerosol precursor species that are provided by some few models (e.g., $NO_x$, $NH_4$ or VOCs) are not analysed.

Emissions are highest for sea salt (4980 Tg / yr), followed by dust (1440 Tg / yr), $SO_4$ (143 Tg / yr), OA (116 Tg / yr, of which ca 75 Tg / yr are due to primary emissions), $NO_3$ (33 Tg / yr), and BC (10 Tg / yr). Compared to AP1, the median emissions have decreased for all species except organic aerosols. For prescribed anthropogenic emissions, the differences between AP1 and AP3 may partly be due to differences in the emissions inventories. AP1 used inventories for the year 2000 whereas, here, the 2010 emissions are used (for details see supplement 1, section S6). Differences are likely also due to changes in the modelling setups and emission parameterisations.

Changes in parameterisations of online calculated natural DU and SS emissions are an explanation for their decreased emissions, 20% and 13%, respectively, compared to AP1. DU diversity has increased slightly relative to AP1, while SS diversity has decreased, however, with a standard deviation of circa 150%, it is still very large. As in AP1, the reasons for diversity in DU and SS emissions can be found in a range of parameters: surface winds, regions available to act as a source (semi-arid and arid areas for DU, sea-ice free ocean for SS), power functions used in the wind-emission relationship, aerosol size and other factors. As an example, different size cutoffs are applied in the models when computing the source strength (see Sect. 2.2). For instance, EMEP includes dust particles with sizes up to 10 $\mu$m, TM5 and EC-Earth consider sizes up to 16 $\mu$m, while ECMWF-IFS considers sizes up to 20 $\mu$m. While the higher size cut explains higher emissions for the IFS model, it does not explain why the TM5 dust emissions are lower than those in the EMEP model.

The emission strengths of dust and sea salt reflect the surface wind distribution, which exhibits a larger tail in the distribution at higher resolution and in free-running atmospheric models. Meteorological nudging that was required for AP3-CTRL leads to lower emissions (e.g., Timmreck and Schulz, 2004). Most of the models in the AP1 simulations had implemented free-running atmospheric models, but operated at lower resolution, which should cancel out to a certain degree and make AP1 and AP3 similar when it comes to effective surface wind distribution. Better documented wind distributions could help explain emission differences. For instance, SPRINTARS (one of the highest resolution models, see Tab. 2) exhibits a negative departure from the median in SS emissions, but an above average DU source (ca 1900 Tg / yr). The latter is comparable to that of OsloCTM3 and EMEP, which both use reanalysis winds at different resolutions. Also noteworthy are considerable differences in SS emissions between the two ECHAM models (ECHAM-SALSA emits ca 30% less SS but 18% more dust than ECHAM-HAM) even though these two models use the same emission parameterisation (see Sect. 4 in supplement 2), the same meteorology for nudging and have the same resolution (see Tab. 2). This indicates that nudging and higher resolution in AP3 are not the sole explanation for the AP3 decrease in the dust and sea salt emission strengths against AP1 and that inconsistencies remain.

Considerable diversity is also observed for OA emissions (64%), which is a result of multiple organic aerosol sources, represented differently by the models (supplement 1). Uncertainties are associated with the primary organic particle emissions (POA, diagnosed in only four models) and biogenic and anthropogenic secondary organic aerosol formation (SOA), DMS derived MSA, as well as biomass burning sources. As can be seen in supplement 1, there are also considerable differences among the models related to the conversion of organic carbon (OC) from the different sources to total organic mass. For instance, some models use a constant factor for all types of OC "emissions" (most commonly 1.4, though Tsigaridis et al., 2014, had suggested this value is too low) while others use different conversion factors for fossil fuel and biomass burning sources (ranging between 1.25-2.6). Conversion factors of 1.14 are reported for the NorESM model for monoterpene and isoprene as well as 8.0 for MSA (which is formed in the atmosphere via oxidation of DMS). Moreover, models show considerable differences in OA related emission inventories used. All these differences combined explain the high diversity associated with OA "emissions", which deserves further attention.

The decrease of $SO_4$ "emissions" compared to AP1 can not be explained by a change in anthropogenic $SO_2$ emissions between 2000 and 2010. Although Klimont et al. (2013) showed a decrease, the updated CEDS inventory (Hoesly et al., 2018) shows an increase of $SO_2$ emissions and was used in AP3. The increased variability in sulphate "emissions" may be due to considerable differences in the treatment of natural sulphur sources. The anthropogenic emissions are prescribed by CEDS and should be more consistent among the models, although loss of $SO_2$ and the chemical formation of $SO_4$ certainly contribute to "emission" variability. Estimates of volcanic sulfur emissions range between 1 – 50 Tg / yr ($SO_2$, e.g., Andres and Kasgnoc, 1998; Halmer et al., 2002; Textor et al., 2004; Dentener et al., 2006; Carn et al., 2017). Note that ECMWF-IFS did not consider volcanic emissions, and EMEP only considered major European sources (i.e., degassing from Etna and the Aeolian Islands, and the 2010 Eyjafjallajökull eruption in Iceland), which explains their comparatively low $SO_4$ emissions. GEOS, despite including volcanic emissions, also shows comparatively low $SO_4$ emissions (ca 95 Tg / yr). This could be due to a too inefficient conversion of $SO_2$ (and DMS) to $SO_4$ in GEOS. In terms of BC emissions, models agree well which is not surprising, since most models used the CMIP6 BC emission inventories (see supplement 1). Note that ECLIPSE BC, SOx, NOx and $NH_3$ emissions, used by EMEP, are somewhat lower compared to CMIP6. Emissions of $NO_3$ show a remarkable high diversity of 286% (see Fig. 2) with values ranging from 5.4 Tg / yr (TM5) up to 128 Tg / yr (GEOS), which is on the same order of magnitude as $SO_4$ and OA. Natural sources of $NO_x$ (soil, lightning) and formation of secondary $NO_3$ with ammonium, dust and sea salt provide several degrees of freedom for model formulation. $NO_3$ has only been implemented in some models in recent years and was not considered in the AP1 simulations.

The lifetimes (computed from burden and total deposition) are shown in the second panel in Table 3. Associated diversities are illustrated in Figure 2. OA has the longest lifetime with 6 days, followed by BC (5.5 d), $SO_4$ (4.9 d), $NO_3$ (3.9 d), DU (3.7 d) and SS (0.56 d). The largest differences compared to AP1 are found for BC which shows a decrease in lifetime of ca 15%, and in $SO_4$ and SS, showing increased lifetimes of ca 20% and 37%, respectively. In addition, the latter two species show a notable increase in lifetime diversity compared to AP1. In the case of sulphate, the increased variability is in agreement with the changes in emissions discussed above (i.e., it may reflect an increase in the natural fraction). This is consistent with the

increase in $SO_4$ lifetime compared to AP1, since DMS derived and volcanic emissions are often released into the free tropo-
sphere, where the residence time is larger. For sea salt, the increased lifetime relative to AP1 could indicate a shift towards
smaller particle sizes, but could also be due to differences in assumptions about water uptake. These changes in SS lifetime
and lifetime diversity will impact the conversion to optical properties, as shall be seen below. The decreased BC lifetime may
be due to changes in the treatment of BC in the models. For instance, in AP1 most models assumed external mixing (see Tab.
2 in Textor et al., 2006) while many models in AP3 treat BC as an internal mixture (e.g., with hygroscopic $SO_4$, see supple-
ment 1). This may impact the effective hygroscopicity of aged BC and thus, the wet scavenging efficiency. Earlier studies also
showed that BC in older models was likely transported too efficiently to the upper troposphere, with a too long lifetime as a
consequence (Samset et al., 2014). The dust lifetime is slightly decreased compared to AP1 and, with ca 56% the associated
inter-model diversity is slightly increased. The fact that the DU lifetime diversity is larger than the diversity in DU emissions
and burden indicates differences in the models regarding dust size assumptions. For instance, ECMWF-IFS shows the lowest
lifetimes both for dust and sea salt, which is subject of an ongoing development[3]. In the case of sea salt, the short lifetime for
ECMWF-IFS is related to the emission scheme used (based on Grythe et al. (2014)), resulting in too coarse SS particles. In the
case of dust, the scheme used by ECMWF-IFS is based on Nabat et al. (2012) and tends to produce too much dust. In addition,
it is possible that the DU emission size distribution (which is based on Kok, 2011) is coarser than in the other models (which
is also reflected by a below average DU MEC).

The modelled atmospheric mass burdens are shown in the third row of Table 3. They are essentially a result of their "emis-
sions", and lifetimes, discussed in the previous paragraphs. Consistently, the largest burdens are found for dust, followed by
SS, OA, and $SO_4$, while burdens for $NO_3$ and BC are small.

Compared to AP1, a notable decrease of ca 40% in the BC burden is found, which is in agreement with decreased emissions
and lifetimes discussed above. However, the associated variability of the simulated BC burdens (ca 50%) is comparatively large.
Since the BC emissions are relatively harmonised among the models, this variability is likely due to (ageing / mixing induced)
differences in the BC removal efficiencies, particularly in strong source regions such as China and India (e.g., Riemer et al.,
2009; Matsui et al., 2018). The sea salt burden is increased by ca 36% relative to AP1. This can be explained by the increased
lifetime compared to AP1, suggesting a shift towards smaller particle sizes for sea salt. The observed high diversities in sea
salt emissions (54%) and lifetime (92%) have a compensating effect on the variability in the associated burden, which is only
38% (see Fig. 2f). This indicates discrepancies in the assumptions about the associated size distributions of this predominately
coarse aerosol (see e.g., ECMWF-IFS vs. NorESM in Tab. 3). However, not all models show such a "compensation effect" of
emissions and lifetime for SS. For instance, both SPRINTARS and EMEP exhibit below median SS emissions and lifetimes,
resulting in the lowest SS burdens for these models. The lower SS emissions of EMEP are due to the fact that only SS particles
below $10\,\mu$m are simulated by the model.

---

[3]Personal communication with Z. Kipling and S. Rémy

The dust burden is decreased by ca 20% compared to AP1, which can be explained by the lower AP3 emissions and lifetimes. The associated diversity in dust burden is approximately the same as for the emissions and lifetimes (i.e., unlike for SS, for dust no "compensating" effect of emissions and lifetimes can be seen, see Fig. 2).

The sulphate burden is only slightly decreased relative to AP1, a result of the decrease in emissions, which is almost coun-
terbalanced by the increased lifetime. In terms of diversity, however, inter-model differences in $SO_4$ "emissions" and lifetimes have an enhancing effect on the associated $SO_4$ burden diversity (72%). Interestingly, models that have below average $SO_4$ emissions also tend to have below average $SO_4$ lifetimes and vice versa (in contrast to sea salt, where a compensating effect was observed).

The OA burden is slightly increased compared to AP1, however, the variability is comparable. Because the OA lifetime
decreases slightly between AP3 and AP1, changes in the burden are due to differences in emissions. This is difficult to tease out as organic aerosol treatment and the inclusion of different sources is very different than it was in AP1.

$NO_3$ shows the highest diversity in burden (>300%), with values ranging from 0.08 Tg (OsloCTM3) to 0.93 Tg (GEOS). This is likely associated with the wide range in the corresponding emissions, indicating disagreement in the formation of nitrate aerosol. According to the AeroCom phase III nitrate experiment, the majority of $NO_3$ formed in the atmosphere is associated
with atmospheric dust and sea salt in the coarse mode (Bian et al., 2017). Differences in the association of $NO_3$ with coarse particles and, thus, nitrate particle size can explain the large diversity in $NO_3$ lifetime (ranging from 2.5 to 10.4 days). For instance, TM5 and EC-Earth show the longest $NO_3$ lifetimes, which is likely due to the fact that nitrate is described only by its total mass and assumed to be present only in the soluble accumulation mode (van Noije et al., 2014). The comparatively small nitrate burden of OsloCTM3 (0.08 Tg) is because the reported $NO_3$ diagnostics only include fine nitrate and coarse $NO_3$
particles are included in the sea salt diagnostics, however, with almost no impact on the burden of sea salt. A careful budget analysis for nitrate would need more information on its chemical formation and particle size distribution and most importantly, more consistency among the models in the reported nitrate diagnostics.

### 3.1.2  Diversity in optical properties: speciated MECs, MACs and ODs

Global annual average MEC values for each species are provided in the 5th part of Table 3. They represent here the link between
*dry* aerosol mass and its size distribution and the resulting *ambient* air total light extinction (i.e., absorption + scattering of the water containing particles) associated with a given species. Since the MEC values here were computed via $OD_{i,amb}/Burden_{i,dry}$ ($i$ denoting an aerosol species) they represent the whole atmospheric column and they include the effects of water uptake (while the species specific burden values represent just the dry aerosol component). Because the MEC (and MAC) values reported here will include the water contribution to the species OD they will be larger for hygroscopic aerosol such as sea salt or sulfate
than the corresponding values for dry conditions (e.g., Table 5 in Hand and Malm, 2007). This is partly balanced by smaller specific extinction for larger particles. Notably, the model-derived MECs for dust shown in Table 3, are fairly consistent with measurement-based dry mass scattering efficiencies for dust (Hand and Malm, 2007). This consistency is reassuring because dust is typically considered to be hydrophobic in models, meaning there shouldn't be a large discrepancy between MEC for dry and ambient conditions. Note that the split of total AOD into speciated ODs is not trivial for internally mixed aerosols. The

general recommendation for such diagnostics is to split proportional to the dry volume fractions of the species. The latter may result in too much water uptake associated with hydrophobic particle fractions. This can have implications for the computed BC MECs as discussed below.

The species specific MECs found in this AP3 analysis are mostly similar to those reported for AP1. The largest difference is in the DU MEC (ca 20% decrease), suggesting that the AP3 models tend to simulate larger dust particles compared to AP1.

This is consistent with the observed slight reduction in DU lifetime for AP3. AP1 models likely simulated too fine (or a too large fine fraction) dust particles as suggested by comparisons with AE by Huneeus et al. (2011). As shall be seen in Sect. 4, the AP3 models considered here still tend to overestimate AE in dust dominated regions. This combination of results suggests that the simulation of dust aerosol size has been improved since AP1, however, dust particles are likely still too fine.

MECs of sea salt and sulphate are comparable with AP1, however, both show a decreased inter-model variability. While 640 one could conclude that this may suggest better agreement in the modelled size distributions of SS and $SO_4$, the dramatic increase in the variability of their lifetimes suggests differently. The better agreement in MEC may also be linked with similar assumptions in microphysical properties in AP3 (e.g., refractive index or density, see supplement 1) or assumptions about (and impacts of) hygroscopic growth of these hydrophilic species (e.g., for SS most of the light extinction is linked to high water uptake). However, from the broad diagnostic overview provided here, it is difficult to understand what drives this behaviour.

$NO_3$ shows the highest MEC variability of all species, though, again, only 9 models consider this species. However, compared to the spread in its burden and emissions, the $NO_3$ MEC diversity is "small" (see Fig. 2) and is similar to the corresponding lifetime diversity (<100%). TM5 and EC-Earth exhibit the largest $NO_3$ MECs because in these models the particles are associated with the optically more efficient accumulation mode. Other models (such as GEOS) appear to have their $NO_3$ more tied with DU and SS and exhibit smaller $NO_3$ MECs.

The BC MEC values exhibit a diversity of ca 20%, smaller than in AP1, however several models were excluded from the ensemble calculations. The MAC values (shown in the 6th part of Table 3) represent the absorptive fraction of MEC (computed as $AAOD_{i,amb}/Burden_{i,dry}$) and should fulfill by definition: $MAC_i < MEC_i$. However, for some models (ECHAM-HAM, ECHAM-SALSA, EMEP and INCA), BC MAC values are up to 3 times larger than their BC MEC values. TM5 and EC-Earth did not submit $AAOD_{BC}$ but $AAOD_{tot}$ and BC MAC was estimated for those two models via $MAC_{BC}* = AAOD_{tot} / Burden_{BC}$

(assuming BC was the dominant absorber). This resulted in estimated MAC values exceeding 12 $m^2$/g, even in regions where the weak absorbers DU and OA do not add absorption. BC MECs for TM5 and EC-Earth are around 6 $m^2$/g (comparable to the ECHAM MEC values), meaning the BC MACs for these two models also exceed their MECs. In the case of EMEP, the $MAC_i > MEC_i$ discrepancy results from an inconsistency between the choice of the prescribed MAC literature value and the ways MECs are computed in the model (for details, see Sect. 4 in supplement 2). In the case of INCA, the apparent

inconsistency between BC MEC and MAC is related to BC absorption enhancement for external mixing which is based on Wang et al. (2016) and improves the agreement with BC MAC observations. However, the BC absorption enhancement effects are only considered for the computation of $AAOD_{BC}$, not $OD_{BC}$. For the other affected models (EC-Earth, TM5, ECHAM-HAM, ECHAM-SALSA) discrepancies between BC MAC and MEC likely arise from nonlinear internal mixing rules that

may not be properly accounted for when computing the component optical depths based on the species volume fraction, as recommended by the AeroCom protocol.[4]

These inconsistencies with calculating aerosol optical properties will primarily impact the BC OD and MEC estimates, because (a) aerosol absorption contributes a large fraction of the column extinction making up BC OD and (b) the OD due to absorbing aerosol is small (around 0.002) relative to the other (mostly scattering) species. The inconsistent BC MAC, MEC and OD values for the affected models are indicated in brackets in Table 2 and were excluded from the computation of the corresponding AP3 ensemble median and diversity estimates (and accordingly, also from Fig. 2). These inconsistencies may have already been an issue for some models in Kinne et al. (2006) where similar recommendations related to the computation of component ODs were given by AeroCom.

As can be seen in Table 3, BC is by far the most efficient absorbing species, suggesting a BC MAC value of around $8.5 \text{m}^2/\text{g}$ – almost 2 orders of magnitude more efficient than dust or OA at 550 nm. This value is slightly larger than MAC values suggested for fresh BC based on extensive surveys of fresh BC MAC values reported in the literature, for instance, $7.5\pm1.2 \text{ m}^2/\text{g}$ (at 550 nm) recommended by Bond and Bergstrom (2006) or $8.0\pm0.7 \text{ m}^2/\text{g}$ from Liu et al. (2020). Bond and Bergstrom (2006) note that for aged BC, the MAC may be enhanced by 35 – 80%, with the enhancement due to coatings as well as changes in morphology. They suggest BC MAC values in the range 9 – 12 $\text{m}^2/\text{g}$ for ambient BC. Measurements across Europe indicate MAC values ranging between 4.3 – 22.7 $\text{m}^2/\text{g}$ (at 637 nm) as summarised by Zanatta et al. (2016). They propose a value of 10 $\text{m}^2/\text{g}$ (at 637 nm) to be representative for a mixed boundary layer at European background sites, which would translate to 11.6 $\text{m}^2/\text{g}$ at 550 nm assuming AAE=1.

The range in species dependent MACs in Table 3 is ca 50% after excluding the models mentioned above. Given the harmonised BC emissions used, this diversity indicates differences in the BC treatment in the models. The BC MAC values derived from the AP3 models reflect uncertainties related to assumptions about optical properties of aged BC (e.g., absorption enhancements due to coatings discussed, for example, by Bond and Bergstrom, 2006; Schwarz et al., 2008; Liu et al., 2017) but could also indicate uncertainties related to the size and density (which varies between 1 – 2.3 $\text{g/m}^3$ among the AP3 models) and / or assumptions about the refractive index $m = n + ik$ of BC (see supplement 1). We suspect though, that the largest differences are due to assumptions about BC aging (e.g., coatings) and processes affecting its size distribution since models using nearly the same $m$ exhibit large differences in MAC. For instance, GEOS, OsloCTM3 and SPRINTARS all use $m \approx 1.75 \pm 0.45i$ at 550 nm but show MACs of 7.8, 13.0 and 3.1 $\text{m}^2/\text{g}$, respectively. The variations in MAC may also impact BC lifetime - coatings may result in lifetime variations by changing the hygroscopicity of BC - however, we find no clear relationship between BC lifetime and MAC.

---

[4]For the two ECHAM models, $OD_{BC}$ and $AAOD_{BC}$ are diagnosed as follows: $OD_{BC}$ is computed from the BC volume fraction ($dV_{BC}$), relative to other abundant species (i.e., $OD_{BC} = OD_{tot}*dV_{BC}$), while for the computation of $AAOD_{BC}$, $dV_{BC}$ is weighted by the respective imaginary refr. indices of all species in the mixture. For instance, if BC is the only absorber, then $AAOD_{tot}=AAOD_{BC}$. $AAOD_{tot}$, however, is computed in the model via $OD_{tot}*(1-SSA)$, where SSA is the single-scattering-albedo of the mixture. Then, if $1-dV_{BC} > SSA$ it follows that $AAOD_{BC} > OD_{BC}$ (i.e., in cases where the SSA of the mixture is smaller than the scattering volume fraction). A correction suggested by H. Kokkola (through pers. communication) would be to compute the scattering component as $SCOD_{tot}=OD_{tot}*SSA$ and then computing $SCOD_{BC}$ accordingly by weighting its volume fraction with the real part of the refr. indices (of all species in the mixture), then $OD_{BC}=SCOD_{BC}+AAOD_{BC}$.

A comparison of MECs and MACs for DU and OA suggests that ca 5% and ca 2% of their total extinction is due to absorption at 550 nm, respectively. DU MAC and MEC diversities are similar. However, we find a slightly larger increase in DU MAC diversity compared to DU MEC, when considering a consistent model ensemble. This is likely linked to a larger disagreement in the DU imaginary refractive index ($ik$) compared to its real part (see supplement 1). It reflects uncertainties related to assumptions about dust absorptive properties, which depend on mineralogy and size, for instance, relative hematite and soot content (Kandler et al., 2007). MAC diversity for OA also exhibits an apparent increase compared to MEC and shows considerable variability in $ik$, based on a small model ensemble.

The following paragraphs focus on the discussion of speciated optical depths, that is, (1) how they result from the above discussed parameters, (2) how they contribute to total AOD and (3) how they compare with the AP1 data.

SS makes the largest contribution to total AOD with a median value of 0.044, followed by $SO_4$ (0.035), OA (0.022), DU (0.021) and to a lesser degree, $NO_3$ (0.005) and BC (0.002) (see Tab. 3). The largest diversity is found for optical depth of OA, followed by $SO_4$, SS, $NO_3$, DU and BC. Figure 2 illustrates how OD diversity is linked to diversities in emissions, lifetime and MEC. The diversity in SS ODs is almost twice as large as SS diversities in burden and MEC. This reflects that models with a high SS mass burden also tend to have an above average MEC (and vice versa), possibly, simulating smaller and thus, optically more efficient SS particle sizes (e.g., NorESM2). However, the wide spread in SS parameters is likely also linked to varying contributions of water due to different parameterisations of SS hygroscopicity (see discussion in Burgos et al., 2020). The role of water adds another level of complexity to the relationship between aerosol lifecycle and optical properties for hygroscopic species such as SS.

Similar to sea salt, an "amplifying" combination of burden and MEC diversities is found for OA. As can be seen in Figure 2d, this results in a prominent departure from the median for all modelled OA ODs, with none of the models being close to the ensemble median. Sulphate ODs exhibit a similar spread as the sulphate burdens, while the range in MEC is smaller. This suggests that models tend to agree better in their simulated $SO_4$ sizes (also supported by comparatively low diversity in lifetime) and optical properties. It further suggests that the disagreement in OD is primarily related to uncertainties in the $SO_4$ "emissions".

The diversity in BC and dust OD is smaller than the associated burden and MEC diversities for these two species (see Fig. 2a), indicating a compensating effect on OD variability. We can partly explain this. For instance, in NorESM2 a low DU burden (a result of low emissions and lifetimes) is compensated by a large MEC, resulting in an close-to average dust OD. This appears to be contradictory, as one would expect that a shorter DU lifetime would reflect larger particles and thus a small MEC. However, NorESM2 assumes some hygroscopicity for dust (i.e., DU hygroscopic growth factor $\kappa = 0.069$, see supplement 1), which may lead to an efficient wet-removal pathway also for small DU particles. In addition, MEC values are not only size dependent. However, variations in the dust refractive index shows good agreement in its real part among the models (see supplement 1).

The speciated ODs are also shown in Figure 3 as a stacked bar-chart. The plot also includes median values from AP1 from Kinne et al. (2006) as well as global estimates of the total AOD from AERONET and the merged satellite product. The latter were scaled to represent global averages, using a scaling factor that was computed from ENS-MED, by dividing the global

average of ENS-MED with its average when co-located with AERONET and the merged satellite, respectively. On average, the global total AOD from models has not substantially changed from AP1 to AP3, although it is lower than the AOD from both observational data sets (details will be discussed below in Sect. 4). Most notable compared to AP1 is a shift in the natural contribution from SS and DU, respectively. AP3 models show a shift towards more sea salt, with SS making up 2/3 of the natural OD. Interestingly, this shift is likely not due to the changes in the emitted mass of these species (since SS emissions have decreased more than DU since AP1), but likely originate in changes in the simulated size distributions, with changes in DU and SS lifetimes and MECs as discussed above. Figure 3 also shows that the BC OD is decreased by a factor of 2 compared to AP1. This marks a substantial change in this important species and can be explained by the combination of decreased BC emissions and lifetime (and thus, burden) and MECs. This manifests in substantial model underestimates of surface absorption coefficients, as shall be seen below in Section 4.

Based on 8 models, water makes up between approximately 40 – 65% of the total AOD. The water OD is correlated with SS OD (R=0.63; and R=0.80 when SPRINTARS is excluded). SPRINTARS appears to be an outlier, its SS OD is only 0.013, related to issues described above. Also the fraction of water (water OD / total OD) to fraction of sea salt (SS OD / total OD) exhibits a strong correlation of R=0.81 (SPRINTARS excluded). Textor et al. (2006) notes that water in AP1 makes up about 1/2 of the aerosol mass and also observed a relationship between water and SS mass.

Also indicated on Figure 3 is whether models provided clear-sky (CS) or all-sky (AS) AOD. Five models report only AS AOD, six models report only CS AOD and three models report both. The ensemble median AOD values (CS AOD=0.132 and AS AOD=0.128) are slightly higher than the median AOD value of approximately 0.125 reported in Kinne et al. (2006) (see Fig. 1 therein). AOD diagnostics used by Kinne et al. (2006) were based on undocumented clear-sky/all-sky assumptions so it is difficult to explain the slight increase in AOD. The comparison with remote sensing observations discussed in the following Section 4 is performed using CS AOD if available. For a given model, CS AOD is smaller than AS AOD (where both are available). This is expected, since CS AOD is not as much affected by hygroscopic growth, while AS AOD reflects conditions which include supersaturated environments needed for cloud formation. This is also reflected in the larger diversity in AS AODs compared to CS AODs (see Tab. 3) as models utilize different RH ceilings for AS conditions. For the three models that report both AS and CS AOD, the largest difference is found for SPRINTARS where the CS AOD is almost 50% smaller compared to the AS AOD, while the results from NorESM and ECHAM-HAM suggest circa 10% lower AOD under clear sky conditions. This could indicate that SPRINTARS exhibits higher global cloud coverage or increased impacts of hygroscopic growth. Its water AOD, however, is close to the median.

EMEP and ECMWF-IFS reported AS AOD; surprisingly they are among the models with the lowest AOD. In the case of ECMWF-IFS this seems to be mostly due to too small OA and $SO_4$ ODs. $SO_4$ is likely too low due to missing volcanic degassing (see supplement 1). Low OA OD could be related to size and / or assumptions about hygroscopicity, as both MECs and lifetimes are comparatively low. In the case of EMEP, on the other hand, the low AOD is a combination of too little SS, $SO_4$ and DU which is, to a certain degree, compensated for by its large OA OD. The latter is a result of comparatively strong "emissions" (POA+SOA). EMEP also simulates the largest $NO_3$ OD (due to contribution from both fine and coarse nitrate).

Finally, it is interesting to note that, despite the spread in OD values for different species, the simulated total AOD (both CS and AS, where provided) indicates much better agreement among the models (within 8% for CS AOD and 24% for AS) – a characteristic that has not changed since Kinne et al. (2006).

## 3.2 Modelled annual global distributions of optical properties and their diversity

Figure 4 shows global maps from the ensemble median (Sect. 2.3.1) for each variable (left) and corresponding diversities (right). The diversity maps provide insights into the regional model-spread. They are useful to identify regions where models tend to disagree. This may also help to explain differences between models and observations. The AOD and AE diversity maps from this study can be compared with the diversity maps for these properties presented in Kinne et al. (2006) (their Figure 4). Kinne et al. (2006) report the largest diversity in AOD at high latitudes and over central Asia which is similar to what we see here. The AE diversity map in Kinne et al. (2006) is also similar - the highest AE diversity is observed over northern Africa and the southern ocean. They note that these are regions dominated by dust and sea salt respectively, suggesting that assumptions about aerosol size for these species are important to properly simulate AE. The high diversity in the ambient column AE in the South Pacific, is likely due to both differences in assumptions about the initial SS size distributions and in the simulated water uptake of SS.

Figure 4 also presents maps for $AOD_f$ and $AOD_c$. On a global scale $AOD_f$ dominates the total AOD and the median values indicate a fine-mode fraction of ca 60% (note that the total AOD field includes one more model than $AOD_f$ and $AOD_c$, thus here $AOD \neq AOD_f + AOD_c$, see Sect. 2.3.1). $AOD_f$ is highest over regions with strong anthropogenic sources (e.g., China, India, eastern US, etc.) but also in regions associated with biogenic emissions and / or biomass burning (e.g., Amazonia, central Africa). The diversity in $AOD_f$ is highest in remote regions (high latitudes and high altitudes such as the Andes and the Himalayas). This could be due to differences in aging and removal processes affecting long range transport, or differences in local sources (e.g., few models include oceanic POA emissions). However, it could also be affected by "inter-model noise", for instance, because these regions typically show smaller burdens, or because of differences in the methods that models use to diagnose $AOD_f$ and $AOD_c$ (e.g., some models use the dry radius for the split, others the ambient radius , see supplement 1, and discussion below in Sect. 4). Another region with low $AOD_f$ and high diversity is the Bay of Bengal and the Indian Ocean, likely linked to differences in the modelling of the mass outflow from the heavily polluted Indo-Gangetic Plain, in combination with complex meteorology prevalent in this region (e.g., Pan et al., 2015).

Outside of high latitudes, $AOD_c$ is most diverse over China, southern Africa, the eastern US and western South America. $AOD_c$ exhibits the opposite pattern as AE, which is expected as AE is inversely related to size. However, the patterns in diversity for AE and $AOD_c$ are different. Outside of high latitudes, the AE diversity is highest in regions dominated by natural aerosol, while the highest $AOD_c$ diversity occurs primarily over anthropogenic source regions. Both AE and $AOD_c$ show a high diversity over the Sahara, which is consistent with the considerable variability related to the DU lifecycle and optical properties, discussed in the previous section. For remote oceanic regions, the coarse AOD shows much less diversity than AE. This indicates large diversity related to SS size assumptions and modelling. For instance, an increased variability in $AOD_f$ is found in the South Pacific, where AE also shows high inter-model variability but $AOD_c$ diversity is rather small. This again

highlights the need for the models to reevaluate fine and coarse SS, due to the increased extinction efficiency at finer particle sizes, even though most of the SS mass resides in the coarse mode. China is also one of the regions showing considerable diversity in $AOD_c$. This could be related to dust storms, which regularly affect China (e.g., Sun et al., 2001). The fact that models tend to agree in AE in this region indicates similar assumptions in the modelled dust size. However, this needs further investigation due to the complex relationship between AE and size (Schuster et al., 2006), particularly for regions such as China, which are affected both by fine (anthropogenic) and coarse (natural) aerosol (also indicated by the AE$\approx 1.25$ prevalent over China).

The overall highest diversity is found for the simulated surface in situ aerosol absorption coefficients, in particular in Amazonia, a region of substantial regular biomass burning events (e.g., Rissler et al., 2006), which were peaking in early September in 2010 (see Mortier et al., 2020c). Reasons for these differences may be a combination of the different treatments of SOA formation (and absorptive properties of OA), or potential differences in the emission altitudes of smoke plumes (see supplement 1 for details). The diversity in simulated in situ surface absorption is also high in Australia, another region affected by regular biomass burning events as well as dust emissions. Interestingly, models tend to agree in major source regions such as China and India, simulating low diversity in aerosol absorption at the surface.

The dust dominated Sahara region also shows considerable diversity in simulated surface absorption but little diversity in simulated surface scattering. We explain this by noting the considerable differences in the assumed imaginary indices for dust at 550 nm (see supplement 1). The high diversity in AE in this region also suggests differences in the simulated dust size distributions and is linked with the increased diversity seen in $AOD_c$. These results reflect the diversity among models for dust emissions, burdens, and lifetimes, as well as MECs and MACs (Tab. 3) as suggested already by Huneeus et al. (2011).

Several clean regions exhibit high diversity in one or more variables. The South Pacific and Antarctica exhibit a zone of high diversity in simulated surface absorption (but not scattering) in addition to the diversity in AE, $AOD_f$ discussed above. In addition, models tend to show high variability in $AOD_c$ over land in this remote area. This behaviour points to considerable differences in long range transport of the aerosol (e.g., dust exhibits $> 50\%$ diversity in lifetime, see also Li et al., 2008), or treatment of organic emissions from the ocean. Elevated and/or mountainous desert regions such as the Southern Peruvian and Northern Chilean Andes and Tibet also show high diversity in $AOD_c$. Unfortunately, most ground-based observations provide little or no coverage in these remote regions, where the models exhibit high diversity. These model results therefore lend support to the idea of expanding measurements in undersampled locations, in order to better evaluate models.

While presenting model variability in terms of percentages sheds some light into differences into the modelling of the aerosol in remote and clean regions (with low aerosol loading), it gives equal weight to variability independent of the abundance of aerosol. However, the diversity in pristine areas is small in terms of absolute value and is thus unlikely to have a significant impact on the global radiation budget.

## 4 Results and discussion - Optical properties evaluation

This section presents and discusses the results from the optical properties evaluation, shown in Table 4 and Figures 5 & 6. Most of the discussion is based on the results from the ensemble model (ENS-MED). A detailed assessment of individual models on a regional or seasonal scale is beyond the scope of this paper. However, where there are clear outliers among the individual models we note model assumptions and attempt interpretation. Note that detailed results for each model and observation data set are provided online via two web interfaces: a) the new interactive visualization (see Mortier et al., 2020c), including regional statistics and b) the old AeroCom web interface (see Schulz et al., 2020), obtained with a different analysis tool AeroCom (IDL based), which also allows interactive viewing of earlier AeroCom results.

Figure 5 shows global maps of annual mean biases retrieved when co-locating the ensemble median model (ENS-MED) against some of the various observational parameters and datasets (see Sect. 2.3 for methods). Note that not all satellite datasets used later are shown in this figure.

The individual panes in the bias maps indicate that, in general, models simulate lower values for all parameters considered, even though there are some regions / locations where models overestimate the observations. It should be noted that the differences in NMB for $AOD_c$ and AE between AERONET and AATSR-SU for the models primarily reflect the respective biases found in the satellite assessment (i.e., ca -15% for $AOD_c$ and +15% for AE, see Sect. 2.1.5). For instance, the AE results from AATSR-SU suggest that the ENS-MED model underestimates AE by 22% (Tab. 4). Figure 5 suggests that most of this bias is due to underestimations of AE over the oceans. This would suggest that the AP3 models tend to overestimate the ambient particle size in marine environments. However, the comparison with AERONET suggests that AE in AATSR-SU is overestimated by ca 14%, which may be linked with the different wavelength regimes used in AATSR and AERONET to retrieve AE (see Sects. 2.1.1 and 2.1.5, see also Schuster et al., 2006). The extent to which this bias translates to the oceans would need to be investigated to make a clear statement about whether current models are capable of simulating AE in marine environments. A detailed investigation of this is beyond the scope of this paper but desirable, given the important role of sea salt also for cloud formation, lifetime and optical properties in clean and remote marine environments, and the impact on assessing the indirect aerosol effect (e.g., Fossum et al., 2020, and references therein).

Table 4 shows performance matrices of the normalised mean bias (NMB, top) and the Pearson correlation coefficient (R, bottom). These are displayed for each model, variable and observation data set used, respectively. They represent averages over all site locations, for each observation platform. The evaluation results from the AeroCom ensemble median and first and third quantile fields (ENS-MED, Q1, Q3, Sect. 2.3.1) are displayed in the rightmost columns. Note that for the AOD comparison with the MERGED-FMI product, 3 different results for NMB and R are provided, 1. representing the whole globe (denoted MERGED-FMI), 2. only over land (MERGED-FMI-LAND) and 3. only over ocean (MERGED-FMI-OCN). The land / ocean filtering was done using gridded masks provided by the Task Force on Hemispheric Transport of Air Pollution (TF HTAP), which are constrained to a latitude range from 60°S to 60°N.

While most models moderately underestimate the selected optical properties compared to the observations, they show surprisingly good agreement in terms of correlation (e.g., ENS-MED R is between 0.72 and 0.88, see lower pane in Tab. 4). This

suggests that spatial and temporal variations are mostly captured by the models, despite underestimating the absolute magnitude of the investigated optical parameters. However, for some variables individual models perform quite poorly (e.g., $AOD_c$ by ECHAM-HAM and NorESM2). Compared to the individual satellite AOD data sets, the models show high correlation ($\geq 0.78$ for ENS-MED) and differences in NMB mostly reflect biases of the satellites established against AERONET (see Tab. 1, i.e., the largest AOD underestimate of ca. -35% is found against the MODIS satellites).

TM5 and EC-Earth appear to be the best performing individual models when all optical variable comparisons are considered - they exhibit mostly low biases ($<\pm10\%$) and correlations close to the ensemble model for most parameters. The similarity in their results is not surprising, given the similarity of their model setups (see Sect. 4 in supplement 2). SPRINTARS is the model that most consistently underestimates observations, which is not surprising as it was the model that consistently had the lowest burdens and component ODs in the inter-model comparison, particularly for DU and SS (see Tab. 3). ECHAM-HAM, ECHAM-SALSA and NorESM2 are the models exhibiting the least correlation with the observational data sets, particularly for $AOD_c$ and $SC_{dry}$, and over the oceans (Tab. 4). This could be due to their assumptions related to SS, discussed above. Indeed, there is a tendency for lower correlation in $SC_{dry}$ at coastal GAW sites (see Mortier et al., 2020c). The comparatively long SS lifetimes for these three models could result in larger particles (e.g., due to more swelling and less wet deposition) which will impact the ambient SS MECs and ODs (which are largest for these models, see Fig. 3). Also, these three models exhibit the highest $H_2O$ AODs (among the models who submitted this diagnostic), which likely results from enhanced impacts of SS hygroscopic growth. Additionally, the importance of SS parameterisation is demonstrated by comparisons with Tegen et al. (2019) who find higher correlations and a good agreement with observations of size distributions for ECHAM-HAM, using a different SS emission parameterisation scheme than is used by ECHAM-HAM in this study. Tegen et al. (2019) also show a positive bias in AE in ECHAM-HAM compared to AERONET, suggesting the model simulates more fine particles than are observed, while the version of ECHAM-HAM used here shows a negative bias in AE compared to AERONET (and AATSR-SU), suggesting too many coarse particles are being simulated. These results suggest that the versions of ECHAM-HAM, ECHAM-SALSA and NorESM2 used here overestimate the sea salt size, either due to the parameterisation (e.g., online emission and dry size distribution) or hygroscopicity or some combination of both. This could also explain the lower correlation in these models, which is particularly apparent over the oceans as can be seen from the different AOD results over land and ocean in Fig. 4 (i.e., from the MERGED-FMI product): RH over the oceans is, on average, likely more smoothly distributed in space and time than the actual SS emissions (which strongly depend on near surface wind speeds) which could have a smoothing effect on the spatio-temporal variability of the SS AOD signal, manifesting in a lower correlation compared to observations (which have less swelling).

Figure 6 presents another way of looking at the NMB. As with the top pane of Table 4, it is clear that the models have a tendency to underestimate the observations, with largest underestimates (>25%) for $AOD_c$ (against AERONET) and $SC_{dry}$ (at GAW sites).

## 4.1 AOD, $AOD_f$ and $AOD_c$

This section presents and discusses the results for total AOD as well as $AOD_f$ and $AOD_c$. The latter two diagnostics can
provide insights into the differences associated with the modelling of anthropogenic aerosol (more fine dominated) and natural
aerosol (dominated by dust and sea salt).

As mentioned above, models typically underestimate observed total AOD regardless of measurement platform (i.e, by ca.
20% against AERONET and 16 – 37% against the various satellite products, see Tab. 4). However, the model AOD biases
established against the four satellites mostly resemble the biases of each satellite found when compared with AERONET
(see Tab. 2). Thus, we concentrate the discussion of satellite AOD results on the MERGED-FMI data set, which includes
both MODIS and AATSR and shows good performance at AERONET sites (NMB=-5.5%, R=0.89). This is the reason why
individual results from AATSR-SU and MODIS (Terra and Aqua) are not shown in Figure 6.

The consistent AOD underestimate in the models means that they are either simulating too little mass loading in the column
and / or underestimating the column optical extinction efficiency (e.g., related to assumptions about size, MEC and / or compo-
sition). Figure 6 suggests that most of the AOD bias is due to missing (or optically too inefficient) coarse aerosol, which also
exhibits the largest diversity (in terms of bias) among the models and shows lower correlation than AOD and $AOD_f$ (Tab. 4).
However, as pointed out above, about half of the models computed $AOD_f$ using the dry particle radius (see also supplement
1). These models likely attribute some extinction to the fine mode that should be attributed to $AOD_c$ if hygroscopic swelling
was accounted for.

The impact of this uncertainty was investigated using 7 models that submitted diagnostics of fine and total SS OD. Four
of these, CAM5-ATRAS, GEOS, GISS-OMA and SPRINTARS computed $AOD_f$ based on dry radius, while the other three,
EC-Earth, TM5 and ECMWF-IFS used the ambient particle radius. The first 4 models suggest a SS fine-mode fraction of circa
26% while the latter 3 models suggest 15%. Thus, circa 11% of the SS OD is erroneously attributed to $AOD_f$ when the dry
radius is used to split fine and coarse mode. On a global scale, ca 33% of the total AOD is due to SS (Tab. 3). Hence, circa 3%
too much of the total AOD is attributed to $AOD_f$ if the dry radius is used, assuming that SS is the dominant species affected by
this error. This is a fair assumption as other hydrophilic species typically reside in the accumulation mode, and dust is assumed
to be hydrophobic in most models.

Thus, for the affected models, $AOD_c$ is likely slightly shifted towards less negative biases by ca 3%, while $AOD_f$ would
show larger underestimations, accordingly. The diversity in $AOD_c$ could be in parts due to the different methodologies used in
the models to determine the size threshold, however, this is not the sole explanation, since the $AOD_f$ exhibits less diversity.
Also, the diversity in $AOD_c$ is consistent with the large diversity found for dust and sea salt aerosol among the models (see
Sect. 3) and highlights the large uncertainties associated with the modelling of these natural aerosols (see Sect. 3). Attempts
to address these uncertainties are also reflected in the substantial changes in AP3 compared to AP1 (e.g., less and optically
more inefficient dust and more sea salt in AP3). Figure 5b does not allow a clear statement related to over or underestimates
of dust and sea salt. For instance, in several remote ocean regions ENS-MED exhibits positive biases (e.g., South Pacific),
whereas in other regions slightly negative biases or good agreement are found. This suggests that, overall, models manage to

capture the overall magnitude of the sea salt contribution. Most of the coarse AOD bias seems to be associated with continental land-masses (e.g., SE-Asia, Arabian Peninsula, Fig. 5d), which is also indicated by the lower underestimate when comparing $AOD_c$ against AATSR, instead of AERONET (Tab. 4).

We stress that the fact that models tend to match the magnitude of the AOD in sea salt dominated regions does not necessarily reflect that the processes leading to these AODs are represented correctly. In this context we refer once more to the large diversity (and compensating effects) associated with SS emissions (computed online), burdens, lifetime and MECs and the substantial changes since AP1 (see Sect. 3 for details). This is further supported by the typically decreased correlations found when comparing models with satellite datasets (which "see" both oceans and continental land masses, see, for example, results

from MERGED-FMI, LAND versus OCN in Tab. 4). For instance, ENS-MED exhibits comparatively low correlation in the SW and S-Pacific compared to the satellites (see online results, Mortier et al., 2020c). However, these regions are affected by high cloud coverage throughout the year, and thus the lower correlation may also be due to representation errors in the monthly satellite aggregates used.

    ECHAM-SALSA and INCA are the only models which slightly overestimate AOD, when compared with two satellite

products (MERGED-FMI and AATSR). The overestimates by ECHAM-SALSA and INCA have some of the largest contributions of SS and sulphate to AOD (Fig. 3). Both models exhibit overestimates of AOD over the ocean as can be seen in comparison with MERGED-FMI-OCN (see Tab. 4) suggesting an overestimated contribution of sea salt to AOD. In the case of ECHAM-SALSA, this is further confirmed by the overestimated $AOD_c$ (by ca 24%) compared to the AATSR data set (in addition, ECHAM-SALSA is one of the models using dry radius to compute $AOD_f$, thus likely overestimating the fine-mode

fraction diagnosed through $AOD_c$ and $AOD_f$). However, note again that $AOD_c$ from AATSR is underestimated by ca 15% at AERONET sites (Tab. 1). Thus, it is difficult to draw clear conclusions about the magnitude of the model bias over marine environments.

    AOD underestimates are mostly associated with SE-Asia and Amazonia, Siberia and high latitudes in general (see Fig. 5a,b). The latter may be associated with insufficient transport towards the poles (e.g., Stohl, 2006) or associated phenomena

affecting the radiative properties of the Arctic atmosphere (such as arctic haze, e.g., Tunved et al., 2013) - perhaps linked with an insufficient attribution of the extensive wildfires in Russia in the summer of 2010 (e.g., Mielonen et al., 2012). The negative bias in S-America (i.e., AERONET AOD~0.5 while ENS-MED~0.3, see web results, Mortier et al., 2020c) mostly arises from an underestimate in the biomass burning season in 2010. However, compared to the AP1 simulations (which simulated a too early biomass burning season in S-America, Kinne et al., 2006) the AP3 models match the timing of the biomass burning

season well (which peaked in Aug.-Sept. in 2010).

    Explaining the AOD underestimate in SE-Asia is not trivial. SE-Asia has substantial anthropogenic emissions which are fairly harmonised among the AP3 models (cf. Sect. 3). This can also be seen by the comparatively low diversity in surface $SC_{dry}$ and $AOD_f$ in Figure 3.2. The region is also strongly affected by a pronounced seasonality including an intensive biomass burning season, dust inflow (e.g., transported from the Arabian Peninsula), monsoon seasons (i.e., seasonality in wet

deposition), and other factors impacting the abundance and properties of aerosol (e.g., fog and hygroscopicity), making it difficult for models to simulate regional and seasonal aerosol loadings (e.g., Pan et al., 2015). However, as can be seen in

Figure 5d most of the AOD bias in this region is due to missing (or optically too inefficient) coarse particles, which is also supported by the overestimated AE in that region (e.g., Fig. 5e). Thus, the bias could be related to insufficient dust transport or too coarse (and optically less efficient) dust particles. This hypothesis is consistent with the generally low burden, MEC and OD associated with dust in AP3 discussed above (Sect. 3). A detailed investigation of the AP3 models in this important region is desirable but beyond the scope of this paper, due to the complexity of the prevailing processes.

It is also interesting to compare the relative differences in model simulations of AOD for the different observation platforms. NorESM2, for instance, is the only model that seems to exhibit a weaker performance (in terms of bias) simulating AERONET AOD than simulating any of the satellite AODs. The large underestimate over land could be linked with its comparatively low $SO_4$ and dust optical depths (see Tab. 3), resulting in substantial AOD underestimates compared to AERONET. The seemingly better performance when compared to the satellite products is due to an overestimate of the SS contribution (particularly in the southern oceans), which compensates to some degree for the underestimate over land (see also Fig. 3). The latter is likely due to the SS size assumptions discussed above, resulting in a shift towards finer, and optically too efficient sea salt aerosol. The overestimated sea salt optical depth manifests particularly in the comparison with the $AOD_c$ data from AATSR (NMB=44%) and the 6% overestimate compared to MERGED-FMI-OCN. As discussed above, the lower correlation over the oceans suggests too much hygroscopic swelling, likely resulting in a smoothing of the spatio-temporal variability of marine aerosol.

In contrast, EMEP and GEOS exhibit larger underestimates in AOD when compared with the satellites. In the case of EMEP, this is likely due to too little SS mass, for the reasons discussed above, and it can be seen particularly in the increased underestimate in $AOD_c$. In the case of GEOS, the larger AOD underestimate against the satellites could be due to too coarse and optically too inefficient sea salt particles (comparatively low SS MEC, see Tab. 3), in agreement with Bian et al. (2019).

In contrast to the large underestimate of $AOD_c$, models tend to agree better in the $AOD_f$, showing less underestimation and higher correlation (Tab. 4). To some degree, this may be linked with more harmonised emissions of the associated anthropogenic aerosol species. However, as was shown in Section 3, models also show considerable diversity in fine aerosol species (e.g., OA, $SO_4$, see Fig. 2), a result of differences associated with secondary formation but also due to differences in MEC. Hence, it is interesting to see that models tend to agree better in $AOD_f$ when compared with observations. In addition, the associated particle sizes in the accumulation mode are optically more efficient. Therefore, the $AOD_f$ may be influenced also by dust or sea salt, dependent on the region - even though their mass primarily resides in the coarse regime. For instance, as discussed above, approximately 15% of the sea salt OD resides in the fine mode. The dust fine-mode fraction was found to be circa 30% and was retrieved based on dust fine mode ODs submitted by the same models used above to estimate the SS fine-mode fraction (i.e., CAM5-ATRAS, GEOS, GISS-OMA, SPRINTARS, EC-Earth, TM5, ECMWF-IFS). Compared to AERONET, ENS-MED overestimates $AOD_f$ in several regions including the Mediterranean, the US, Australia and the Arabian Peninsula (see Fig. 5c). The latter could be an indication of too fine dust size distributions contributing to the $AOD_f$. Overall, these overestimates of $AOD_f$ in some regions tend to have a compensating effect on the underestimate in $AOD_c$, resulting in a seemingly better model performance for total AOD (see Fig. 5). This could be due, in part, to uncertainties related to aerosol size (e.g., too much fine dust, which is optically more efficient and results in an overestimate of $AOD_f$, while missing in the

coarse mode). However, it could also be simply due to too much fine anthropogenic aerosol such as $SO_4$ and too little coarse aerosols. GFDL-AM4, for instance, exhibits considerable overestimates of $AOD_f$ compared to both AERONET and AATSR, while strongly underestimating $AOD_c$, resulting in a quite good agreement in total AOD. All three parameters (AOD, $AOD_f$ and $AOD_c$) exhibit high correlation. Compared to the ensemble median, this model shows above average MECs and ODs for both OA and $SO_4$, which could explain the overestimated $AOD_f$. DU and SS, on the contrary, are close to the ensemble median, which would explain the underestimate of $AOD_c$. Other models that overestimate $AOD_f$ include ECHAM-HAM, OsloCTM3 (both against AERONET and AATSR) and EMEP (only against AERONET). It is difficult to identify a common characteristic amongst these models that might explain the overestimate, but in the case of ECHAM-HAM and OsloCTM3 it may be due to comparatively large sulphate burdens and ODs which could be linked with above average DMS emissions (not shown). For EMEP, on the other hand, it is more likely due to its relatively large contributions from OA, NO3 and NH4 (see Fig. 3), since $SO_4$ appears to be underrepresented (see Tab. 3). Another possible reason for EMEPs overestimation of $AOD_f$ is that fine sea salt and dust particles are assumed to have diameters smaller than 2.5 $\mu$m, so that the extinction due to sea salt and dust aerosols with diameters between 1–2.5 $\mu$m contributes to the (overestimated) fine mode rather than the (underestimated) coarse mode. These results suggest a complex interplay among various model assumptions related to composition and size. (Mortier et al., 2020b) also find poorer model performances (larger biases and higher inter-model variability) in long term-trends of the $AOD_c$, as compared to measurements of total AOD and $AOD_f$.

## 4.2 Column Angstrom Exponent (AE)

Models are fairly consistent in their underestimate of AE (see Tab. 4). This suggests that they are (1) either simulating larger particles than are observed or (2) overestimating the fine-mode fraction. EMEP shows the largest overestimate in AE, both compared to AERONET and AATSR-SU. This is likely related to the lower MEC and burden for SS and DU relative to the ensemble model (Tab. 3) and the cut-off for fine SS and DU at 2.5 $\mu$m. INCA, in contrast, exhibits the largest underestimate of AE for both AERONET and AATSR-SU which is likely linked with the comparatively long dust and sea salt lifetimes, and the corresponding high burdens (Tab. 3). The two ECHAM models exhibit similar species optical depths to those simulated by INCA (Fig. 3), so other factors (e.g., assumptions about coarse mode size distributions) may also play a role.

As discussed above, the comparison with AATSR-SU (Fig. 5f) suggests that models underestimate AE over the oceans, however, there is a large uncertainty due to difficulties in validating satellite products over the oceans. In this context, the apparent large overestimate of AE over Australia compared to AATSR (Fig. 5f) is likely associated with retrieval errors in the satellite product. The few AERONET sites in Australia suggest that ENS-MED underestimates AE (Fig. 5e).

Figure 5e shows that models tend to overestimate AE in dusty regions (e.g., Sahara, Arabian Peninsula), suggesting that they tend to simulate too small dust particles. In the case of the Arabian Peninsula this is consistent with the overestimated $AOD_f$ and the underestimated $AOD_c$ (Fig. 5c,d).

However, several regions exhibit a larger underestimate in $AOD_c$ than $AOD_f$ in ENS-MED, while also underestimating AE (e.g., Europe). As can be seen in Table 4, five out of the 14 models (ECHAM-HAM, ECMWF-IFS, GISS-OMA, OsloCTM3, SPRINTARS) show such apparent inconsistencies among AE, $AOD_f$ and $AOD_c$ in comparison with AERONET. To some

degree, these seemingly inconsistent results for these three parameters may be affected by uncertainties related to the separation of $AOD_f$ and $AOD_c$ both in the models (as discussed above) and the observations (as discussed for AERONET in Sect. 2.1.1). GISS-OMA, OsloCTM3 and SPRINTARS are among the models which use the dry particle radius to split AOD into $AOD_f$ and $AOD_c$. In addition, OsloCTM3 simulates dust and sea salt in 8 size bins, and one of these bins contributes both to the
R<0.5$\mu$m and R>0.5$\mu$m regime but is accounted for in the $AOD_f$ diagnostic computed for AeroCom. In the case of ENS-MED, the results may be affected by the averaging choices related to AE and $AOD_f$ (see Sect. 2.3.1). However, even though these uncertainties are not negligible they cannot explain the large differences observed in the biases. ECHAM-HAM, for instance, shows an AE underestimation of 23% compared to AERONET, while at the same time overestimating $AOD_f$ by ca 11% and underestimating $AOD_c$ by ca 57%.

This analysis clearly shows the difficulty of interpreting model AE biases averaged over whole regions and on an annual basis. As Schuster et al. (2006) point out, the relationship between AE and the aerosol size distribution is complex and allows only qualitative statements. Here, we found that models tend to underestimate the global median AE, which suggests that overall they tend to simulate an excess of fine particles or relatively too little coarse aerosol. In order to investigate this in more detail, an additional sensitivity study was performed in which the model biases were analysed as a function of the observed
AE. The results of this study are shown in Figure 7 where all models were co-located with AERONET AE observations and segregated by different AE intervals. For coarse mode dominated aerosol with AE<1 (i.e., Bins 1&2 in Fig. 7) the AP3 models tend to overestimate AE, with INCA being the only exception. For instance, ENS-MED overestimates AE substantially (ca +140%) if only AE measurements below values of 0.5 are considered. On the other hand, for more fine mode dominated aerosol (i.e., AE>2, Bins 5&6), all models tend to show larger underestimates in AE compared to the general results presented
above that consider the full range (non-binned) AE measurements. For instance, ENS-MED shows a bias of ca -60% if only AE>2 measurements are considered. These results suggest that simulated fine-dominated aerosol is larger than it should be (based on the observations) while coarse dominated aerosol is smaller than it should be. The latter is consistent with the above observation that AE is substantially overestimated in dusty regions.

## 4.3 Dry surface scattering coefficient (SC$_{\mathrm{dry}}$)

This and the following section present and discuss the results from the comparison with surface "dry" (RH<40%) in situ measurements at GAW sites (see Fig. 1g,h for site locations). Compared to the previously discussed columnar variables, these comparisons are more sensitive to modelling uncertainties associated with the vertical dimension (such as transport and mixing). On the other hand, these comparisons of "dry" variables are "closer" to the simulated aerosol, as they minimise (but do not eliminate) the impact of water uptake on the optical properties.

As with loading through the column, most models also underestimate loading at the surface as indicated by the primarily blue shading for surface scattering in Table 4. The model median bias is -35% for dry scattering, however, the individual model results show considerable diversity, similar to those for $AOD_c$ (Fig. 6). Note that these "global" surface scattering values are primarily representative of Europe and the US, where the in situ site density is highest. However, as can be seen in Figure 5a,g the magnitude of underestimation appears to be much larger at many GAW sites in Europe and the US, relative to the

AERONET results (which also appear to show spatially smoother variation in the biases). For comparison, over Europe and the US, ENS-MED exhibits an AOD bias of -13% compared to AERONET (see web results, Mortier et al., 2020c).

    It is difficult to observe any spatial tendencies due to the sparsity of sites, although it appears that models tend to overestimate scattering in the eastern US, a region that also shows fairly good performance in AOD, both compared to AERONET and MERGED-FMI (see Fig. 5). In addition, models tend to underestimate $SC_{dry}$ at the few polar sites, a pattern that can also be 1070  observed in the AOD data, suggesting problems related to poleward mass transport.

    The discrepancy in biases between surface and column loading could be related to the simulated vertical profiles or could indicate that too much light extinction is attributed to water uptake. The latter would be consistent with a recent study by Burgos et al. (2020) who find that current climate models tend to overestimate the scattering enhancement due to hygroscopic growth. NorESM2, for instance, shows a large underestimate of $SC_{dry}$ at GAW sites (ca 62%). It also shows a large underestimate 1075  of ambient column AOD over land (compared to AERONET ca -46%), which suggests that the model is missing aerosol mass, particularly more fine mode dominated species such as $SO_4$ and OA (cf., Tab. 3). The larger underestimate of $SC_{dry}$ compared to AOD could indicate that water uptake may compensate ambient AOD to some degree for the missing fine mass. A similar behaviour can be observed for the two ECHAM models. ECHAM-HAM, ECHAM-SALSA and NorESM2, are among the models with the largest contribution of water to AOD (Tab. 4) and they are the models that show the largest 1080  underestimate and lowest correlation in $SC_{dry}$. SPRINTARS is another model showing a large negative bias, however it exhibits a slightly better correlation with the observations compared to ECHAM-HAM, ECHAM-SALSA and NorESM2. SPRINTARS' underestimation of $SC_{dry}$ is also more comparable with the large underestimations found in the other variables suggesting that this model is missing mass rather than over or under emphasizing water availability or uptake.

    Overall, the correlation between modelled and observed $SC_{dry}$ is generally a little lower than the columnar variables. This 1085  is not surprising since the surface measurements do not represent a whole atmospheric column and are thus, more sensitive to associated uncertainties in the vertical transport (e.g., mixing in the boundary layer, convection and associated changes in lifetime and long range transport) which determine the near surface aerosol mixture, size distributions and associated scattering.

    ECMWF-IFS and GISS-OMA are the only two models that show a slight overestimation of $SC_{dry}$ (12% and 18%, respectively), and both exhibit fairly good correlation (R~0.60). Based on the other results for these models (see Tabs 3&4), no clear 1090  explanations can be provided for these results and it would require more detailed investigations of the temporal and spatial distributions (particularly in the vertical), predominant aerosol types and mixtures, and the resulting size distributions.

    Issues with simulated aerosol size likely also play a role in the ability of models to simulate surface scattering. Spectral scattering measurements are available for most surface sites so AE for the surface scattering measurements could be compared with model simulations of AE. Many sites also measure surface scattering at two size cuts (PM1 and PM10, Andrews et al., 1095  2019) which could provide a further constraint on the aerosol size evaluation.

    However, note again that this inter comparison bears some uncertainties, for instance, the model data corresponds to RH=0% while the measurements correspond to RH<40%, which is typically considered "dry". Measurements by Zieger et al. (2013) suggest that hygroscopic growth at low RH could lead to an enhancement in scattering of up to 20% for the so-called "dry"

aerosol measured at RH=40%. On average, the measurements considered here were performed at ~24% RH, thus the impact of light scattering enhancement in the observations should be well below 20% (although that will be site / season specific).

Another uncertainty is the use of a climatology utilizing data available between 2005-2015. Comparison with climatological values was done to increase the number of sites considered as many sites began measuring after 2010 (see Figure 3 in Laj et al., 2020). However, we investigated the differences and found that $SC_{dry}$ is larger for most models when comparing them with only 2010 surface measurements (not shown). For instance, ENS-MED shows an underestimation of -42% when compared to 2010 observations, as opposed to -35% when compared to the 2005-2015 climatology as was done here. The same comparison was done for $AC_{dry}$, indicating minimal differences in the $AC_{dry}$ bias (ENS-MED bias of -21.3% for 2010 and -20.3% using the climatology). The 2010 results are also available online and can be compared with the climatology based results (see EBAS-CLIM, EBAS-2010 in Mortier et al., 2020c). One factor that likely impacts the climatological results is that (1) the site density increased in more recent years (e.g., Laj et al., 2020), particularly in Europe and the US, and (2) these regions are associated with negative trends in scattering and absorption (Collaud Coen et al., 2020; Mortier et al., 2020b). This could shift the weight in the climatology to more recent measurements, or to regions affected by larger changes between 2005-2015. While it would be useful to further explore the impact of climatology versus exact temporal matching, such effort is beyond the scope of this paper. Fortunately, our choice of using the surface data climatology for model evaluation does not appear to impact our results substantially, primarily because of the large differences among the models and their tendency to underestimate these variables substantially (see Tab. 4).

## 4.4 Dry surface absorption coefficient ($AC_{dry}$)

Models tend to underestimate the surface aerosol absorption coefficients. The results of ENS-MED suggest an average underestimation of ca 20%. Interestingly, compared to $SC_{dry}$, the $AC_{dry}$ observations are better correlated with the model simulations (R~0.75) and also less underestimated. This could be linked with the fact that $AC_{dry}$ depends less on hygroscopic growth and that absorption is mostly associated with a single species (BC), which reduces impacts of mixing on size distributions and optical properties. This is particularly the case for anthropogenic source regions like Europe and the US, which is consistent with the low inter-model diversity in $AC_{dry}$ in Europe and the US, compared to $SC_{dry}$ (see Fig. 4j,l). Note however, that the $AC_{dry}$ results primarily represent Europe (and less the US compared to $SC_{dry}$) as can be seen in Figure 5g,h.

At the few arctic sites, models tend to significantly underestimate $AC_{dry}$ (apart from Barrow, where good agreement is found) as suggested by the ENS-MED results shown in Figure 5g,h. This could be due to insufficient mass transport into the Arctic or could also be linked with arctic winter time phenomena resulting in increased domestic BC and OC emissions, combined with trapped air-masses due to typically strong winter time inversions, with substantial impacts on arctic climate change (see e.g, Sand et al., 2013, and references therein). The Russian wildfires in 2010 (e.g., Mielonen et al., 2012) may also have played a role in this. Another pattern is an apparent overestimate of $AC_{dry}$ at the few northern Scandinavian sites, as opposed to the underestimations in most of Europe. Whether this behaviour is linked with 2010 wildfires and / or insufficient transport and local emissions, should be investigated in more detail. Two sites in Asia (both on the Korean peninsula) also show overestimated $AC_{dry}$, while exhibiting underestimated $SC_{dry}$ (see Fig. 5g). Reasons for that should be examined in detail but

may be linked with overestimated impacts of local anthropogenic emissions and / or transport. The same applies for the Cape Point site in South Africa.

With a bias of -55% SPRINTARS exhibits the largest underestimate of $AC_{dry}$. This is consistent with the results found for all other extensive variables investigated for this model. SPRINTARS also exhibits the lowest BC MAC value (which was also the case in AP1, see, Koch et al., 2009). This would most certainly contribute to the underestimate in $AC_{dry}$ as the BC burden for SPRINTARS is comparable to those for the other models. This further suggests that the assumptions describing the conversion from BC mass to absorption may play a role and are likely linked with the simulated size distributions (which depend on aging and mixing) and with the lifetime of BC. Unfortunately, SPRINTARS did not provide the diagnostics to derive BC lifetime. However, as all other species show below average lifetimes in SPRINTARS, it is not unlikely that BC does so too. In summary, the short lifetimes and large underestimations in loading point to too efficient removal processes in SPRINTARS.

Five models overestimate surface absorption (EC-Earth, TM5, ECMWF-IFS, GFDL-AM4 and OsloCTM3). With a bias of +58%, OsloCTM3 shows the largest overestimate of $AC_{dry}$. This is related to the implementation of new absorption parameterisations in the model, resulting in a fairly high BC MAC of 13 $m^2/g$, in combination with a strong vertical gradient of BC (and OA) in the lower atmosphere (not shown). The considerable difference in OsloCTM3's ability to simulate $AC_{dry}$ (overestimated) and $SC_{dry}$ (underestimated) are likely linked with the fact that $SO_4$, one of the main contributors to scattering, does not show such a strong vertical gradient[5].

As with ECMWF-IFS's overestimate of $SC_{dry}$, no clear explanation can be provided for the positive bias in near surface $AC_{dry}$ for this model. However, the fact that both variables are overestimated in this model suggests that the results are linked with differences in the vertical transport in ECMWF-IFS compared to the other models. This is also supported by the observation that all species column ODs for ECMWF-IFS are average or below-average (see Tab. 3). GFDL-AM4, TM5 and EC-Earth did not provide the diagnostics to compute BC MACs. However, for TM5 and EC-Earth, indications of fairly high BC MACs were found above (see discussion in Sect. 3.1.2). This, together with their above average BC mass burdens, explains their higher $AC_{dry}$ values compared to most of the other models. However, as for OsloCTM3, vertical transport may also play a role and would need to be investigated to fully explain the results.

Koch et al. (2009) found that AP1 models tend to underestimate column AAOD (compared to AERONET). However, they overestimated near surface BC mass concentrations over Europe (by ca 120%) and the US (by ca 20%) while underestimating in Asia (by ca 60%). Their comparison with aircraft measurements over the Americas suggests that aloft, models overestimate BC in the tropics and at mid-latitudes by a factor of ca 8, while underestimating high latitude BC concentrations at altitude. Based on these results (i.e., underestimated AAOD, overestimated BC mass), they conclude that BC mass to optics conversion is likely underestimated, and they suggest investigation of possible improvements to the associated optical parameters (e.g., BC refractive index, particle size, coating). The mostly underestimated AAODs described in Koch et al. (2009) are consistent with the underestimated surface $AC_{dry}$ found here, but not directly comparable since AAOD represents the whole column and is also sensitive to potential water uptake of aged BC. However, recall that, compared to AP1, the simulated BC mass is significantly lower in AP3 (see Sect. 3). A comparison of our results for BC MACs and refractive indices with the AP1

---

[5]Pers. communication with G. Myhre.

values (summarised in Table 1 in Koch et al., 2009) suggests a slight increase in BC MAC from ca 7.4 m$^2$/g (AP1) to 8.5 m$^2$/g (AP3). Also, some of the AP3 models simulate increased absorption by assuming a BC imaginary refractive index of around $ik = 0.75i$ (e.g., CAM5-ATRAS, ECHAM models, NorESM2, TM5, GISS-OMA) while most of the AP1 models used values around $ik = 0.45$ which will increase their MACs for a given BC size distribution. The extent to which this translates into over or underestimations of surface absorption or BC mass concentrations and AAOD should be investigated in detail, particularly also due to the importance of the vertical distribution of absorbing aerosol for forcing estimates (e.g., Samset et al., 2013). Based on AeroCom phase II (AP2) experiments, Samset et al. (2014) find that a BC lifetime of less than 5 days is required to reproduce observations of BC in remote regions. Samset et al. (2013) emphasize the importance of correctly treating the BC vertical uplift and associated long range transport in order to properly assess the impacts on forcing. Considering these findings, the AP3 BC lifetime of 5.5 days is still too long, but is an improvement compared to the lifetime in AP1 (6.5 days). To summarise, it appears that the AP3 models tend to simulate less BC mass but have worked on improving their parameterisations that determine the conversion of BC mass into optical properties.

To conclude, as with SC$_{\mathrm{dry}}$, additional model diagnostics and observations are needed to clearly diagnose impacts of vertical transport and mixing near the surface. Relating surface measurements to vertical profiles is also key (e.g., Leaitch et al. (2019)).

### 4.5 Representativity of the results

As described in Section 2.3, monthly aggregates of the models and observations were co-located in space and time. The resulting monthly mean values from all sampling coordinates (sites / aggregated satellite pixels) were then used to compute the biases (NMB) and correlation coefficients (R). Based on these metrics, the performance of individual models was assessed in the previous sections. The comparison of the (often) temporally incomplete observational records (that are sampled at distinct locations) can introduce considerable representation errors both on spatial and on temporal scales (see e.g., Schutgens et al., 2016, 2017; Wang et al., 2018; Sayer and Knobelspiesse, 2019, and references therein). These errors can affect established biases between model and observations but also other performance measures such as correlation coefficients. We consider this to be the major source of uncertainty for the network averaged statistics (NMB, R) used in this study for the model assessment and inter-comparison (e.g., Tab. 4). Therefore, several sensitivity studies have been performed in order to investigate how potential spatio-temporal representation errors affect the global monthly statistical parameters used in this study.

The following three sensitivity studies were performed in order to investigate temporal and spatial representation uncertainties:

1. (Temporal): Using GAW in situ AC$_{\mathrm{dry}}$ measurements to investigate impacts of temporal representativity errors at the surface. Note that only 2010 observation data was used for this sensitivity study (instead of the 2005-2015 climatology of in situ measurements). The in situ data are co-located in the provided hourly resolution with hourly TM5 data from the AeroCom INSITU experiment. Network averaged statistics (NMB, R) are computed from this hourly co-located data and are compared with results obtained using our analysis strategy (based on monthly averages, see Sect. 2.3). Note that this TM5 model version is slightly different from the AP3-CTRL TM5 model run used in this paper. This is because for

AP3-CTRL, no high resolution data was available for this sensitivity study. However, the choice of the model should have limited impact on the results of this study, as most temporal representativity issues arise from incomplete observation records.

2. (Temporal): Same as 1., but using AOD from AERONET in order to investigate impacts of temporal representativity errors on columnar variables. For this comparison, the AERONET all-points product is used (instead of the daily product) and is compared with 3-hourly output from the ECMWF-IFS model (see Sect. 4 in supplement 2).

3. (Spatial): Spatial representation uncertainties were investigated by using a selection of AERONET sites that is considered representative on spatial scales of a typical model grid. The selection was done based on Wang et al. (2018), using only sites that show an absolute spatial representation error smaller than 10%. This subset was co-located with the AP3 ensemble median (ENS-MED) and results were compared with our results (without AERONET site selection).

The results of these three sensitivity studies are summarised in Table 5. In general, the retrieved differences in NMB and R are small (well below 10% change in NMB and up to ca 0.2 change in R). For instance, for experiments 1 and 2 the difference in NMB is only 0.2% and 1%, respectively and the correlation coefficients are slightly improved in the monthly resolution (which is not surprising, as the coarser resolution will lead to smoother results). Also shown in Table 5 are monthly NMBs and Rs retrieved without applying the 25% time sampling coverage constraint for the observations. For these results, the departures from the high-resolution data set are increased, which illustrates the importance of these resampling constraints.

For experiment 3, the difference in NMB is ca 5%, however, the total number of considered AERONET sites is reduced from 250 to 50, so this difference could also arise from regional shifts in the considered site coverage (e.g., the 50 sites from Wang et al., 2018, could be located in regions where models have a lower bias). Overall, compared to the magnitude and inter-model diversity of biases and correlation coefficients observed in our results (e.g., Tab. 4), we consider these spatio-temporal uncertainties to be acceptable and they do not affect our results (or their interpretation) substantially.

One further uncertainty related to the representativity of the results is that AERONET only measures during daytime, while the models computed 24h averages (as indicated in Sect. 2.1.1). This will cause shifts in the intrinsic weighting applied when computing the network averaged statistics used throughout this paper (e.g., wintertime measurements at high latitudes are restricted to noon-time if they occur at all). In addition, it could introduce systematic errors at locations that show a persistent and pronounced diurnal profile. In this context, note that the GAW in situ observations are not affected by this as they measure continuously, night and day regardless of cloud conditions. The latter is reflected in the very similar results in Test 1 (i.e., hourly vs monthly comparison of $AC_{dry}$). Since the results of test 2 (AERONET 3hourly vs monthly) show very good agreement as well, we believe that uncertainties associated with diurnal variations of AOD are likely small compared to the large uncertainties associated with the correct modelling of the AOD, reflected by the considerable biases (and their diversity) found here among the models. Furthermore, AOD represents the whole atmospheric column and, thus, should be less sensitive to diurnal variations than the near surface measurements. A detailed investigation of associated impacts of diurnal variability is desirable but beyond the scope of this paper. Also in that context, it would be interesting to investigate the extent to which global climate models need to be able to reproduce amplitudes in diurnal variability of certain tracers and physical processes

and which phenomena can be sufficiently parameterised in lower temporal resolution.

For non-geostationary satellites, the absolute temporal representation errors are likely larger due to the low sampling coverage, combined with cloud contamination in certain regions (e.g., the South Pacific). A detailed investigation of these uncertainties is beyond the scope of this work. Nonetheless, a further simple sensitivity study was performed that investigates how our choice of resolution for the satellite / model inter-comparisons affects the retrieved metrics (NMB and R) presented in Tab. 4.

Similar to the three experiments above, this was done by comparing co-location results based on the original satellite products (i.e., $1° \times 1°$, daily aggregates) with results based on our analysis strategy (i.e., $5° \times 5°$ , monthly aggregates), using models that also provided daily diagnostics for the optical properties investigated. The results of this investigation are presented in Section 3 in supplement 2 (see Tab. 3 therein). They include results for the variables AOD, $AOD_f$, $AOD_c$ and AE. In most cases, co-location in the higher spatio-temporal resolution results in positive shifts in the respective model NMB and the as-

sociated differences can be up to +13% (e.g., AE SPRINTARS vs. AATSR-SU). However, in most cases the differences are marginal and are well below 5%. Correlation coefficients are generally higher in the lower resolution data, for the same reasons mentioned above (i.e., due to the smoothing effect, intrinsic in the data aggregation).

Finally, we want to stress that the results from these sensitivity studies give insights into uncertainties in evaluation metrics (i.e., bias, correlation) based on monthly means, and averaged over many observation locations around the globe. We emphasise

that representation uncertainties may be significantly larger over sub-domains or at specific locations and times, as shown in the various literature referred to above.

## 5   Conclusions

In this study a comprehensive inter-comparison of 14 models from the AeroCom phase III (AP3) control experiment has been performed. Inter-model diversity of key parameters associated with the aerosol lifecycle and optical properties was investigated

for the major aerosol species. These results were compared with results from the AeroCom phase I (AP1) experiments to identify significant differences and improvements in the modelling of the global aerosol. In addition, the models were compared to aerosol observations made at mid visible wavelengths. For this comparison, remote sensing observations of the columnar aerosol optical properties AOD, $AOD_f$, $AOD_c$, and AE from AERONET and several satellite products were used. Furthermore, for the first time, the models were compared with near surface in situ dry scattering ($SC_{dry}$) and absorption coefficients ($AC_{dry}$)

at GAW sites (mostly located in Europe and the US). Finally, the spatial and temporal representativity of the results was evaluated.

The results suggest that overall, models tend to underestimate all optical properties investigated. Comparison of the modelled AODs with AERONET (mostly land based) and a merged satellite AOD product (better global coverage) show mostly consistent results and suggest that AP3 models underestimate the AOD by approximately -20$\pm$20% (based on computed ensemble

median and IQR). A large fraction of the AOD bias is due to an underestimation of the AOD due to coarse particles – the ensemble median results suggest that the AP3 models underestimate $AOD_c$ by 46% (compared to AERONET) while $AOD_f$

is only underestimated by 13%, however note that these are relative biases (i.e., the average fine mode fraction at AERONET sites is ca. 70%). However, some uncertainties and inconsistencies remain in the way models diagnose $AOD_c$ and $AOD_f$, and whether the modelled size cuts are comparable with the observed size cuts.

The large underestimation of $AOD_c$ is associated with uncertainties in the modelling of the natural aerosols sea salt and dust. These two species exhibit wide diversity in their online computed emission strengths and lifetimes, suggesting differences in the simulated size distributions among models, which has clear implications for the conversion to optical extinction. Models tend to agree slightly better in their dust and sea salt MECs, compared to the diversity in lifetimes and emissions, which is likely due to the fact that coarser particle sizes exhibit less spectral variability at mid visible wavelengths than fine particles.

Notably, while the total AOD is comparable to that found during the AP1 evaluations, the relative contribution of dust and sea salt to total AOD has changed substantially since AP1 (where both species contributed approximately equally to AOD). In the AP3 simulations, sea salt is the dominant natural species, contributing approximately 2/3 of the combined SS+DU contribution to AOD. While the emissions of both species decrease in AP3, this shift in their relative contribution is mostly due to lifetime changes. In the case of sea salt the emissions decrease is largely compensated for by an increase in its lifetime, which suggests

that smaller sea salt particles are being simulated (or removal pathways are less efficient). On the other hand, the dust lifetime is slightly decreased compared to AP1, resulting in a ca 25% lower global dust burden.

     Interestingly, the comparison of simulated AE with measurements of the AE from AERONET suggests that models (still) simulate too fine dust aerosol or overestimate the fine mode fraction of coarse dominated aerosol. This would likely translate into an overestimation of the dust MEC which is, however, reduced compared to AP1 and deserves further attention.

Unfortunately, a clear assessment of the modelled sea salt size distributions based on the AE is difficult within the scope of this paper. This is mostly due to a lack of ground based measurements over the oceans and large uncertainties in the satellite AE data, but also because of the impact of water uptake on the simulated size distributions. The longer sea salt lifetime compared to AP1 suggests smaller dry sea salt particles in AP3 models, which has implications for water uptake and associated light scattering enhancement and possibly also for cloud formation and properties (e.g., cloud lifetime and albedo)

in clean marine environments. Indications for and implications of smaller sea salt particles are reflected in results from a few models showing above average sea salt lifetimes, burdens, MECs and water contribution to AOD, as well as overestimates (and decreased correlation) of total AOD and $AOD_c$, particularly in comparison with the satellites over the oceans. An overestimated contribution of water uptake to the light extinction is also supported by a considerable underestimate and inter-model spread in the near surface "dry" scattering coefficients with a bias of approximately -35±25% found when comparing the ensemble

median with GAW in situ aerosol scattering measurements.

     Similar to AP1, the contribution of the other aerosol species ($SO_4$, OA and BC) to total AOD shows considerable variability among the models, while the simulated total AODs are more consistent. The AP3 $SO_4$ lifecycle and optical parameters are mostly comparable with AP1. The source strength of OA (POA + secondary formation) shows an increase of ca 20% relative to AP1 which translates into a similar increase in its mass burden and the contribution of OA to AOD. However, the contribution

of OA to AOD exhibits large variability among the models, which is mostly due to the fact that models that tend to simulate an above average OA burden also show an above average MEC and vice versa. The BC emissions are harmonised, however,

lifecycle processes related to BC exhibit considerable diversity, resulting in a decrease in the BC mass burden by almost a factor of 2 compared to AP1, mostly due to a decrease in BC lifetime from 6.5 days in AP1 to 5.5 days in AP3. At the same time BC MAC is slightly increased with a median of 8.5 $m^2$/g in AP3 but is still lower than recommended literature values. The results indicate improvement in the BC size assumptions and impacts of aging (e.g., hygroscopicity due to internal mixing, or absorption enhancement effects) but are also likely related to shifts in the assumed BC refractive index towards more absorbing aerosol (in some models). These changes from AP1 to AP3 reflect the considerable effort that went into improving models treatment of BC. However, considering the finding of Samset et al. (2014) the BC lifetime in AP3 remains too long.

The lower BC burden is also reflected in considerable underestimations (bias ca -20±18% based on ensemble and IQR) of the near surface dry absorption coefficients at GAW sites. However, there is considerable diversity among the individual 39 GAW sites considered (most of which are located in Europe), indicating impacts of transport and vertical mixing. The $SC_{dry}$ results show larger underestimates (-35%) and more inter-model diversity than the $AC_{dry}$ comparisons. These results reflect uncertainties related to mixing and water uptake for scattering and - since the ambient AOD bias is less negative (-20%) - are consistent with recent findings by Burgos et al. (2020) indicating that current climate models overestimate the light scattering enhancement associated with aerosol water uptake.

A detailed investigation of model biases compared to AERONET AE in different observed AE regimes suggests that models overestimate size (or underestimate the fine mode fraction) in fine mode dominated regimes, while coarse mode dominated regimes indicate the opposite, that is, models simulate not coarse enough particles (or overestimate the contribution of fine aerosol to extinction). Even though the AERONET AE measurements are mostly land based, this could further indicate that not only the hydrophobic dust, but also sea salt particles are too small. This could explain a possible overestimation of the light scattering enhancement, which impacts fine particles more than coarse particles (Zieger et al., 2013). However, we note that such a hypothesis would need to be investigated in more detail for the AP3 models to make clear statements related to the sea salt size and water uptake.

The newly introduced $NO_3$ aerosol component in 9 of the 14 models exhibits very large variability in all lifecycle and optical parameters investigated and it was shown that this is mostly linked to differences in the assumed size distributions but, to some degree, also due to inconsistencies in the associated diagnostics submitted to AeroCom. However, the impact of these uncertainties on total AOD are small as $NO_3$ contributes a small fraction to the total AOD (<5% based on the median results of this study).

The quality of the constructed AeroCom median ensemble model is a very solid reference for the parameters investigated. In terms of correlation, only in a few instances do individual models outperform it; however no single model is better than the ensemble model on all parameters. This paper and its associated supplements, the AeroCom database and web interfaces are available as reference for further investigations, in particular for developing recommendations for global aerosol modelling. Model diagnostics are comprehensive but clearly not enough to understand all aspects on how models simulate the path from emissions to optical properties. The documented inconsistencies in aerosol life cycle and mass to absorption / scattering coefficients are hopefully an encouragement for modellers to further investigate their individual parameterisations and diagnostics.

The consistency of model performance against different AOD observational datasets speaks in favor of the quality of the observations. The performance variation against different AOD datasets provides at the same time a rough error estimate or robustness of our evaluation method. Our sensitivity tests with respect to spatio-temporal resolution and site selection indicate that the results are robust. Surface in situ scattering and sun photometer / satellite derived AOD data have been used for the first time in a consistent way for evaluating a multi-model ensemble revealing an underestimate of aerosol AOD, though smaller for anthropogenic, fine mode aerosol. Using fine mode AOD as a proxy for present day aerosol forcing estimates, our results suggest that models underestimate aerosol forcing by circa -15%. However, the associated inter-model spread (quantified by the IQR) is between -35% and +10% and summarizes the large disparity between the individual models, stressing the need for further research.

In future studies the biases found in this study should be investigated, for instance, by incorporating additional aspects into the analysis such as model resolution (particularly vertical), regional and seasonal variations, profile extinction data or column water content (to assess hygroscopic growth). Delving into the details of assumed size distributions, particularly for natural aerosol could also help resolve some of the discrepancies reported on here. In addition, a comparison with mass concentration measurements at the surface and aloft could provide valuable insights related to the question of whether the models are missing mass or whether assumptions about optical properties are causing the underestimated scattering coefficients and optical depth. Such an analysis would certainly benefit also from a better global coverage of surface measurement sites. The pre-industrial aerosol state (although available in the AeroCom AP3 control experiment) has not been incorporated yet, but should be included to provide more insight into aerosol forcing estimates and link to the CMIP6 model ensemble results on historical climate evolution.

*Code and data availability.* Most of the data analysis was performed using the open source software pyaerocom (version 0.10.0, release upcoming). All additional analysis scripts are stored in a private GitHub repository and can be provided upon request. All data used in this study is stored on servers of the Norwegian Meteorological Institute and can be provided upon request. The GAW data used is accessible through the EBAS database.

*Author contributions.* JGl, MS and AM designed the study. JGl did most of the analysis and wrote most of the manuscript, together with substantial contributions from MS, EA. AM, JGr and MS contributed the interactive websites associated to this study. EA, AH, PL, PN, CM, JGr and LS helped analysing and interpreting the observational data used. AB and JGr helped organising the AeroCom database and AeroCom infrastructure needed. YB, SB, HB, RCG, MC, PG, ZK, AK, HK, PS, ML, HM, GM, DN, TN, DO, SR, TT, KT and ST provided model data and model interpretations. All co-authors gave feedback to the manuscript.

*Competing interests.* No competing interests.

*Acknowledgements.* Data providers from all observation networks and satellite products and all people involved in the production of these products are highly acknowledged for sharing and submitting their data and for support. JGl wishes to thank O. Landgren and H. Svennevik for providing data and code used in this study, and S. Kinne for helpful comments via personal communication. TT was supported by the supercomputer system of the National Institute for Environmental Studies, Japan, Japan Society for the Promotion of Science (JSPS) KAKENHI (JP19H05669), and Environment Research and Technology Development Fund (JPMEERF20202F01) of the Environmental

Restoration and Conservation Agency, Japan. TvN and PLS acknowledge funding from the European Union's Horizon 2020 research and innovation programme project CRESCENDO under grant agreement No 641816. DN acknowledges funding from the European Union's Horizon 2020 research and innovation programme project FORCeS under grant agreement No 821205. SEB and KT acknowledge funding from NASA's Atmospheric Composition Modeling and Analysis Program (ACMAP), contract number NNX15AE36G. PG acknowledges partial funding from NASA's Earth Surface Mineral Dust Source Investigation (EMIT), program number NNH12ZDA0060-EVI4. They

also thank Jingbo Yu for running the GISS model. Resources supporting this work were provided by the NASA High-End Computing (HEC) Program through the NASA Center for Climate Simulation (NCCS) at Goddard Space Flight Center. HK acknowledges the Academy of Finland Projects 317390 and 308292. The ECHAM-HAMMOZ model is developed by a consortium composed of ETH Zurich, Max Planck Institut für Meteorologie, Forschungszentrum Jülich, University of Oxford, the Finnish Meteorological Institute and the Leibniz Institute for Tropospheric Research, and managed by the Center for Climate Systems Modeling (C2SM) at ETH Zurich. PN and AH are

supported under the ESA Aerosol Climate Change Initiative (Aerosol CCI). HM acknowledges funding from the Ministry of Education, Culture, Sports, Science, and Technology and the Japan Society for the Promotion of Science (MEXT/JSPS) KAKENHI Grant Number JP17H04709. JGl, AK, DO, and MS acknowledge support from the Research Council of Norway funded projects EVA (229771), INES (270061), and KeyClim (295046), and the Horizon 2020 project CRESCENDO (Coordinated Research in Earth Systems and Climate: Experiments, Knowledge, Dissemination and Outreach, no. 641816). High performance computing and storage resources were provided by

the Norwegian infrastructure for computational science (through projects NN2345K, NN9560K, NS2345K, and NS9560K). GM has received support from the project SUPER (no. 250573), funded through the Research Council of Norway. HB acknowledges funding from NASA's Atmospheric Composition Modeling and Analysis Program (ACMAP), contract number NNX17AG31G. MC acknowledges support by the NASA Earth Science programs. We thank two anonymous reviewers for careful reading and constructive suggestions.

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

1805

**Table 1.** Observations used in this study, including relevant meta data information. ID: name of observation network; Source: Data source or subset; Var: Variable name; NMB: Normalised mean bias of satellite product at AERONET sites (monthly statistics); $\lambda$: Wavelength used for analysis (may be different from measurement wavelength, for details see text); Ver: Data version; Lev: Data level; Freq: Original frequency of data used to derive monthly means; Res: Resolution of gridded data product; Clim: Use of a multi-annual climatology or not; #st: Number of stations / coordinates, with observations used; Date: Retrieval date from respective database. See text in Section 2.1 for additional quality control measures that have been applied to some of these data sets.

| ID | Source | Var. | NMB [%] | $\lambda$ [nm] | Ver. | Lev. | Freq. | Res. | Clim. | st. | Date |
|---|---|---|---|---|---|---|---|---|---|---|---|
| GAW | EBAS | $AC_{dry}$ | | 550 | | 2 | hourly | | Y | 39 | 2020/04/01 |
| GAW | EBAS | $SC_{dry}$ | | 550 | | 2 | hourly | | Y | 39 | 2020/04/01 |
| AERONET | AOD | AE | | 4-$\lambda$ fit[1] | 3 | 2 | daily | | N | 245 | 2020/08/13 |
| AERONET | AOD | AOD | | 550 | 3 | 2 | daily | | N | 235 | 2020/08/13 |
| AERONET | SDA | $AOD_f$ | | 550 | 3 | 2 | daily | | N | 222 | 2020/08/13 |
| AERONET | SDA | $AOD_c$ | | 550 | 3 | 2 | daily | | N | 222 | 2020/08/13 |
| MODIS-T | Terra DT/DB | AOD | +16.5 | 550 | 6.1 | 3 | daily | 1x1 | N | 246 | 2019/11/22 |
| MODIS-A | Aqua DT/DB | AOD | +8.9 | 550 | 6.1 | 3 | daily | 1x1 | N | 246 | 2019/11/25 |
| AATSR-SU | Swansea | AOD | -4.2 | 550 | 4.3 | 3 | daily | 1x1 | N | 246 | 2016/09/30 |
| AATSR-SU | Swansea | AE | +14.3 | 550-865 | 4.3 | 3 | daily | 1x1 | N | 257 | 2016/09/30 |
| AATSR-SU | Swansea | $AOD_f$ | +1.6 | 550 | 4.3 | 3 | daily | 1x1 | N | 233 | 2016/09/30 |
| AATSR-SU | Swansea | $AOD_c$ | -14.7 | 550 | 4.3 | 3 | daily | 1x1 | N | 233 | 2016/09/30 |
| MERGED-FMI | FMI | AOD | -5.5 | 550 | | | daily | 1x1 | N | 246 | 2019/10/21 |

[1]AERONET's 4-$\lambda$ fit is based on these four wavelengths: 440-500-675-870.

**Table 2.** Models used in this study including relevant additional information. Kinne et al., 2006: name of model in Kinne et al. (2006) (see table 2 therein; where applicable); Lat./Lon.: horizontal grid resolution; Levs.: number of vertical levels; Type: type of atmospheric model; Aerosol module: name of aerosol module; Scheme: type of aerosol scheme; Meteorology: meteorological data set used for the simulated year 2010; CS: clear-sky optics available (Y/N); AC.: availability of dry surface absorption coefficient fields for comparison with GAW observations; References: key references. More details about the models can be found in the supplementary material 1&2.

| Model name | | | | | | | | | | |
| This study | Kinne et al., 2006 | Lat./Lon. | Levs. | Type | Aerosol module | Scheme | Meteorology | CS | AC | References |
| --- | --- | --- | --- | --- | --- | --- | --- | --- | --- | --- |
| CAM5-ATRAS | N/A | 1.9 x 2.5 | 30 | GCM | ATRAS | Sectional | MERRA2 (nudged, above 800 hPa) | Y | Dry | Matsui (2017), Matsui and Mahowald (2017) |
| EC-Earth3-AerChem | N/A | 2.0 x 3.0 | 34 | GCM | TM5-M7 | Modal | ECMWF-IFS (online) | Y | Dry | van Noije et al. (2014), van Noije et al. (2020) |
| TM5 | TM5 | 2.0 x 3.0 | 34 | CTM | TM5-M7 | Modal | ERA-Interim (driven) | Y | Dry | van Noije et al. (2014), van Noije et al. (2020) |
| ECHAM-HAM | MPI-HAM | 1.9 x 1.9 | 47 | GCM | HAM-M7 | Modal | ERA-Interim (nudged) | Y | Dry | Tegen et al. (2019) |
| ECHAM-SALSA | N/A | 1.9 x 1.9 | 47 | GCM | SALSA | Sectional | ERA-Interim (nudged) | Y | Dry | Bergman et al. (2012), Kokkola et al. (2018) |
| ECMWF-IFS | N/A | 0.4 x 0.4 | 137 | GCM | AER | Bulk / Sectional | ECMWF-IFS | N | Dry | Rémy et al. (2019) |
| EMEP | N/A | 0.5 x 0.5 | 20 | CTM | N/A | N/A | ECMWF-IFS (driven) | N | Dry | Simpson et al. (2012), Schulz et al. (2012) |
| GEOS | GOCART | 1.0 x 1.0 | 72 | ESM | GOCART | Bulk | MERRA2 (nudged) | N | Dry | Colarco et al. (2010), |
| GFDL-AM4 | N/A | 1.0 x 1.2 | 33 | GCM | GOCART | Bulk | NCEP-NCAR re-analysis (nudged) | N | Amb. | Zhao et al. (2018) |
| GISS-OMA | GISS | 2.0 x 2.5 | 40 | ESM | OMA | Mass based, sectional for SS & DU | NCEP-NCAR | Y | Dry | Koch et al. (2006, 2007), Tsigaridis et al. (2013) |
| INCA | LSCE | 1.3 x 2.5 | 79 | GCM | INCA | Modal | ERA-Interim (nudged) | Y | N/A | Balkanski et al. (2004), Schulz et al. (2009) |
| NorESM2 (CAM6-Nor) | UIO_GCM | 0.9 x 1.2 | 32 | GCM | OsloAero | Production tagged (size resolving through offline lookup tables) | ERA-Interim (nudged) | Y | Dry | Kirkevåg et al. (2018), Olivié et al. (2020), Seland et al. (2020) |
| OsloCTM3 | UIO_CTM | 2.2 x 2.2 | 60 | CTM | OsloCTM3 | Bulk / Sectional | ECMWF-IFS (driven) | N | Dry | Lund et al. (2018); Myhre et al. (2009) |
| MIROC-SPRINTARS | KYU | 0.6 x 0.6 | 56 | GCM | SPRINTARS | Modal | ERA-Interim (nudged) | Y | Amb. | Takemura et al. (2005) |

**Table 3.** Global annual averages for each aerosol species, grouped by aerosol emissions, lifetimes , burdens, optical depths (ODs), mass extinction coefficients (MEC) and mass absorption coefficient (MAC), for models participating in the AP3-CTRL experiment, year 2010. Also shown in the OD section are total AOD for all-sky (AOD (AS)) and clear-sky (AOD (CS)) conditions as well as AOD due to water ($H_2O$). The following columns show the median from all model values (MED) and associated diversities as interquartile range and standard deviation ($\delta_{IQR}$, $\delta_{std}$). AP1 median and standard deviation are based on values given in Table 10 in Textor et al. (2006) and Table 4 in Kinne et al. (2006). Colors illustrate the bias of individual model and AP1 median values with respect to the AP3 median. Units of emissions and burdens are full molecular weight (for OA and POA, the total organic weight is used). Note that the "emissions" of $SO_4$, $NO_3$ and OA are really secondary chemical formation in the atmosphere plus primary particle emissions. They are computed using total deposition as a proxy (indicated with ↓). For BC↑, DU↑, POA↑ and SS↑ the provided emission data were used. For OsloCTM3 an additional OD of 0.0086 due to biomass burning was reported and is not included here. See further details on parameter computation in section 2.2. Values in brackets indicate erroneous or inconsistent values (i.e., BC OD, MEC and MAC from some models) and are not included in the corresponding AP3 median value (MED) and diversities (details are discussed in Sect. 3.1).

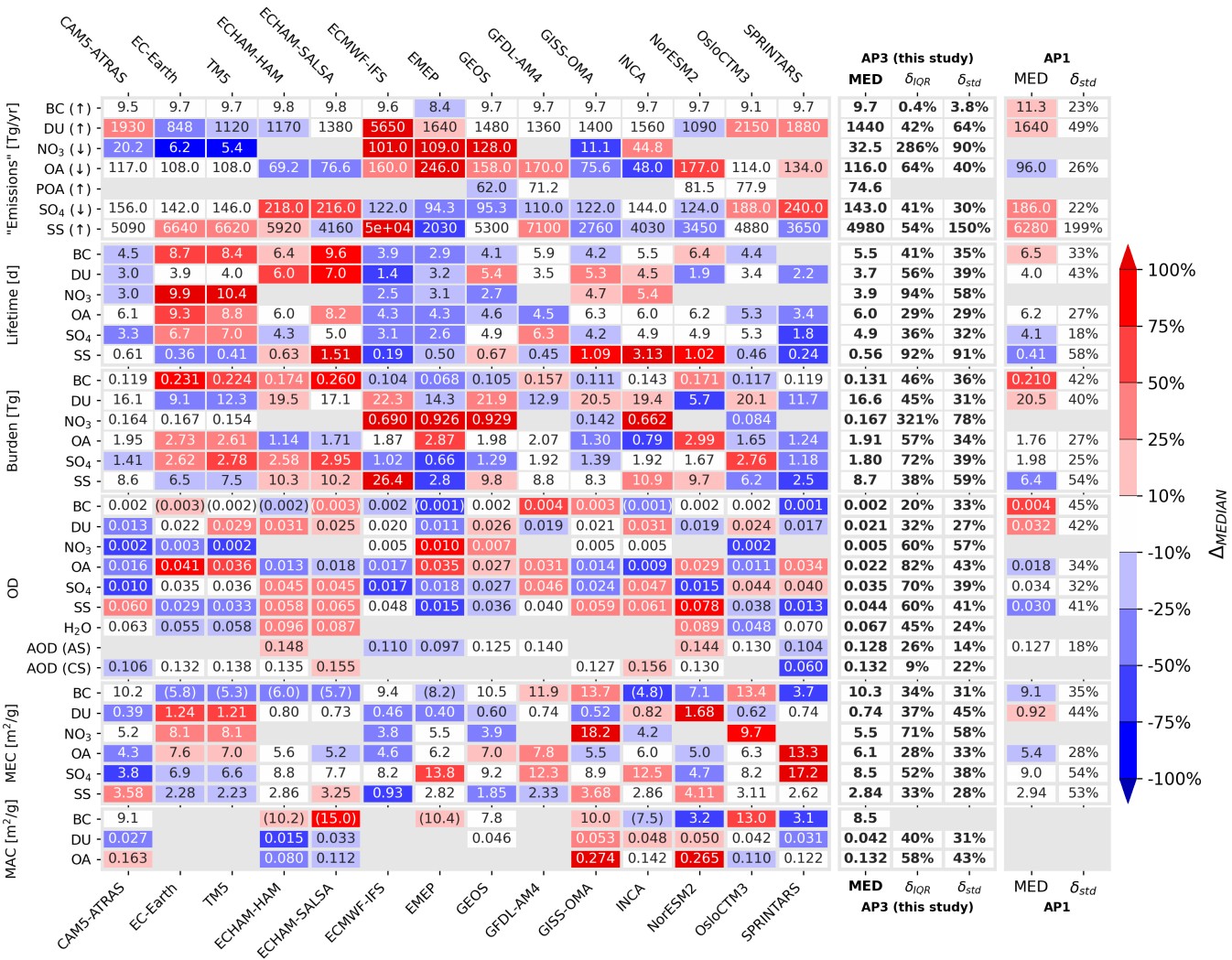

**Table 4.** Normalised mean bias (NMB, top) and Pearson correlation coefficients (bottom), computed from the monthly co-located data for each model (columns) and observation / variable combination (rows). For the comparison with the gridded $5° × 5°$ satellite products, area weights were applied to compute the average metrics shown here. Please note that the biases do not represent global averages but the site / sampling locations of each data set with more weight given to regions with higher spatial density (e.g., Fig. 1). Please also note potential offsets in the absolute biases arising from uncertainties in the observation retrievals, particularly for the satellite products (see Tab. 1). The rightmost columns show evaluation results from the ensemble median (ENS-MED) as well as first and third quantile fields (Q1, Q3). The latter indicate the spread of the results. Model values for AODs are for clear-sky conditions unless only all-sky were available.

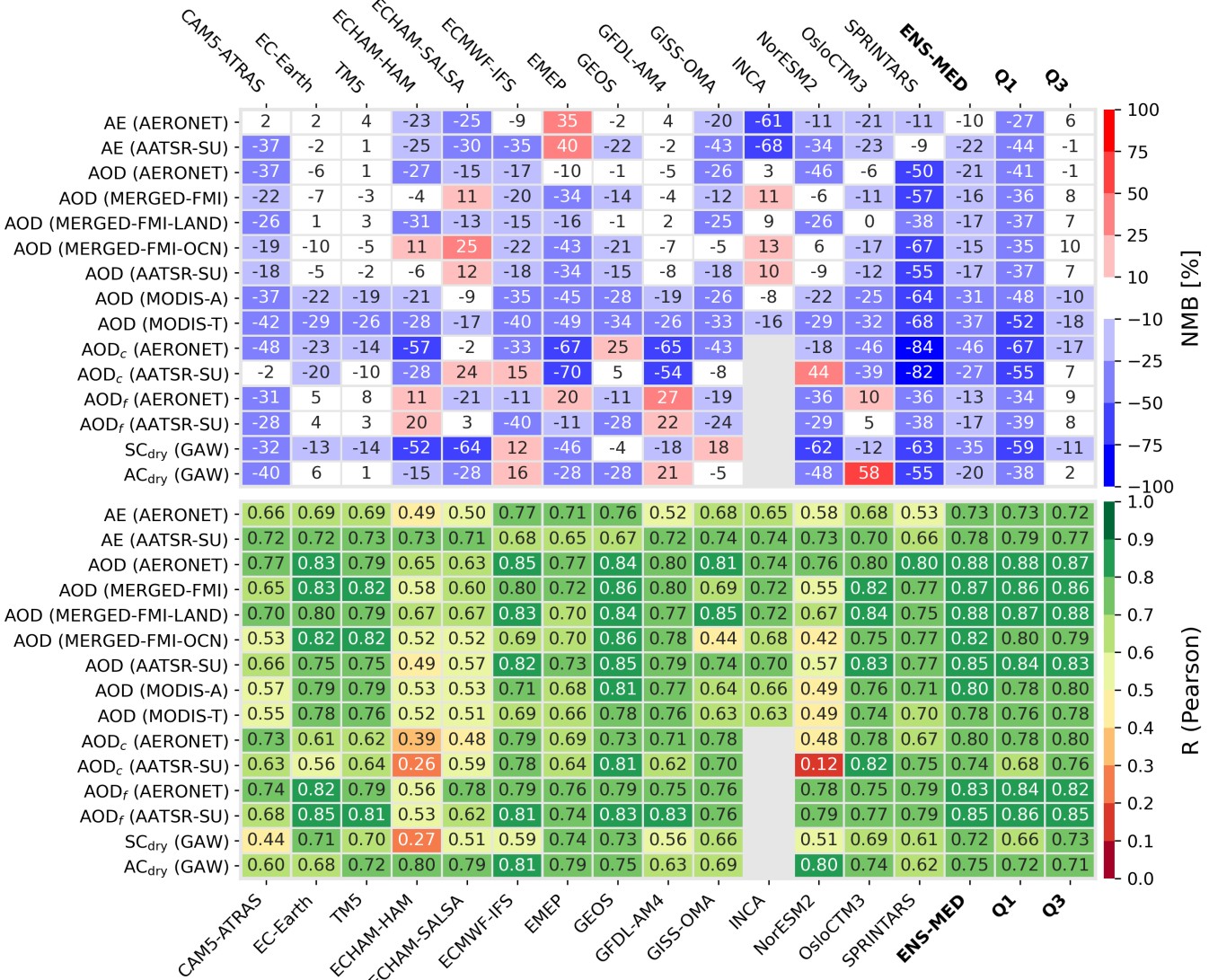

**Table 5.** Results from sensitivity studies related to spatio-temporal representation errors. AERONET* indicates that two different site selection schemes were used (see text in Sect. 4.5 for details).

| | Test type | Var. | Model | Obs. | Freq. | NMB [%] | R | Comment |
|---|---|---|---|---|---|---|---|---|
| 1 | Temporal | $AC_{dry}$ | TM5 (INSITU) | GAW (surface, 2010) | hourly | -5.6 | 0.41 | |
| | | | | | monthly | -5.8 | 0.50 | with 25% sampling coverage (as done in this paper) |
| | | | | | monthly* | -8.2 | 0.59 | No coverage constraint |
| 2 | Temporal | AOD | ECMWF-IFS | AERONET | 3-hourly | -18.4 | 0.71 | |
| | | | | | monthly | -17.4 | 0.85 | with 25% sampling coverage (as done in this paper) |
| | | | | | monthly* | -20.1 | 0.68 | No coverage constraint |
| 3 | Spatial | AOD | ENS-MED | AERONET* | monthly | -16.3 | 0.91 | Selection of sites from Wang et al. (2018) |
| | | | | AERONET | monthly | -20.9 | 0.88 | All sites |

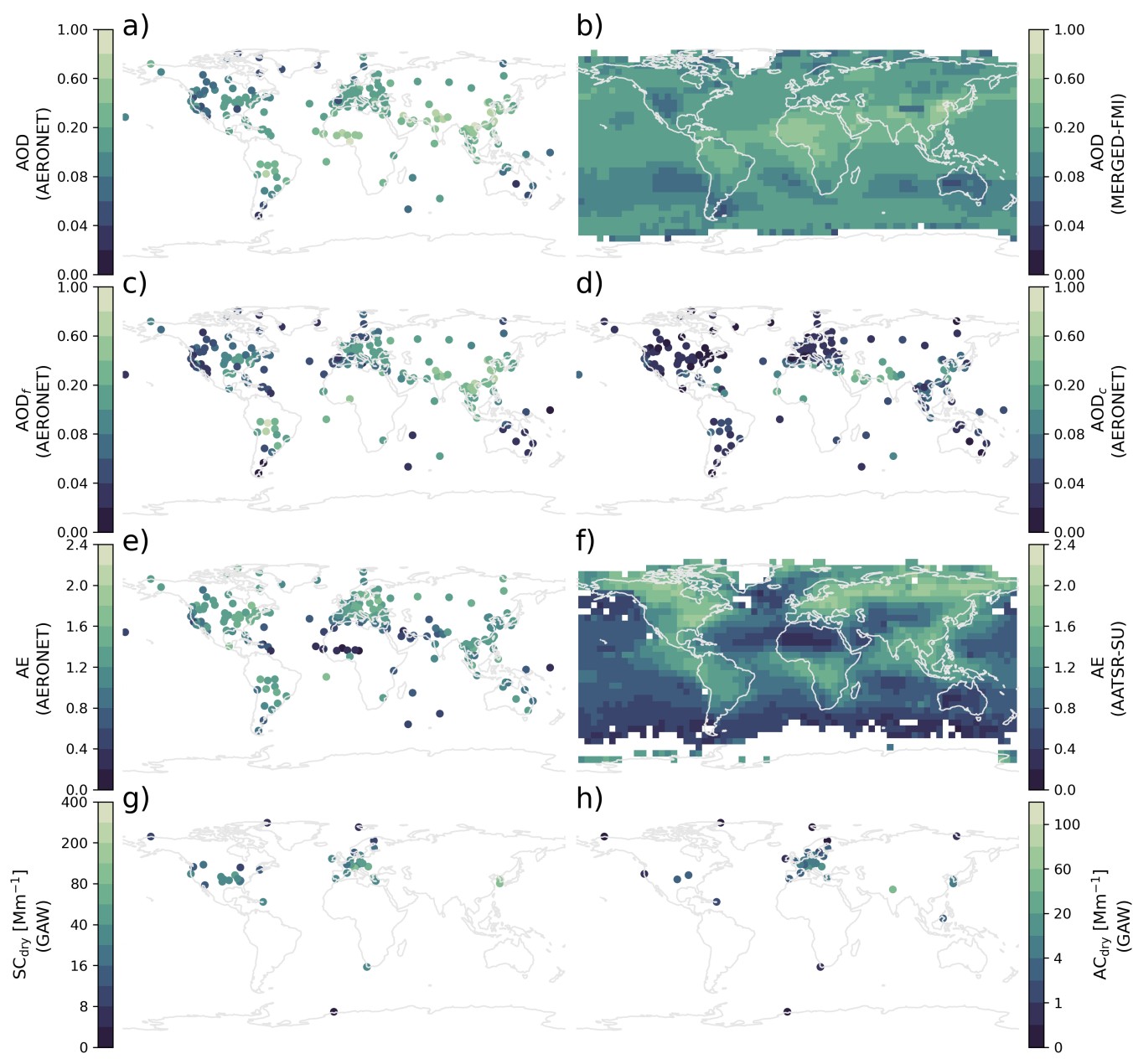

**Figure 1.** Overview of data used for model evaluation. Yearly averages of AODs from a) AERONET; b) merged satellite data set, c) fine and d) coarse AOD from AERONET, e) AE from AERONET, f) AATSR, g) dry scattering and h) dry absorption coefficients from surface in situ observations.

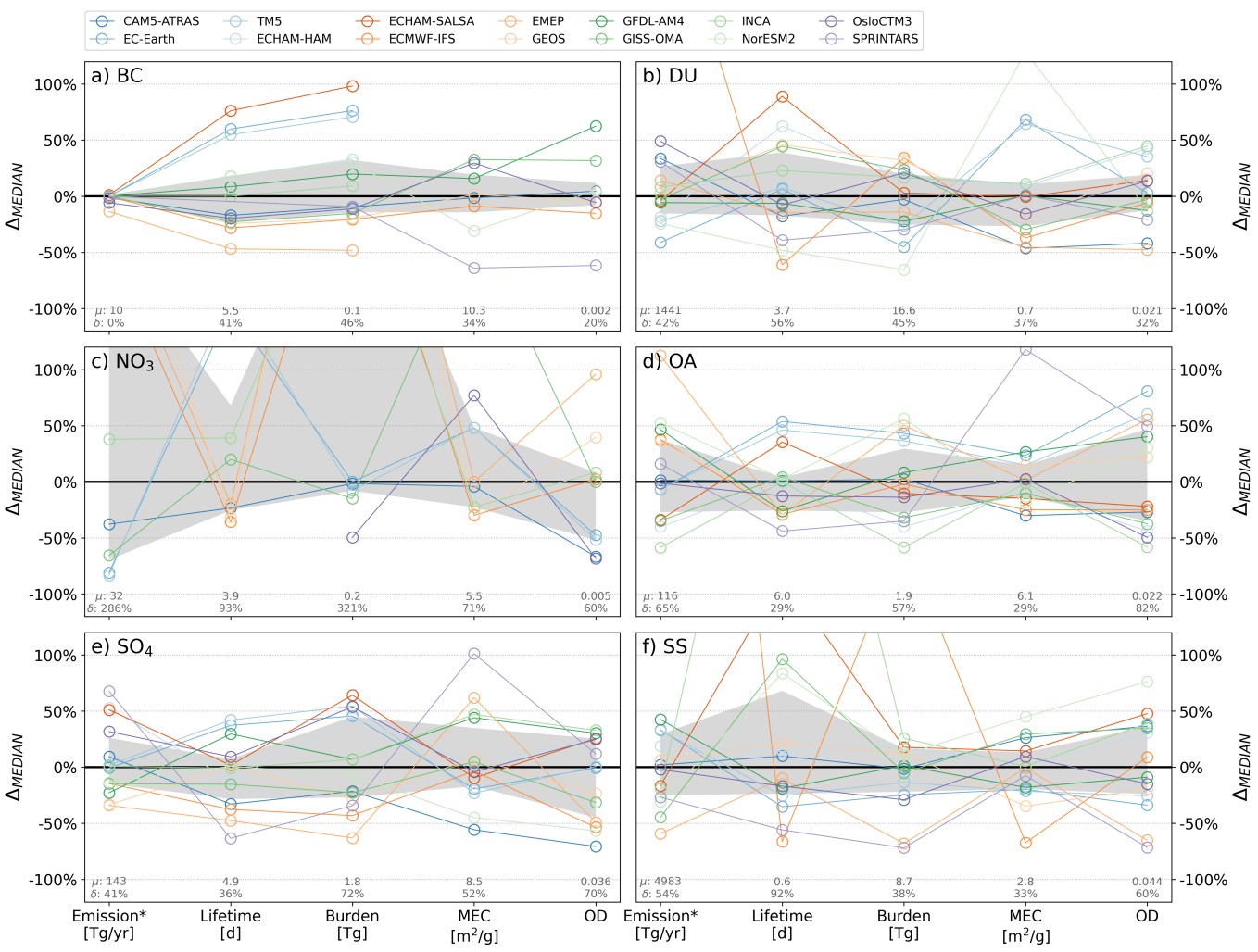

**Figure 2.** Relation between aerosol lifecycle and optical parameters for individual models along with model diversity. The individual panels show model spread of global annual averages for each of the considered lifecycle and optics variables (x-axis) and for each model (different colors). The y-axis corresponds to the percentage bias from the ensemble median. Also plotted are the model spread (gray shaded area, IQR) as well as the numerical values of median and IQR (in gray colors at the bottom of each subplot; values correspond to Table 3 but may differ due to rounding errors). Note that some models reported erroneous BC MECs and ODs which are not included here (for details see Tab. 3).

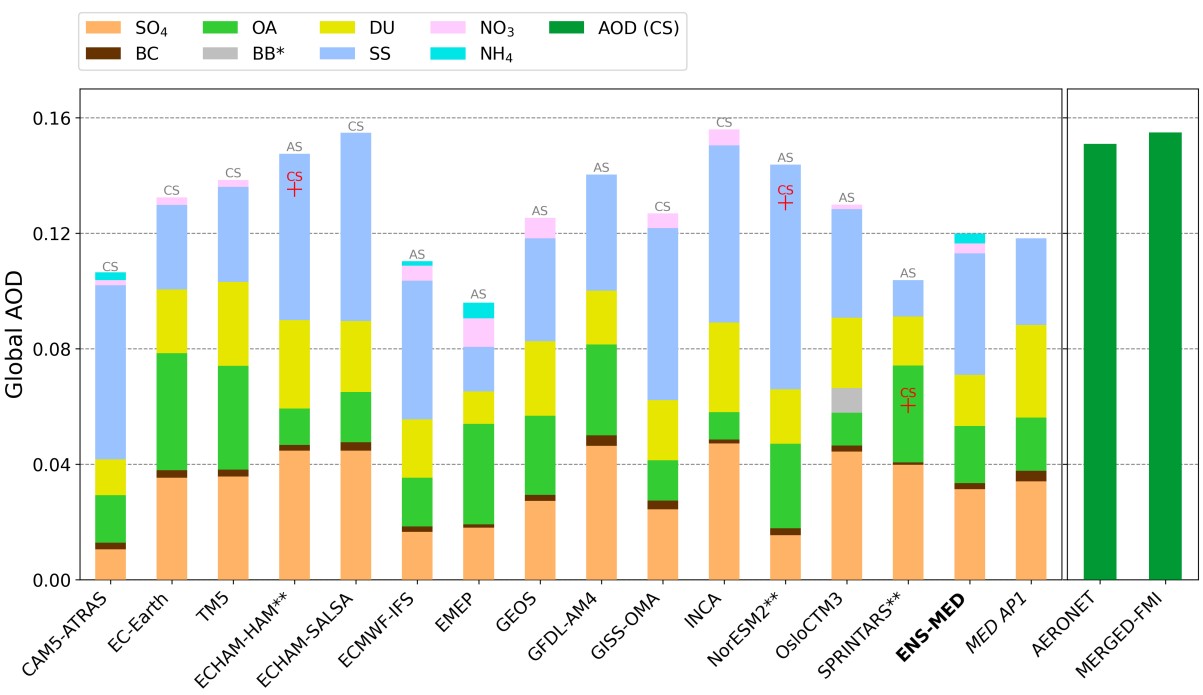

**Figure 3.** Species contributions to total AOD for each model (annual global average). The type of total AOD (AS or CS) is indicated at the top of each bar. BB* denotes biomass burning OD in OsloCTM3. Models with ** in their name submitted speciated ODs for AS conditions and total AOD for both CS and AS. The corresponding CS AODs are indicated in red with a + symbol. Also shown are estimates of total global CS AOD from AERONET and MERGED-FMI (see main text for details), similar to Fig. 3 in Kinne et al. (2006).

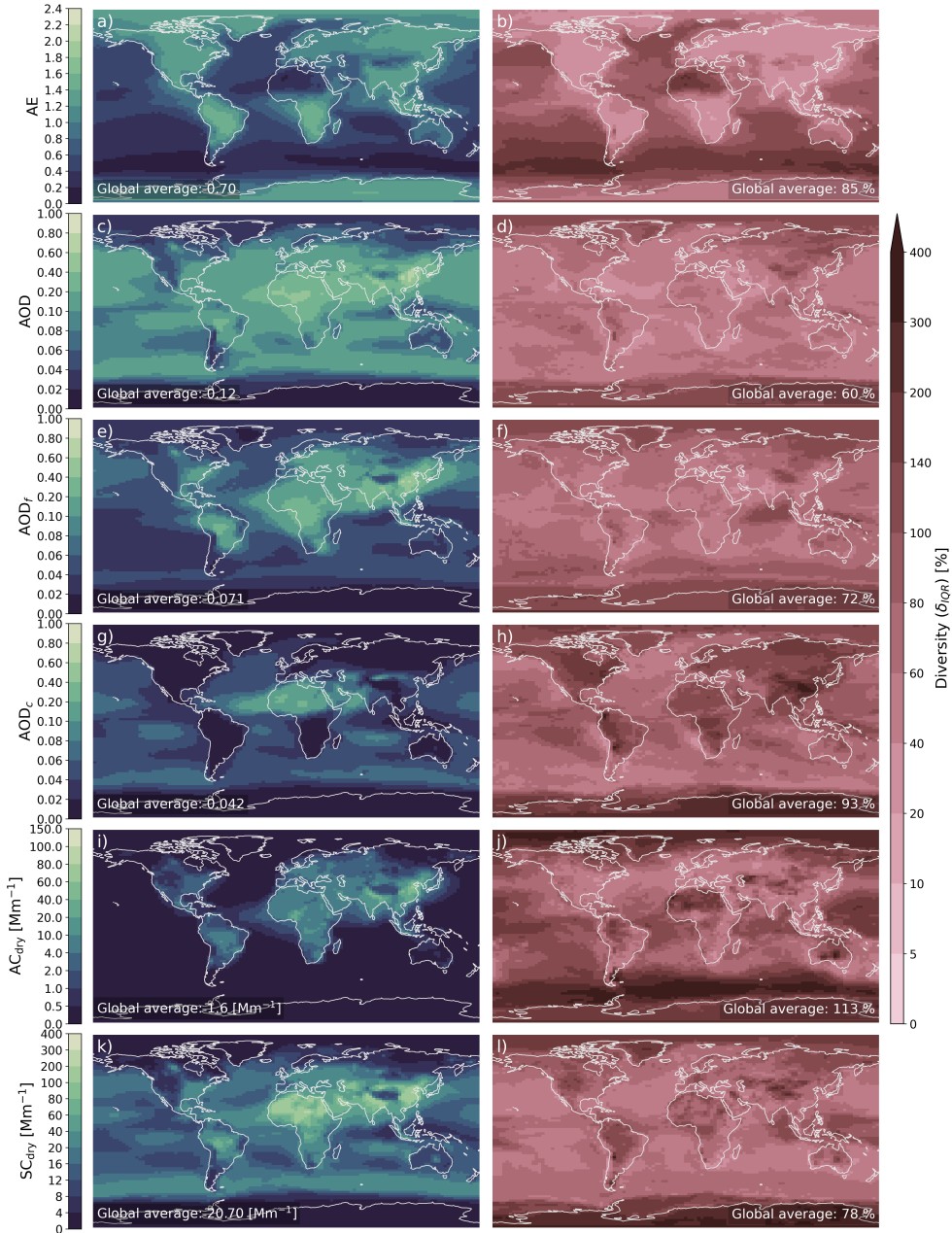

**Figure 4. Left**: Maps showing yearly averages of optical property variables from the AeroCom ensemble median (ENS-MED). The number in the lower left corner of each map represents the yearly global average values from the ensemble. **Right**: Corresponding diversity fields ($\delta_{\text{IQR}}$) for each variable, including global average diversity in lower right corner.

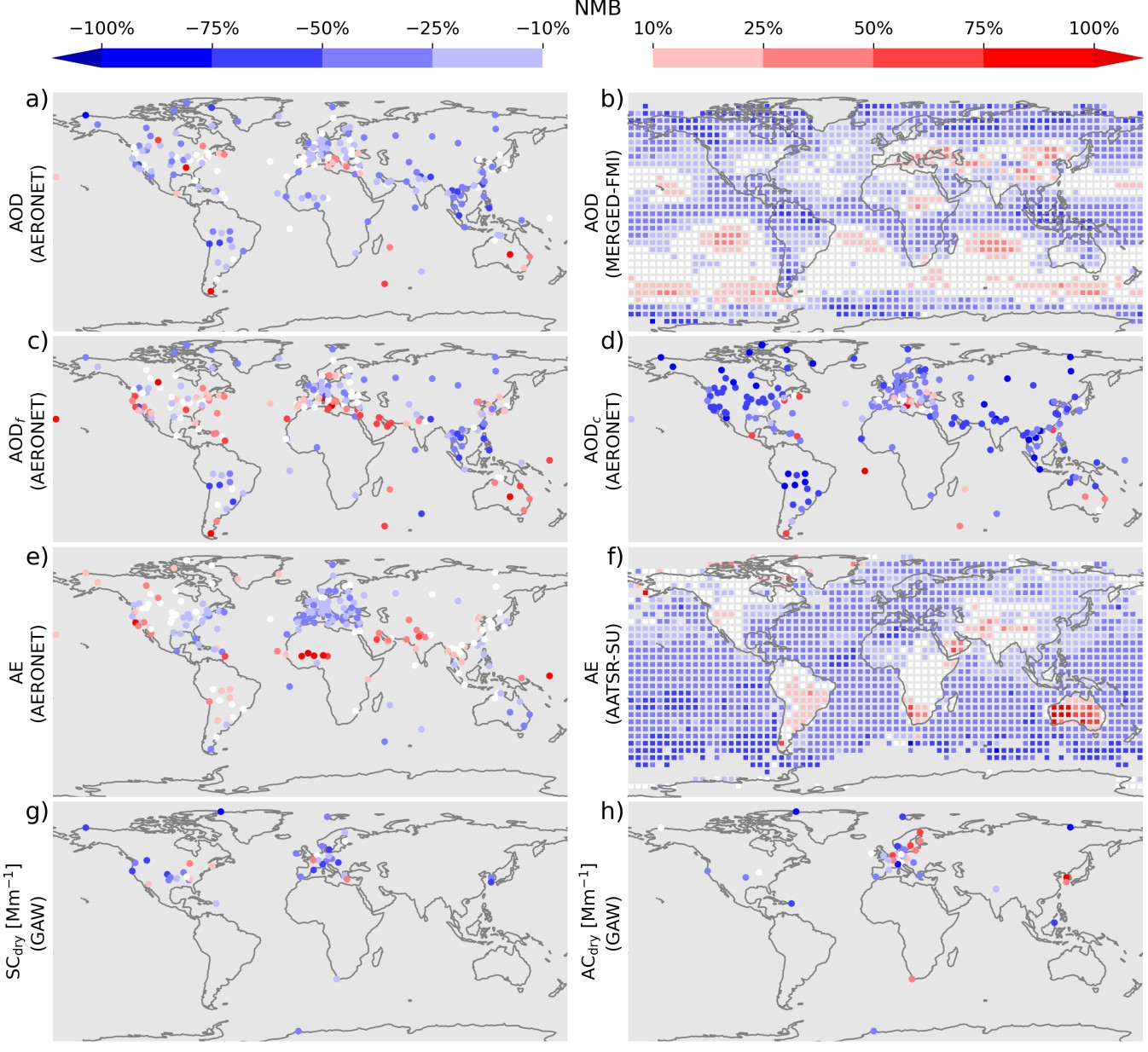

**Figure 5.** Yearly average NMBs of the ensemble median AOD against several observation records. **1st row:** AODs from AERONET and merged merged satellite data set. **2nd row:** fine and coarse AODs from AERONET. **3rd row:** AE from AERONET and AATSR-SU satellite data set. **4th row:** surface dry scattering and absorption coefficients from GAW in situ sites.

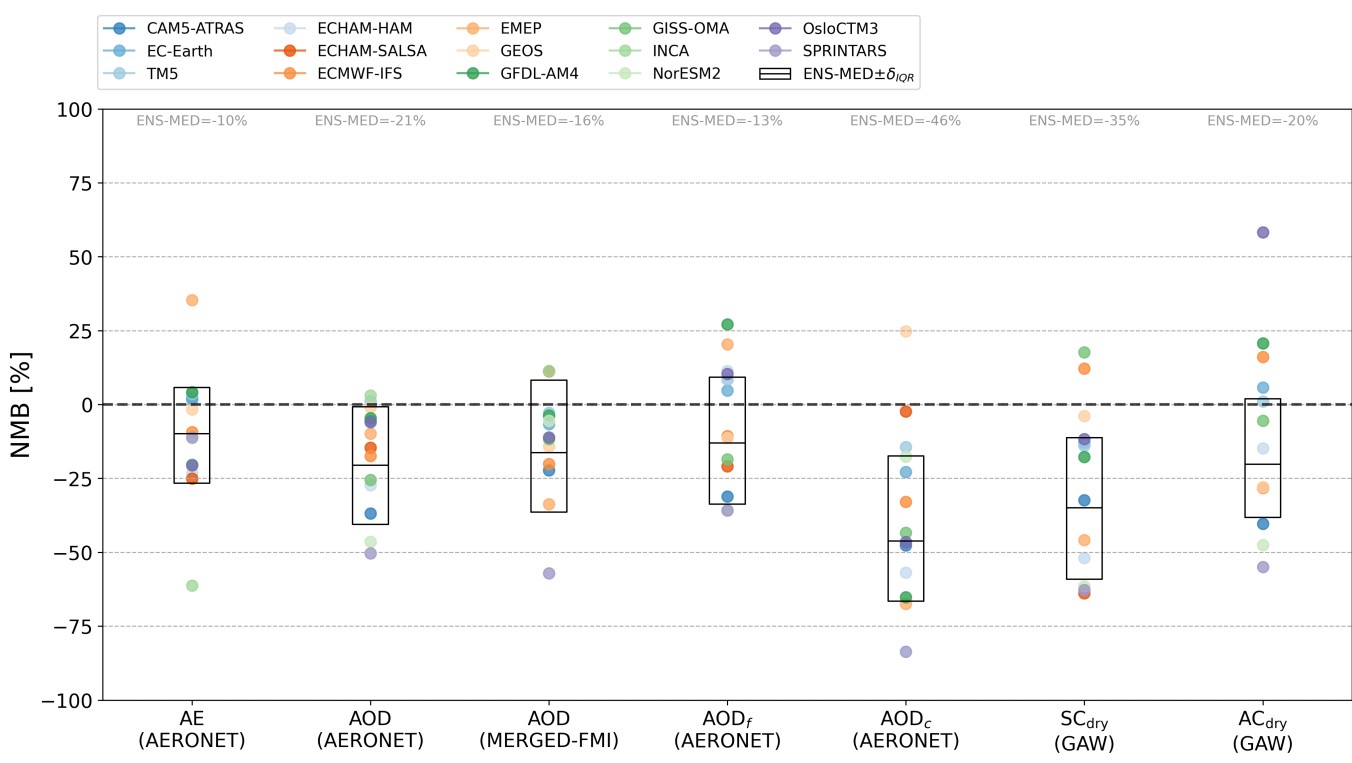

**Figure 6.** Summary of results from comparison of models with ground-based observation networks and MERGED-FMI satellite AOD data set. The y-axis indicates the retrieved biases (NMBs) for individual models (indicated as circles). The black boxes indicate results from the ensemble median (ENS-MED), together with the associated spread ($\delta_{\text{IQR}}$).

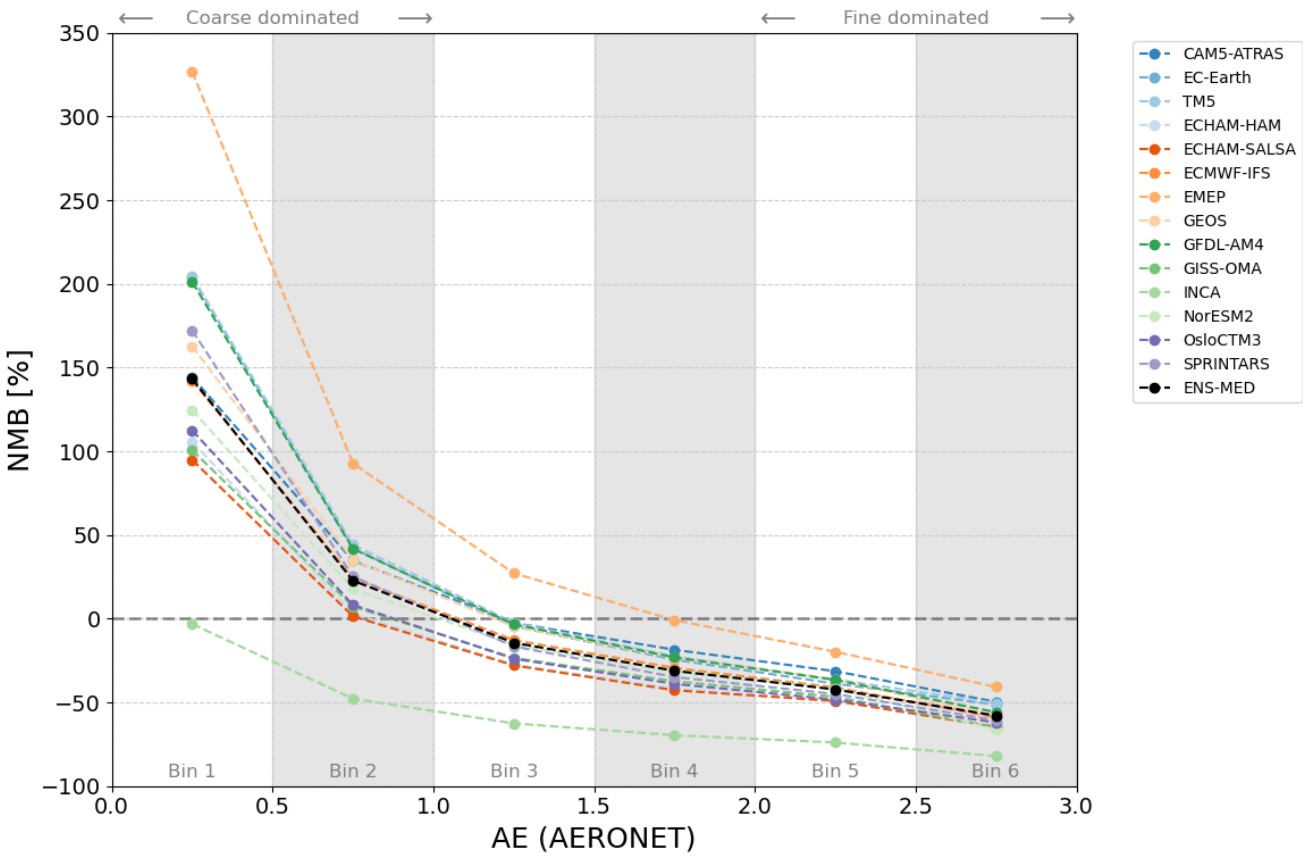

**Figure 7.** AE model biases in different AE regimes. The biases for each model (y-axis) are retrieved by co-location with AERONET observations, using only measurements that fall into the respective AE bin.