# Peer review of "AeroCom phase III multi-model evaluation of the aerosol lifecycle and optical properties using ground and space based remote sensing as well as surface in situ observations"

_Atmospheric Chemistry and Physics, 2019_

## Referee Comment (RC1) · Anonymous Referee #1 · 15 Apr 2020

Review report of acp-2019-2014 manuscript

The submitted work deals with the intercomparison of models contributing to the AEROCOM initiative as well as to the evaluation of key simulated aerosol optical properties against corresponding measurements provided by ground-based networks and satellite sensors. It is clear that the topic fits well to the scientific purposes of ACP.

[Figure]

Nevertheless, after reading carefully the text I have the feeling that it looks more as a technical report rather than a scientific paper. My concern is that there is a "disparity" between the number of figures (including also Appendices) and the discussion (interpretation) of the outcomes. Moreover, it is needed a reconstruction of the structure in order to facilitate the reader to understand the tools, the methods and the findings. Summarizing, the submitted paper can be published after taken into account the comments listed below.

1. The abstract is too long providing a lot of numbers. It is better to reduce it, highlighting the major findings of your work without stating in detail the metrics obtained from the evaluation/intercomparison analysis. 2. Section 2: I cannot understand why you have to discuss your results here. It is more straightforward to move them in a sub-section of the relevant part of the manuscript (i.e., Results). Also, consider renaming Methods to Observations and models (or Data). 3. Section 2.2: Introduce here all the models used in your analysis. 4. Section 2.2 must be improved. Please consider rewriting both paragraphs. 5. I would suggest changing the title in Section 2.4. Please move this part to Results. Also, in this section (as well as in many parts of text) the interpretation is poor containing just statements from the metrics. 6. Why is useful for your analysis the evaluation of the satellite products since their reliability has been assessed in depth in previous relevant studies? 7. In general, it is missing the intercomparison (connection) of your results with those reported in Kinne et al. (2006). 8. Line 132: Provide wavelengths for AERONET AOD and Angstrom. 9. Line 173: Why the AAE is universally constant and not aerosol-type dependent? 10. Lines 177-179: Please be more specific on how the corrections of PAMB and TAMB are applied. 11. Lines 184-189: Provide a short description and interpretation of the obtained findings, both for scattering and absorption coefficients. 12. Please put more effort on explaining the results for the absorption coefficient. 13. Figure 2: Could you please explain how the discrimination of fine and coarse AOD has been done for MODIS? Why for MODIS-Terra there are results for fine/coarse AODs and not for MODIS-Aqua? 14. Section 4: See my comment 3. Present the results based on the considered parameters instead

of separately for each model. 15. Lines 560-561: Clarify better this sentence.

---

## Referee Comment (RC2) · Anonymous Referee #2 · 16 Apr 2020

General remarks:

The present manuscript presents the results of the annual evaluation (for the year 2010) of the aerosol optical properties of 14 global aerosol models participating in the AeroCom Phase III Control Experiment. The observational products used in this exercise include in-situ observations (AERONET and GAW) and satellite retrievals (AATR-

[Figure]

SU, MODIS and MERGED-FMI). It is obvious the significant effort that the authors are doing for summarising all the results. However, the large amount of results related to the assessment of the observations and the ensemble and individual model evaluation results makes it difficult to follow all the discussions.

While the results of the study are interesting to be published, their presentation and discussion are not yet sufficient enough to be published at Atmospheric Chemistry and Physics in the current form. Consider being publishing after addressing revisions which are explained below.

General comments:

As a general comment, I would like to emphasise the effort of the authors for synthesizing all the information in this manuscript. However, a large number of figures, tables and supplementary material can introduce some confusion to the reader. You refer Tables and Figures that at the same time refer other Tables and Figures from the Appendix (see, for example, Table 3).

Also, it is hard to can get clear conclusions of the comparison because of the mixture of different models and variables. I mean the use of some models for the analysis of the representativity (Section 2.4) are not considered in the AeroCom ensemble (see Table A1) or some models that are providing AOD and others, AODclear-sky for the comparison. As it is shown in Table A1, there are some models (as INCA and ECMWF-IFS) are not considered in the AeroCom ensemble. Is there any advantage to keeping some models outside the AeroCom ensemble?

Is there any reason for selecting the year 2010 for the comparison? Maybe, the observation's availability or the emissions considered in the modelling simulations? You should include some words in the manuscript.

It is value the effort that the authors include for the assessment and representativity of the different observational datasets used in the comparison. As it is indicated in

Table 1, authors are considering different time-frequency for the various databases. I understand that all the observational datasets are converted to monthly averages for the comparison with the monthly averages of the model. Could you include information about the delivered output frequency of the model?

Moreover, you mention that you are computing the AeroCom mean and median at 2° x 3° and use the raw resolution of the rest of the models for the AERONET and GAW comparison. Meanwhile, you regrid all the models to 5° x 5° for the satellite comparison that also you compare with AERONET. Are there the results of AeroCom mean/median consistent in both comparisons?

Also, the observational aerosol products that you are considering are only available during the daytime, can you quantify this uncertainty? Figures A3-A5 should support this discussion because you are comparing the impact of considering 3hourly (Figure A3) and hourly (Figure A4) basis vs monthly, which is 14% and 8% respectively. However, it is difficult to understand the impact when you are comparing different models and different variables.

From the satellite comparison, you are considering for different datasets MODIS-Aqua, MODIS-Terra, AATSR-SU and MERGE-FMI, is there is a recommendation that you can provide in the manuscript about the most reliable for model evaluation purposes?

To help the reader, I would move the results of the individual models (Sect. 4) to a Supplement. I would keep the most important findings related to the multi-model comparison in the main discussion. Considering that you are including different models, it would be expected to find more discussion about possible improvements to have into consideration for the model community as aerosol emissions, size distribution, hygroscopicity or aerosol optical properties used.

In Section 5, it is where you introduce the results considering AOD clear-sky from model experiments. Is there any difference in the comparison with satellites between those models that delivered AOD or AODclear-sky? Finally, please, revise the references to figures, tables and sections there is a mixture of formats.

Minor comments:

Page 2 Line 16: Capital letters in Aqua and Terra, i.e., MODIS-Aqua and MODIS-Terra.

Page 2 Line 21: Capital letters in Terra, i.e., MODIS-Terra.

Page 3 Lines 51-55: Add a reference as Boucher, O., Randall, D., Artaxo, P., Brether-ton, C., Feingold, G., Forster, P., ... & Rasch, P. (2013). Clouds and aerosols. In Climate change 2013: the physical science basis. Contribution of Working Group I to the Fifth Assessment Report of the Intergovernmental Panel on Climate Change (pp. 571-657). Cambridge University Press.

Page 3 Lines 58-59: Add a reference.

Page 3 Lines 64-65: Add a reference related to the quantification of DMS.

Page 4 Line 83: Introduce the GAW acronym.

Page 4 Line 102: What are the advances between the set of models used in Kinne et al. (2006) and the ones considered in the present study? Is there any common/different feature between both studies? Is there any major improvement in the optical properties calculation from the modelling side?

Page 4 Line 103: Remove "aerosol optical depth", it is already introduced.

Page 5 Line 121: Introduce MEC and OD.

Page 5 Line 137: In Figure 1 in the AATS-SU AE map, it is observed fine aerosols (high AE values, > 1) in Antarctica, could you add a comment on it?

Page 6 Section 2.1.1: Consider to mention that AERONET SDA products (AOD coarse and AOD fine) are provided at 500nm. Also, you should include the number of sites used for the comparison in Table 1 and do reference to the location in Figure 1, for example.

Page 6 Line 163: Introduce STP.

Pages 6-7 Section 2.2.2: You should include the number of the final selection of sites used for the comparison shown in Table 1, instead of mentioning the ones excluded and do reference of the location in Figure 1, for example.

Page 8 Lines 216: What Level are you considering in the study? I suppose that it is Level 3, but it is better to mention again here.

Page 8 Section 2.1.5: It should be mentioned that MODIS and AATS products are considered inside this MERGED-FMI.

Page 8 Line 234: Table is in capital letters.

Page 9 Line 255: Indicates that this is Appendix C

Page 9 Line 265: Specify the reference to cf. 1.

Page 10 Lines 296-298: In Section 2.4, for the representativity of the results you are combining different models and observational datasets for concluding that "the overall difference is of the order of 10% and 0.2 for NMB and correlation, respectively". This is supposed shown in Table 3, but the numbers are not coincident.

Page 11 Section 2.5: Because you mention the bias to Europe and North America because of the density of sites. Is there any regional results in the comparison satellite vs AERONET that you can consider to include?

Page 12 Line 341: Add Figure 3.

Page 13 Line 370: Missing ".".

Page 13 Line 379: Extra space.

Table 2: Add a reference to the supplementary material (the excel table).

Figure 2: Replace the continuous colour palettes for a new one with categories as in Figure 1. It is easier for the reader to associate each colour to the corresponding

category.

Figures 4-8: Add the corresponding legend associated with the colours. Another possibility is to keep the numbers and use the colour scale to indicate what models are above/under the AeroCom median/mean which is the reference.

Figures 10-12: Replace the continuous colour palettes for a new one with categories as in Figure 9. It is easier for the reader to associate each colour to the corresponding category.

Figures A1 and A2: The sites in the heatmap are organised by alphabetical order, to can distinguish a pattern, maybe it would be better to create clusters per continent, latitude, longitude. Replace the continuous colour palettes for a new one with categories as in Figure 9. It is easier for the reader to associate each colour to the corresponding category.

---

## Author Comment (AC1) · 11 Sep 2020

The attached document contains a short letter to the editor as well as our responses to the comments from both reviewers. In addition, it provides information about updates in the model and observation data used (compared to the initial submission) as well

as a summary of the major changes that have been applied in the revised manuscript, in response to the comments from both reviewers.

Please also note the supplement to this comment:
https://acp.copernicus.org/preprints/acp-2019-1214/acp-2019-1214-AC1-supplement.pdf

―――――――――――――――――――――

[Figure]

**Supplement:**

**Author's response to revisions of acp-2019-1214 (Gliss et al. AeroCom study)**

Dear Nikolaos Mihalopoulos,

We would like to thank the reviewers for their constructive and helpful comments which we address below. We would also like to thank you for granting the required extensions, which enabled us to address all reviewer comments in great detail.

We followed most of the reviewers' suggestions, which resulted in major revisions and restructuring of the document. As a result of the revisions, we slightly modified the title of the manuscript. We changed the title from:

*Multi-model evaluation of aerosol optical properties in the AeroCom phase III Control experiment, using ground and space-based columnar observations from AERONET, MODIS, AATSR and a merged satellite product as well as surface in situ observations from GAW sites*

To:

*AeroCom phase III multi-model evaluation of the aerosol lifecycle and optical properties using ground and space based remote sensing as well as surface in situ observations*

We are convinced that we were able to address all comments and we believe that the revised manuscript is now adequate for publication in ACP.

Below we answer the comments of both reviewers. Before that, we summarise major changes that have been made in the revised manuscript.

Please also note that we did not submit a *diff* file as requested by ACP, as we believe that such a file would not help in this case, given the considerable changes applied with most paragraphs rewritten. We hope that you and the reviewers see this in the same way. Please let us know if you want us to provide such a diff file regardless.

Kind regards,
The authors

**Changes in data used**

Before addressing the comments from the reviewers below, here we first summarize all changes and updates that have been applied to the model and observational data used in the paper (including bug fixes in the analysis that affect the results).

**1. Changes in observation data used**

- EBAS dataset was updated (this affects the GAW in situ results, however, differences between both versions were investigated and had minor impact on our results).
- AERONET was updated (also with minor impacts on results).
- MODIS was updated now using combined DT and DB products. This was a recommendation from R. Levy through private communication. It resulted only in minor changes when evaluating the models with the updated MODIS satellite dataset, with slight improvements of NMB of ca. +1% and R of ca. 0.03 - 0.05 (when colocating in 5x5 degree resolution, based on monthly statistics).
- All satellite AOD products: due to increased uncertainties in satellite products, only measurements showing AOD>0.02 were considered. Changes in global NMB results (models vs. satellites) are minor (less than 1% difference in NMB).
- AATSR4.3-SU: due to larger uncertainties in satellite retrievals of AE, only AE values were used where 0.05 < AOD < 1.5. This decision is based on recommendations by P. North through private communication.

**2. Changes in model data used**

- **ECHAM-HAM:**
    - Deposition rate diagnostics were corrected (with impacts on lifetime estimates).
    - Resubmitted speciated optical depths at ambient conditions, as required by AeroCom (before, species optical depths represented dry aerosol).

- **ECHAM-SALSA:** resubmitted speciated ODs at ambient conditions, as required by AeroCom (before, species optical depths represented dry aerosol).

- **ECMWF-IFS:** was updated to more recent version 46r1 (before 45r1 was used), now with a complete set of required variables (fine / coarse AOD, surface dry scat. and abs. coeffs.). Differences can be seen in the online web visualization of the results: https://aerocom-evaluation.met.no/overall.php?project=aerocom&exp=PIII-optics2019-P

- **SPRINTARS:** was resubmitted due to an error in the calculation of the Angstrom exponent (AE). This mostly affected results in AE (large improvement) and other investigated variables to a minor degree.

- **GISS-OMA:** in the initial submission version, the fSST (i.e., no nudging applied) run was used as a temporary solution since CTRL was erroneous before submission in Dec 2019. The latter was corrected and resubmitted and is now used instead of fSST. Changes in bias between both versions are around 5-15% but signs of biases are mostly preserved (only surf. scattering seems to be shifted towards more positive bias). The results for both versions are also available online (see link above).

- **CAM5-ATRAS:** was updated since biomass burning emissions were incorrect (they were counted twice).

- **OsloCTM:** Updated dry diagnostics for EBAS evaluation, with improvement in dry absorption and dry scattering bias.

- **GFDL-AM4:** Updated results of coarse AOD. Accidentally, in the previous version of the manuscript, coarse AOD was computed as the sum of seasalt and dust optical depth, rather than the difference between total AOD and fine mode AOD (this mistake was due to an old setting from a preliminary analysis configuration file). It has been corrected in the revised version of the manuscript. Normalized mean bias vs. AERONET SDA changed from -24% (when using *AOD>1um=SS+DUST*) to -65% (when using *AOD>1um=AOD-AOD<1um*).

- **AeroCom ensemble MEAN and MEDIAN fields:** were updated accordingly and now include all 14 models (note that INCA is missing fine and coarse AOD and surface dry scat. and abs. data, so this is not included in the corresponding MEAN and MEDIAN fields).

**Relevant changes in the analysis code**

- A bug was fixed in the analysis software pyaerocom, which was related to merging of overlapping time-series data. Overlapping time-series data can happen in the EBAS data files used, due to possible resubmissions of the (same) data from the data providers to the EBAS database. This bug only affects the results from the surface in situ comparisons of dry scattering and absorption data at GAW sites, accessed through the EBAS database. Changes in model NMBs and bias at GAW sites changes by ca 0% - +2% for scattering and +3% - +8% for absorption.

All these updates were incorporated and all affected results and figures/tables were recomputed. We remark that none of these changes had major impacts on the results, with respect to our discussion and interpretation.

**General remarks from the authors based on comments from both reviewers**

We would like to thank the reviewers for their thoughtful comments and very good suggestions. Both reviewers found major flaws in the structure of the paper and in the way the results were discussed and interpreted. Both reviewers thus, suggested major improvements related to the structure and interpretation of the results. We agree with these suggestions and as a result, we have reorganized the structure of the paper substantially and also the discussion and interpretation of our results. We are convinced that this will help the reader to understand our results and their implications in a wider context of the associated literature.

The applied changes directly or indirectly affect / clarify most of the individual comments from the reviewers below. Therefore, we summarise the major updates here and will refer to them when needed in our individual answers to the reviewers below. We will mark our responses and relevant changes in red color.

Major changes in the manuscript between initial submission (ACPD version) and the revised manuscript:

- **Appendices:** We removed the Appendix sections completely and incorporated the content either in the main manuscript, or in the new supplemental material 2. We also removed unnecessary/unused Figures from the Appendix (i.e., scatter plots from sensitivity studies, site bias heatmaps for each model and GAW site for surface scattering and absorption data). Interpretation of model biases at individual sites is certainly of interest but beyond the scope of this overview paper focussing on the global modelling of the aerosol.

- In this context, Table A1 (which contained some more details about models) was removed and relevant information was included in the updated model overview table (Table 2 in the revised manuscript).

- **Section 2:** was renamed from "Methods" to "Data and Methods". Most important changes in subsections:

  - **Observations used (Sect. 2.1):** individual paragraphs of AERONET, GAW in situ data and used satellite datasets have been revised and improved. An introduction paragraph for the used satellite data was added.

  - **Model introduction (Sect. 2.2):** We improved the introduction of the individual models and relevant information is now summarised in the revised model overview table 2 in the manuscript. Section 2 now also includes a short introduction of the content of the supplementary material 1 (questionnaire about model assumptions related to optical properties

and aerosol lifecycle). Detailed introductions of individual models have been separated from the discussion of their results (i.e., they were formerly included in the discussion Section 4) and have been moved into the new supplement 2. Details related to the computation of ensemble MEAN and MEDIAN fields have been moved into the new Sect. 2.3.1.

- ○ **Sect. 2.3 "Data processing and statistics" (formerly "Data analysis"):** As mentioned above, this section now includes the details related to the ensemble model (Sect. 2.3.1). Also the discussion of the applied STP correction for the in situ data was moved from the introduction section of these data into a new subsection 2.3.2 "Model STP correction for comparison with GAW in situ data".

- ○ **Results from spatiotemporal representativity studies (formerly Sect. 2.4):** the results from several sensitivity studies were before included in Sect. 2.4 and were misplaced there. They were thus moved to the end of the (revised) Section 4 which discusses results from the model comparison with observations (details related to changes in Sect. 4 are provided below).

- ○ **Results from satellite evaluation (formerly Sect. 2.5):** We removed the discussion of results from the satellite vs. AERONET intercomparison in (e.g. former Sect. 2.5) into the new supplement 2. This was done because this discussion distracted from the main purpose of the paper, which is model evaluation. Relevant satellite biases compared to AERONET are mentioned in the introduction of the satellite datasets used (in Section 2.1.3f) and are included in Table 1. Associated uncertainties of satellite retrievals that may impact the interpretation of the results from the model assessments are discussed where needed, in the new result and discussion section 4.

- ● **Presentation and discussion of results (Sect. 3&4):** The presentation and discussion of the results was presented before in "*Sect. 3 Results*" and "*Sect. 4 Discussion of results from individual models*". As indicated by both reviewers this organisation of the content was not optimal. Thus, large efforts have been made to improve this, resulting in major restructuring of these sections. In the revised manuscript, results and discussion are organised in 2 sections:

  - ○ **Sect. 3 Results - Model diversity**: this section focuses on presentation and discussion of inter-model diversity related to aerosol lifecycle and optical properties, on a global scale (this section does not include any comparison with observations). This includes much of the content of the previous Sect. 3 (i.e. modelled diversities in emissions, lifetimes, burdens, AODs, MECs) but in addition now also includes a comparison with results

from AeroCom phase I (e.g., Textor et al., 2006, Kinne et al., 2006).

- ○ **Sect. 4 Results - Optical properties evaluation**: this section presents and discusses the results from the optical properties evaluation. As suggested by reviewer 1, the discussion of the results is now categorised by the individual optical parameters that were evaluated, rather than by the individual models (as was done before).

- ● **Figures and tables:** as a result of the above discussed updates and in order to improve the interpretability of our results, most figures and tables have been updated. As a result, we drastically reduced the number of Figures in the main manuscript from before 12 to 7 in the revised manuscript. Here we list the most important changes:

  - ○ We removed results from the AeroCom ensemble MEAN from all affected figures (former Figs. 3-7 and 10-12), since we focus our discussion on the MEDIAN results in the revised manuscript.

  - ○ **Removed former Figure 2.** The figure showed satellite biases and correlations compared to AERONET and was related to the former Sect. 2.5 which has been moved in the supplement (see also comment above).

  - ○ **Figures related to aerosol lifecycle and optical properties:** Former Figures 3-7 (blue colored heatmaps of speciated emissions, lifetimes, burdens, MECs and ODs) have been merged into one table (Table 3 in the revised manuscript). Also, MAC of BC, OA and DU was added to that table for the models that provided the required diagnostics. Emissions of DMS and $SO_2$ were removed and, instead, the total source strength of $SO_4$ and OA is provided now which is derived based on the reported total deposition rates. The color coding was changed to illustrate the deviation from the ensemble median. In addition, results from associated AeroCom phase I (AP1) studies (Kinne et al., Textor et al., 2006) were added and are discussed in the new Section 3. A better comparison with older AeroCom studies was requested by both reviewers and we agree that this adds substantially to the quality of the paper by putting the new results into historical context. The outcomes of this intercomparison are discussed accordingly in the new Sect. 3.

  - ○ **New Figure 2:** In that context, to better investigate intro-model diversity, a new Figure was created that connects diversities (and deviations from the AP3 ensemble median) in emissions, lifetimes, burdens, MECs and resulting optical depths for each species.

  - ○ **New Fig. 3:** Another new Figure illustrates the simulated total AODs for each model as a stacked bar-chart of its component optical depths. It

further includes the median composition from Kinne et al., 2006 and global estimates from AERONET and the merged satellite AOD product, as well as information related to which models provided clear-sky or all-sky optics. This Figure helps the reader to visualize the main changes in AOD composition since AP1.

- **Figure 4 - global maps of annual averages and diversities from the ensemble model (formerly Fig. 8)**: We applied a discrete colour mapping (before a continuous mapping was used) as suggested by reviewer 2. We also removed displaying of the corresponding observation sites (and the associate info in the legends) as these are provided in Fig. 1 and were out of context in this figure which focuses on the intra-model diversity and not on the comparison with observations.

- **Figure 5 - bias maps of ensemble model compared to some observations used (formerly Fig. 9)**: this Figure now shows 8 maps of biases of ENS-MED compared to AOD (AERONET, merged satellite), AODf and AODc (AERONET), AE (AERONET, AATSR-SU) and in situ scattering and absorption. Before only AOD and surf. scattering was shown in a single map. This way it is easier to interpret and separate the results.

- **New Table 4 - Model biases and correlation coefficients compared to the various observations used (formerly Figs.10&11)**: the 2 figures were merged into one table and also here a discrete color mapping is applied now for NMB and Pearson R. It also includes results from ensemble fields of the 25th (Q1) and 75th (Q3) percentile, in order to illustrate the intra-model variability in biases.

- **Figure 6 - summary of results from comparison with observations (formerly Fig. 12)**: added AOD results from merged satellite product (before only AERONET was shown). Also, the color coding was changed: before the colours indicated the correlation coefficients, now they indicate the individual models. Also, the ensemble median results and corresponding interquartile range are now plotted as boxplots.

- **New Figure 7:** this new figure shows model biases in the Angstrom Exponent (AE) compared to AERONET in different AE bins (e.g. 0-0.5, 0.5-1, …, 2.5-3). This is done to visualize differences in model bias between fine and coarse-mode dominated aerosol measurements.

- **Table 5 - results from spatiotemporal representativity studies (formerly Table 3):** The table shows differences in bias and correlation retrieved for the individual tests performed (e.g., results from low vs high resolution experiments). Formerly, only relative differences in biases and

correlations were visualised, now the actual biases and retrieved correlations are reported for each test case and resolution. In addition, for the temporal representativity tests, differences in bias and correlation are presented for model / obs comparison that (1) use monthly averages computed with 25% coverage constraint for temporal resampling (e.g., at least 7 daily values per month) as done in the paper, compared to (2) results that do not require any coverage constraint (i.e., a single daily value is enough to retrieve a monthly "mean"). These results are discussed in the representativity section which was moved from Sect. 2.4 into Sect. 4.6 in the revised manuscript.

- **Former table A1:** Was removed as discussed above.
- **Former table A2 - Sensitivity of model / satellite comparisons to the choice of resolution:** This table was recomputed and was moved from the Appendix into Sect. 3 in the new supplement 2. The associated text was revised. Also, the horizontal resolution of the individual models was added.

- To summarise, this is the new order of tables and figures in the revised manuscript:
  - Table 1: Observations and variables.
  - Table 2: Model overview.
  - Table 3 (coloured): Per-species emissions, lifetimes, burdens, ODs, MECs and MACs for each model as well as AP3 and AP1 ensemble median and diversities (formerly Figs. 3-7).
  - Table 4 (coloured): Results from model evaluations (biases and correlation coefficient) compared to the various observation records (formerly Figs. 10 & 11).
  - Table 5: results from spatiotemporal sensitivity analysis.
  - Figure 1: Maps of annual averages from selected ground and space based observations.
  - Figure 2: Diversity in aerosol lifecycle parameters.
  - Figure 3: AOD composition (stacked bar chart)
  - Figure 4: Maps of annual averages from ensemble median model and diversities.
  - Figure 5: Maps of model biases compared to selected observations (same observations as used in Fig. 1).
  - Figure 6: Summary of model biases compared to selected observations (box plot, models in different colors).
  - Figure 7: Model biases in AE as a function of the considered AE range.

We believe that the new structure and the updated / new visualization will help the reader to better understand and link the results.

**As mentioned above, please note that, as a result of the changes, we slightly changed the title of the paper from:**

*Multi-model evaluation of aerosol optical properties in the AeroCom phase III Control experiment, using ground and space based columnar observations from AERONET, MODIS, AATSR and a merged satellite product as well as surface in situ observations from GAW sites*

To:

*AeroCom phase III multi-model evaluation of the aerosol lifecycle and optical properties using ground and space based remote sensing as well as surface in situ observations*

Below we answer the individual comments by both reviewers. Where appropriate, we refer to the substantial changes that were summarised in this section. We note that in most cases it is not possible to provide explicitly the applied "**Changes to the manuscript**" due to the major changes summarised here. We hope that the reviewers and the editor understand that.

**Author's responses to the comments from 2 reviewers**

**Comments from Reviewer #1**

Review report of acp-2019-2014 manuscript

The submitted work deals with the intercomparison of models contributing to the AEROCOM initiative as well as to the evaluation of key simulated aerosol optical properties against corresponding measurements provided by ground-based networks and satellite sensors. It is clear that the topic fits well to the scientific purposes of ACP.

Nevertheless, after reading carefully the text I have the feeling that it looks **more as a technical report rather than a scientific paper. My concern is that there is a "disparity" between the number of figures (including also Appendices) and the discussion (interpretation) of the outcomes.** Moreover, it **is needed a reconstruction of the structure** in order to facilitate the reader to understand the tools, the methods and the findings.

We have emphasized the scientific dimension of the paper by applying the changes summarised in the general remark above. We are convinced that the updated / new visualisations will help the reader to better understand and link the results in a deeper scientific context.

Summarizing, the submitted paper can be published after taken into account the comments listed below.

1. The abstract is too long providing a lot of numbers. It is better to reduce it, highlighting the major findings of your work without stating in detail the metrics obtained from the evaluation/intercomparison analysis.

The abstract was revised as a result of the better presentation and interpretation of our results. It has also been shortened.

2. Section 2: I cannot understand why you have to discuss your results here. It is more straightforward to move them in a subsection of the relevant part of the manuscript (i.e., Results). Also, consider renaming Methods to Observations and models (or Data).

We agree with the reviewer. This issue has been resolved in the revised version of the manuscript, as a result of the major changes summarised above.

3. Section 2.2: Introduce here all the models used in your analysis.

This issue has been resolved in the revised version of the manuscript, as a result of the major changes summarised above.

4. Section 2.2 must be improved. Please consider rewriting both paragraphs.

We agree that the former model introduction section was not sufficient. As discussed above, the section was revised and information relevant to the paper was added to the model overview table. Furthermore, references and short summaries to additional in-depth information about the models are provided in the new Sect. 2.2.

5. I would suggest changing the title in Section 2.4. Please move this part to Results. Also, in this section (as well as in many parts of text) the interpretation is poor containing just statements from the metrics.

As stated above, the former Section 2.4 (about representativity) is now section 4.6 in the results, however, we kept the title of this section "Representativity of the results" as it describes well what this section is about.
We improved the description, presentation and interpretation of the results from these studies. In addition, as mentioned above, considerable emphasis was placed on improving the discussion and interpretation of our results throughout the paper. In this context, we refer particularly to the revised discussion sections 3 & 4 and the revised conclusion section.

**6. Why is useful for your analysis the evaluation of the satellite products since their reliability has been assessed in depth in previous relevant studies?**

As mentioned above, the discussion of results from the satellite evaluation vs AERONET was moved to supplement 2 and where available, references were added. However, we note that in the case of AATSR SU v4.3 dataset there is not really any published literature available, so we found it important to document at least relevant relative biases between satellites and AERONET. We believe that having these metrics (bias and correlation) vs AERONET available in the paper is helpful for the reader when assessing the results from the model evaluation.

**7. In general, it is missing the inter-comparison (connection) of your results with those reported in Kinne et al. (2006).**

We agree with the reviewer. This issue has been resolved in the revised version of the manuscript, as a result of the major changes summarised above. Comparisons with results from the AeroCom Phase I studies by Kinne et al., 2006 and Textor et al., 2006 are discussed mostly in the new Sect. 3 and also in Sect. 4.

8. Line 132: Provide wavelengths for AERONET AOD and Angstrom.

Comparison wavelengths for each observation dataset and variable are now provided in a dedicated column in Table 1 in the revised version of the manuscript. Discussion of the measurement wavelengths has been expanded in the text.

9. Line 173: Why the AAE is universally constant and not aerosol-type dependent?

We are aware that AAE is certainly not universally constant and is aerosol type dependent. Unfortunately, in the measurements we don't necessarily have spectral absorption information or time varying information about aerosol type. We chose a constant value of 1 for AAE as we believe it is a justified assumption and significantly increases the number of sites which we can use here. We have performed an analysis and determined that the error for the adjustment assuming a constant AAE tends to be relatively small. We understand that this was not discussed well enough and added the following text in Sect. 2.1.2 in the revised manuscript:

**Changes to the manuscript:**
*"For the in situ AC data used in this study, most of the measurements are performed at wavelengths other than 550 nm (see sect. 1 in supplement 2). These were converted to 550 nm assuming an absorption Ångström exponent (AAE) of 1 (i.e., a 1/λ dependence, e.g., Bond and Bergstrom, 2006). This is a fairly typical assumption when the spectral absorption is not measured. For about 50 % of the sites, absorption was measured at ~530 nm meaning that even if the true AAE had a value of 2, the wavelength-adjusted AC value would only be underestimated by ca 4%. For another 25% of the sites, absorption was measured at ~670 nm. For these sites the impact of an incorrect AAE value is larger (ca 26% overestimation for an actual AAE of 2 and ca 6% for AAE=1.25). The remaining 25% of sites typically utilized wavelengths between these two values. Schmeisser et al. (2017) suggest that, across a spatially and environmentally diverse set of sites measuring spectral in situ absorption (many included here), that the AAE is typically between 1 and 1.5."*

10. Lines 177-179: Please be more specific on how the corrections of PAMB and TAMB are applied.

We have now included the assumed standard temperature and pressure values and tried to clarify the text about this adjustment. We moved the discussion into a new subsection 2.3.2 in the revised manuscript.

11.Lines 184-189: Provide a short description and interpretation of the obtained findings, both for scattering and absorption coefficients.

This refers to the bias heatmaps for each EBAS site shown in the appendix. Even though they are interesting themselves, we decided to remove these figures as the investigation of results at individual sites is beyond the scope of this comprehensive global study. (One of the co-authors is working on this in a separate paper). The figures and lines 184-189 have now been removed.

12. Please put more effort on explaining the results for the absorption coefficient.

We agree with the reviewer that the absorption related discussions could see some improvements. This issue has been resolved in the revised version of the manuscript by (1) discussing and connecting better BC lifecycle related parameters and MACs in Sect. 3 and (2) by largely extending the discussion of the results from surface absorption coefficient comparisons in Sect. 4.5 in the revised manuscript. The results are now also discussed in the context of previous studies.

13. Figure 2: Could you please explain how the discrimination of fine and coarse AOD has been done for MODIS? Why for MODIS-Terra there are results for fine/coarse AODs and not for MODIS-Aqua?

This is not relevant anymore for the revised version of the manuscript since Figure 2 is removed in the revised version and also because the fine / coarse data from MODIS was not further used in the paper (i.e. only total AOD is used from MODIS, cf. also answer to point 2 above).

14. Section 4: See my comment 3. Present the results based on the considered parameters instead of separately for each model.

We agree with the reviewer and followed this suggestion as summarised above.

15. Lines 560-561: Clarify better this sentence.

These lines were part of the former Section 4.4 (discussion of results from ECHAM-SALSA), which has been completely removed, as a result of the major restructuring of the paper (see prev. point and information above).

**Comments from reviewer #2**

General remarks:The present manuscript presents the results of the annual evaluation (for the year 2010) of the aerosol optical properties of 14 global aerosol models participating in the AeroCom Phase III Control Experiment. The observational products used in this exercise include in-situ observations (AERONET and GAW) and satellite retrievals (AATR-SU, MODIS and MERGED-FMI). It is obvious the significant effort that the authors are doing for summarising all the results. **However, the large amount of results related to the assessment of the observations and the ensemble and individual model evaluation results makes it difficult to follow all the discussions. While the results of the study are interesting to be published, their presentation and discussion are not yet sufficient enough to be published at Atmospheric Chemistry and Physics in the current form.** Consider being publishing after addressing revisions which are explained below.

We agree with the reviewer. As discussed in the summary above ("General remark from the authors based on comments from both reviewers") the paper has seen major restructuring and improvement in its discussions. We believe that these changes bring more clarity and focus into the paper.

General comments: As a general comment, I would like to emphasise the effort of the authors for synthesizing all the information in this manuscript. **However, a large number of figures, tables and supplementary material can introduce some confusion to the reader.**

We agree that the presentation of the results (and its order) was confusing in the initial submission and we believe that these issues have been resolved in the revised manuscript, as a result of the major updates summarised above.

You refer Tables and Figures that at the same time refer other Tables and Figures from the Appendix (see, for example, Table 3).

This should be resolved due to the major restructuring of the paper (see comments above) and since there is no Appendix anymore. We are convinced that the content is now clearer.

**Also, it is hard to can get clear conclusions of the comparison because of the mixture of different models and variables.** I mean the use of some models for the analysis of the representativity (Section 2.4) are not considered in the AeroCom ensemble (see Table A1)

We agree with the reviewer and refer to the major changes discussed above. We believe that the presentation and discussion of our results are much easier to follow while providing much deeper scientific interpretation at the same time.
Regarding the sensitivity analyses: we remark now clearly in Sect. 4.6 where models were used for representativity studies that are not used throughout the rest of the paper. We also discuss associated implications and assess uncertainties.

**or some models that are providing AOD and others, AOD clear-sky for the comparison**.

We substantially improved the information about which models are providing clear sky and all sky optics (see e.g., model info Table 2 in the paper or new Figure 3) and also highlighted models that only provided all sky in relevant figures. We also discuss differences in AOD between models that submitted clear sky optics and models that submitted all-sky optics.

As it is shown in Table A1, there are some models (as INCA and ECMWF-IFS) are not considered in the AeroCom ensemble. Is there any advantage to keeping some models outside the AeroCom ensemble?

We reconsidered the way the ensemble is calculated and now includes all models considered in the paper. In the initial submission, ECMWF-IFS and INCA had not submitted all diagnostics, i.e. they had not provided fine and coarse mode optical depth diagnostics or dry scattering and absorption coefficient data. Therefore they were also not included in the ensemble fields for AOD and AE, since we wanted consistency between the different variables in terms of which models contributed to the ensemble. This was changed and now all models are included in the ensemble. Please note in this context that the model version of ECMWF-IFS was updated and has now all diagnostics available. INCA is the only model that did not provide fine / coarse AOD and dry surface scattering and absorption and hence, the corresponding MEAN and MEDIAN fields do not include INCA. Differences are discussed in the new paragraph 2.1.3 that discusses the details of the ensemble composition and computation.

**Changes to the manuscript:**

The following sentence was added in Sect. 2.1.3 in the revised manuscript:

"[…]. *Please also note that the ensemble total AOD includes results from INCA which are not included in AODf and AODc (seeTab. 2). This results in a slightly smaller total AOD in the*

*ensemble when inferred from AODf+AODc (which does not include INCA) compared to the computed AOD field (which includes INCA)."*

Is there any reason for selecting the year 2010 for the comparison? Maybe, the observation's availability or the emissions considered in the modelling simulations? You should include some words in the manuscript.

The year 2010 for AeroCom Phase III CTRL was chosen by the AeroCom consortium for better comparability with older AeroCom studies (e.g. phase II) and also because many more measurements became available between 2000 (which was used in AP1) and 2010. We added the following text in Sect. 2.2 in the revised manuscript.

**Changes to the manuscript:**
*"The year 2010 was chosen as a reference year by the AeroCom consortium and is used throughout many phase II and III experiments for inter-comparability of different experiments and model generations. The AeroCom phase I simulations (e.g., Dentener et al., 2006; Kinne et al., 2006; Schulz et al., 2006;Textor et al., 2006) used the year 2000 as a reference year. One of the main reasons to update the reference year from 2000 to 2010 was that many more observations became available between 2000 and 2010 and also to account for changes in the present day climate, for instance, due to changing emissions and composition (e.g., Klimont et al., 2013; Aas et al., 2019; Mortier et al., 2020a)."*

It is value the effort that the authors include for the assessment and representativity of the different observational datasets used in the comparison. As it is indicated in Table 1, authors are considering different time-frequency for the various databases. I understand that all the observational datasets are converted to monthly averages for the comparison with the monthly averages of the model. Could you include information about the delivered output frequency of the model?

We do not think that it will add useful content for the reader to provide the originally delivered frequencies, as we consistently resample to monthly resolution. It may even cause confusion, as to which original frequencies were provided in the diagnostics before resampling to monthly. Thus, we have chosen to not provide this information in the paper. For interested users, the information is readily available in the AeroCom database and also in the "Information" tab in the online visualisation of the results, which is referred to many times in the paper (https://aerocom-evaluation.met.no/infos.php?project=aerocom&exp=PIII-optics2019-P).

**Changes to the manuscript:**

We added the following sentence in the model description Sect. 2.2 in the revised manuscript:

*"Details on the AeroCom phase III experiments can be found on the AeroCom wiki page (AeroCom wiki, 2020). The wiki also includes information on how to access the model data*

*from the different AeroCom phases and experiments, which is stored in the AeroCom database."*

Moreover, you mention that you are computing the AeroCom mean and median at 2x3 and use the raw resolution of the rest of the models for the AERONET and GAW comparison. Meanwhile, you regrid all the models to 5x5 for the satellite comparison that also you compare with AERONET. Are there the results of AeroCom mean/median consistent in both comparisons?

The choice of 2x3 for the ensemble model is due to the fact that the lowest provided model version is 2x3. We have investigated the sensitivity of that choice by comparing with a median computed at 1x1 resolution (i.e., lower resolution models were interpolated to 1x1) and the differences were marginal. The results of this comparison are online available here:

https://aerocom-evaluation.met.no/overall.php?project=aerocom&exp=PIII-optics2019-ens

Regarding the 5x5 choice for the model vs satellite comparisons: this was a compromise between spatial and temporal resolution, i.e. by spatially averaging 25 grid-points into one grid-point we increased the temporal sampling coverage such that the 25% temporal coverage constraint is met, which we require to resample from daily to monthly (which is rarely given in the original 1x1 and also not in 2x3). By regridding to 5x5 degrees instead, before temporal averaging, we found that sufficient temporal sampling coverage was given at most locations. We remark that differences in the results from the satellite / model evaluations were investigated already in the initial version of the paper and were summarised in Table A2 therein. This Table A2 was moved into the new supplement 2 (Table 3 therein) and implications are discussed in the corresponding Section 3 and are also discussed briefly in the main manuscript Sect. 4.5. We emphasize that the discussion of our results in Sect. 3 & 4 aims to address the increased uncertainties associated with the satellite data in a better way than it did in the initial submission.

Also, the observational aerosol products that you are considering are only available during the daytime, can you quantify this uncertainty? Figures A3-A5 should support this discussion because you are comparing the impact of considering 3hourly (FigureA3) and hourly (Figure A4) basis vs monthly, which is 14% and 8% respectively. However, it is difficult to understand the impact when you are comparing different models and different variables.

This is indeed a very important point and we extended the discussion in the revised sections 2.1.1 (AERONET introduction) and 4.5 (representativity of results) accordingly, following the suggestion to incorporate our findings from the high resolution tests:

**Changes to the manuscript:**
We added the following paragraph in Sect. 2.1.1 in the revised manuscript:

*"The sun photometer measurements only occur during daylight and cloud free conditions. Thus, the level 2 daily averages used here represent daytime averages rather than 24h averages (as provided by the models). Because of the requirements for sunlight and no clouds, the diurnal coverage at each site shows a more or less pronounced seasonal cycle depending on the latitude (e.g., only mid-day measurements at high latitudes in winter) and the seasonal prevalence of clouds in some regions. This is a clear limitation when comparing with 24h monthly means output from the models (as done in this study). However, these representativity issues were found to have minor impact for the model assessment methods used in this study (details are discussed in Sect. 4.5)."*

We added the following paragraph in Sect. 4.5 in the revised manuscript:

*"One further uncertainty related to the representativity of the results is that AERONET only measures during the daytime, while the models computed 24h averages (as indicated in Sect. 2.1.1). This will cause shifts in the intrinsic weighting applied when computing the network averaged statistics used throughout this paper (e.g., wintertime measurements at high latitudes are restricted to noon-time if they occur at all). In addition, it could introduce systematic errors at locations that show a persistent and pronounced diurnal profile. In this context, note that the GAW in situ observations are not affected by this as they measure continuously, night and day regardless of cloud conditions. The latter is reflected in the very similar results in Test 1 (i.e., hourly vs monthly comparison of ACdry). Since the results of test 2 (AERONET 3hourly vs monthly) show very good agreement as well, we believe that uncertainties associated with diurnal variations of AOD are likely small compared to the large uncertainties associated with the correct modelling of the AOD, reflected by the considerable biases (and their diversity) found here among the models. Furthermore, AOD represents the whole atmospheric column and, thus, should be less sensitive to diurnal variations than the near surface measurements. A detailed investigation of associated impacts of diurnal variability is desirable but beyond the scope of this paper. Also in that context, it would be interesting to investigate the extent to which global climate models need to be able to reproduce amplitudes in diurnal variability of certain tracers and physical processes and which phenomena can be sufficiently parameterised in lower temporal resolution."*

From the satellite comparison, you are considering for different datasets MODIS-Aqua, MODIS-Terra, AATSR-SU and MERGE-FMI, is there is a recommendation that you can provide in the manuscript about the most reliable for model evaluation purposes?

We believe that the merged dataset is the most complete one and in particular for long-term studies extending before 2000 it should be used. But it is not the only high-quality dataset, and also the individual datasets (which are self-consistent) are available. The analysis of the individual datasets in Sogacheva, et al., 2020 shows possible significant regional biases between them, but also shows similar regional temporal patterns. Knowing the availability and the performance of different products, which is discussed in Sogacheva, et al., 2020, a user can select a dataset covering the period under study for model evaluation or trend analysis. We intend to continue work on the merged dataset to consolidate and extend it further.

However, as we do not go into depth regarding the assessment of individual satellite datasets we are hesitant to give a recommendation in the paper.

To help the reader, I would move the results of the individual models (Sect. 4) to a Supplement.

As described above, we have restructured our results substantially and focus now in Sect. 4 on the individual parameters rather than the individual models. Relevant findings from the individual models from the former Sect. 4 were included in the revised Sections 3 & 4.

I would keep the most important findings related to the multi-model comparison in the main discussion. **Considering that you are including different models, it would be expected to find more discussion about possible improvements to have into consideration for the model community as aerosol emissions, size distribution, hygro-scopicity or aerosol optical properties used**.

Connecting with the previous comment and discussion above, we have largely improved the discussion of individual models based on our findings. However, given the large diversity in our results (see new Sect. 3), it is difficult to give general recommendations in the paper and possible improvements are likely very model specific. However, by combining the results of the aerosol lifecycle and diversity analysis (which includes comparisons of major aspects, such as significantly decreased BC burden, large shift in relative contributions of dust and sea salt with sea salt dominating the natural AOD) with the results from the optical properties evaluation (e.g. many models likely simulate too fine particles for coarse dominated aerosol) we are able to provide indications of possible areas for model improvements. However in this global overview paper it is difficult (and not the purpose) to diagnose individual aspects in the detail necessary to make recommendations on the individual model level. Nonetheless, we provide clues and indications of major flaws of individual models and give recommendations about which aspects should be investigated in more detail.

In Section 5, it is where you introduce the results considering AOD clear-sky from model experiments. Is there any difference in the comparison with satellites between those models that delivered AOD or AOD clear-sky?

In general, it was recommended to provide clear-sky (CS) diagnostics over all-sky (AS) for AOD and other columnar remote sensing variables, since the measurements should represent clear-sky conditions (also the satellites). The new Figure 3 (stacked bar chart of speciated AOD) indicates that intra-model AOD variability is mostly linked with large differences in speciation and individual component ODs rather than the choice of CS vs AS and reasons for the diversity is rather to be searched in the modelled mass and mass-to-optics conversion than the treatment of CS vs AS optics. Thus, we kept the focus on the discussion of the related aspects determining the reported AODs for each

model rather than focusing on CS / AS aspects. Where appropriate, we remark to differences associated with CS / AS treatment in Sect. 3 & 4 (e.g. for SPRINTARS). However, a detailed investigation is beyond the scope of this paper.

Finally, please, revise the references to figures, tables and sections there is a mixture of formats.

Thank you for observing this, we have revised all references and tried to make reference to figures/tables/sections consistent.

Minor comments:
Page 2 Line 16: Capital letters in Aqua and Terra, i.e., MODIS-Aqua and MODIS-Terra.
This has been resolved in the revised version of the manuscript.

Page 2 Line 21: Capital letters in Terra, i.e., MODIS-Terra.
This has been resolved in the revised version of the manuscript.

Page 3 Lines 51-55: Add a reference as Boucher, O., Randall, D., Artaxo, P., Brether-ton, C., Feingold, G., Forster, P., ... & Rasch, P. (2013). Clouds and aerosols. In Climate change 2013: the physical science basis. Contribution of Working Group I to the Fifth Assessment Report of the Intergovernmental Panel on Climate Change (pp.571-657). Cambridge University Press.
The reference has been added accordingly.

Page 3 Lines 58-59: Add a reference.
This refers to the following sentence:

"Both natural and anthropogenic emissions are highly uncertain due to lack of measurements and information or documentation flow."

The sentence has been removed in the revised introduction section.

Page 3 Lines 64-65: Add a reference related to the quantification of DMS.

This refers to the following sentence:

"Marine dimethyl-sulfide (DMS) and volcanic emissions are responsible for approximately a third of the global anthropogenic sulphur budget."

The sentence has been removed in the revised introduction section.

Page 4 Line 83: Introduce the GAW acronym.
This has been resolved in the revised version of the manuscript.

**Page 4 Line 102: What are the advances between the set of models used in Kinne et al.(2006) and the ones considered in the present study? Is there any common/different feature between both studies? Is there any major improvement in the optical properties calculation from the modelling side?**

Advances compared to Kinne et al., 2006 have been collected in an additional question G9 in the optics questionnaire which is included as supplement 1. Main changes since AeroCom Phase I (also in the optics calculations) are now included in the text in Sect. 3 & 4, mostly based on the reported literature values in associated AP1 papers.

Page 4 Line 103: Remove "aerosol optical depth", it is already introduced.

This has been resolved in the revised version of the manuscript.

Page 5 Line 121: Introduce MEC and OD.

MEC and OD are now introduced at the end of the revised introduction section. In addition the new section 2.2.1 "Model diagnostics" introduces them in the scientific context.

Page 5 Line 137: In Figure 1 in the AATS-SU AE map, it is observed fine aerosols (high AE values, > 1) in Antarctica, could you add a comment on it?

The aerosol retrievals show highest accuracy over ocean and darker surfaces, with higher uncertainty over bright desert surfaces, or for measurements at large solar zenith angles (e.g. over Antarctica). We have added a comment to this effect in the manuscript. Please see the revised paragraph in Sect. 2.1.5 "AATSR SU v4.3 data" for comments related to uncertainties.

Page 6 Section 2.1.1: Consider to mention that AERONET SDA products (AOD coarse and AOD fine) are provided at 500nm. Also, you should include the number of sites used for the comparison in Table 1 and do reference to the location in Figure 1, for example.

More information about AERONET wavelengths are now provided in Table 1 and in the AERONET introduction section 2.1.1 in the revised manuscript. Number of stations are listed for each network/measurement platform - labeled '#st.' as described in the table caption. We have added letters to each pane in figure 1 and the individual panes are referenced when a particular measurement/network is discussed.

Page 6 Line 163: Introduce STP.

We have introduced STP including the assumed standard temperature and pressure values and tried to clarify the text about this adjustment (see also answer to comments from reviewer 1 above).

Pages 6-7 Section 2.2.2: You should include the number of the final selection of sites used for the comparison shown in Table 1, instead of mentioning the ones excluded and do reference of the location in Figure 1, for example.

We believe the reviewer refers to section 2.1.2, which introduces the surface in situ data. This section has been revised and includes now also a statement on the total number of sites considered for absorption and scattering. Please also note Section 1 in supplement which provides tables containing detailed information about each GAW site used in the climatological time-series.

**Changes to the manuscript:**

In addition to some further changes in Section 2.1.2 we added the following sentence:

*"After applying the RH constraint, removing urban sites from consideration, and resampling to monthly climatology, data from 39 sites with scattering data and from 39 sites with absorption data (not necessarily the same sites as for scattering) were available for model assessment (see Table 1)."*

Page 8 Lines 216: What Level are you considering in the study? I suppose that it is Level 3, but it is better to mention again here.

Yes, we used level 3 data and we clarified this in the revised paragraph 2.1.5 (before 2.1.4), by not mentioning level 2 anymore and stating the following.

**Changes to the manuscript:**

In addition to some further changes in Section 2.1.5 we added the following sentence:

"*This study uses the level 3 output, which is provided at daily and monthly 1x1 resolution, intended for climate model comparison.*"

Page 8 Section 2.1.5: It should be mentioned that MODIS and AATS products are considered inside this MERGED-FMI.

This has been clarified by adding the following sentence in Sect. 2.1.6 (before 2.1.5).

**Changes to the manuscript:**
"*It should be noted that MODIS and AATSR products are considered inside this MERGED-FMI data-set.*"

Page 8 Line 234: Table is in capital letters.

This has been resolved in the revised version of the manuscript.

Page 9 Line 255: Indicates that this is Appendix C

This has been resolved in the revised version of the manuscript. The content from Appendix C was now included in the new Section 2.3 "Data processing and statistics" as part of the major restructuring summarised above (i.e., the revised manuscript does not have any appendices anymore).

Page 9 Line 265: Specify the reference to cf. 1.

This has been resolved in the revised version of the manuscript.

Page 10 Lines 296-298: In Section 2.4, for the representativity of the results you are combining different models and observational datasets for concluding that "the overall difference is of the order of 10% and 0.2 for NMB and correlation, respectively". This is supposed shown in Table 3, but the numbers are not coincident.

We believe that there was a misunderstanding, perhaps the reviewer used an older pdf version of the manuscript that had this wrong reference to Table 3 (which was corrected between initial submission and the ACPD version, which refers to the correct table A2). Nonetheless, we remark that Table A2 is now included in the supplement 2 and the discussion of these results in Section 4.5 in the revised manuscript has also been updated.

Page 11 Section 2.5: Because you mention the bias to Europe and North America because of the density of sites. Is there any regional results in the comparison satellite vs AERONET that you can consider to include?

As discussed above we have removed section 2.5 (comparison of satellites with AERONET) and put some of the key findings in the individual satellite sections. Since most of the analysis is based on global results and most of our key findings based on the comparison with the ground based observations, we did not include any additional results of the satellite assessment.

Page 12 Line 341: Add Figure 3.

We are not sure what the reviewer means with this comment, however, this should be resolved as a result of the major restructuring applied.

Page 13 Line 370: Missing ".".
This was fixed (line 372 in original submission).

Page 13 Line 379: Extra space.

This was fixed (line 382 in original submission).

Table 2: Add a reference to the supplementary material (the excel table).

We added a sentence in the caption of table 2: More details about the models can be found in the supplementary material 1&2.

Figure 2: Replace the continuous colour palettes for a new one with categories as in Figure 1. It is easier for the reader to associate each colour to the corresponding category.

Figure 2 has been removed in the revised manuscript (see comments above for details).

Figures 4-8: Add the corresponding legend associated with the colours. Another possibility is to keep the numbers and use the colour scale to indicate what models are above/under the AeroCom median/mean which is the reference.

This has been resolved in the revised version of the manuscript, see summary above related to updates in these Figures. The Figures were merged into the new Table 3 in the revised manuscript and the colour coding has been updated indicating deviations from the median.

Figures 10-12: Replace the continuous colour palettes for a new one with categories as in Figure 9. It is easier for the reader to associate each colour to the corresponding category.

This has been resolved in the revised version of the manuscript.

Figures A1 and A2: The sites in the heatmap are organised by alphabetical order, to can distinguish a pattern, maybe it would be better to create clusters per continent, latitude, longitude. Replace the continuous colour palettes for a new one with categories as in Figure 9. It is easier for the reader to associate each colour to the corresponding category.

As discussed above, these Figures were removed in the revised manuscript, as a discussion of results at individual sites is beyond the scope of this global study.